# GIPC proteins negatively modulate Plexind1 signaling during vascular development

Jorge Carretero-Ortega[1]*, Zinal Chhangawala[1], Shane Hunt[1], Carlos Narvaez[1], Javier Menéndez-González[1], Carl M Gay[1], Tomasz Zygmunt[1], Xiaochun Li[2], Jesús Torres-Vázquez[1]*

[1]Department of Cell Biology, Skirball Institute of Biomolecular Medicine, New York University Langone Medical Center, New York, United States; [2]Department of Population Health, New York University School of Medicine, New York, United States

**Abstract** Semaphorins (SEMAs) and their Plexin (PLXN) receptors are central regulators of metazoan cellular communication. SEMA-PLXND1 signaling plays important roles in cardiovascular, nervous, and immune system development, and cancer biology. However, little is known about the molecular mechanisms that modulate SEMA-PLXND1 signaling. As PLXND1 associates with GIPC family endocytic adaptors, we evaluated the requirement for the molecular determinants of their association and PLXND1's vascular role. Zebrafish that endogenously express a Plxnd1 receptor with a predicted impairment in GIPC binding exhibit low penetrance angiogenesis deficits and antiangiogenic drug hypersensitivity. Moreover, *gipc* mutant fish show angiogenic impairments that are ameliorated by reducing Plxnd1 signaling. Finally, *GIPC* depletion potentiates SEMA-PLXND1 signaling in cultured endothelial cells. These findings expand the vascular roles of GIPCs beyond those of the Vascular Endothelial Growth Factor (VEGF)-dependent, proangiogenic GIPC1-Neuropilin 1 complex, recasting GIPCs as negative modulators of antiangiogenic PLXND1 signaling and suggest that PLXND1 trafficking shapes vascular development.
DOI: https://doi.org/10.7554/eLife.30454.001

*For correspondence:
carretero_2000@yahoo.com (JC-O);
jtorresv@med.nyu.edu (JT-V)

**Competing interests:** The authors declare that no competing interests exist.

## Introduction

Angiogenic sprouting, the formation of new vessels via branching of pre-existing ones, drives most of the life-sustaining expansion of the vascular tree. Sprouting angiogenesis is also pivotal for recovery from injury and organ regeneration and is often dysregulated in disease (*Carmeliet, 2003*; *Ramasamy et al., 2015*; *Cao, 2013*; *Fischer et al., 2006*). Among the pathways that modulate this process, SEMA-PLXN signaling plays a prominent role. In particular, the vertebrate-specific PLXND1 receptor transmits paracrine SEMA signals from the somites to the endothelium to shape the stereotypical and evolutionarily conserved anatomy of sprouts in the trunk's arterial tree (*Torres-Vázquez et al., 2004*; *Zygmunt et al., 2011*; *Gitler et al., 2004*; *Zhang et al., 2009*; *Gu et al., 2005*; *Childs et al., 2002*; *Epstein et al., 2015*). For instance, we have shown that antiangiogenic Plxnd1 signaling antagonizes the proangiogenic activity of the VEGF pathway to regulate fundamental features of vascular development (*Torres-Vázquez et al., 2004*; *Zygmunt et al., 2011*; see also *Moriya et al., 2010*; *Fukushima et al., 2011*; *Kim et al., 2011*). Specifically, Plxnd1 signaling acts cell autonomously in the endothelium to spatially restrict the aorta's angiogenic capacity to form sprouts, thereby determining both the positioning and abundance of the aortic Segmental (Se) vessels. Plxnd1 signaling also guides Se pathfinding, thus shaping these vascular sprouts.

Upon ligand binding, PLXNs act as guanosine triphosphatase-activating proteins (GAPs) and modulate integrin-mediated cell adhesion, cytoskeletal dynamics, and both ERK and MAPK signaling. However, there is limited molecular understanding of how PLXND1 exerts its vascular effects. Although a handful of putative intracellular effectors and modulators have been identified for this receptor (e.g., endocytic adaptors of the GIPC family, small GTPases belonging to the Ras and Rho families, GTPase regulators, and cytoskeletal proteins), clear functional links between these candidates and in vivo PLXND1-dependent angiogenic patterning remain to be established (*Zygmunt et al., 2011*; *Moriya et al., 2010*; *Fukushima et al., 2011*; *Choi et al., 2014*; *Horowitz, 2007*; *Uesugi et al., 2009*; *Gay et al., 2011*; *Sakurai et al., 2010*; *Sakurai et al., 2011*; *Wang et al., 2012*; *Deloulme et al., 2015*; *Worzfeld et al., 2014*; *Tata et al., 2014*; *Burk et al., 2017*; *Aghajanian et al., 2014*; *Wang et al., 2013*; *Pascoe et al., 2015*).

To address this gap, we focused on elucidating the relationship between GIPCs and antiangiogenic PLXND1 signaling. Based on our identification of GIPC1/Synectin as a PLXND1-binding protein, we previously showed by X-ray crystallography and co-immunoprecipitation (CoIP) assays that all three GIPC proteins can physically interact with the intracellular tail of the PLXND1 receptor and elucidated the structural organization of the GIPC-PLXND1 complex (*Gay et al., 2011*; *Shang et al., 2017*).

Here, we have further evaluated the molecular determinants of the PLXND1-GIPC interaction in the native environment of mammalian cells, firmly establishing a critical role for the last few amino acids of PLXND1 for the formation of the PLXND1-GIPC complex. Moreover, we defined the vascular role of PLXND1-GIPC association using zebrafish and cultured human umbilical vein endothelial cells (HUVECs) as model systems. Our results suggest that GIPCs act redundantly to limit the antiangiogenic signaling of the PLXND1 receptor.

In contrast, prior vascular studies of GIPCs have primarily centered on the role of GIPC1 as a positive modulator of VEGF (Vascular Endothelial Growth Factor) activity. The dominant model is that GIPC1 promotes arterial branching morphogenesis and arteriogenesis by associating with the cytosolic tail of the VEGF co-receptor NEUROPILIN-1 (NRP1), thereby facilitating the trafficking, and thus the signaling, of the VEGF receptor VEGFR2 (FLK1/KDR) (*Waters, 2009*; *Habeck et al., 2002*; *Wang et al., 2006*; *Ren et al., 2010*; *Covassin et al., 2009*; *Lanahan et al., 2013*; *Lawson et al., 2003*; *Hermans et al., 2010*; *Chittenden et al., 2006*; *Dedkov et al., 2007*; *Lanahan et al., 2010*, see also *Burk et al., 2017*).

Hence, our identification of GIPCs as negative intracellular regulators of PLXND1 signaling outlines a novel molecular mechanism by which these endocytic adaptors promote angiogenic development of the vertebrate arterial tree (*Chittenden et al., 2006*; *Lanahan et al., 2010*; *Fantin et al., 2011*; *Lampropoulou and Ruhrberg, 2014*; *Herzog et al., 2011*; *Plein et al., 2014*), highlighting the potential importance of PLXND1 trafficking for proper vascular development.

## Results

### Isolation of GIPC1/Synectin as a PLXND1-binding protein and dissection of the molecular determinants required for PLXND1-GIPC complex formation

To uncover intracellular modulators and effectors of PLXND1, we isolated proteins that bind to the cytosolic tail of wild-type (WT) mouse PLXND1 (C-mPLXND1$^{WT}$) using a yeast two-hybrid screen (*Fields and Song, 1989*). One of the recovered PLXND1-interacting preys (*Gay et al., 2011*) harbored the entirety of GIPC1 (GAIP interacting protein C-terminus or Synectin, among many other names), the founder of the three-membered family of GIPC endocytic adaptors (*Katoh, 2013*; *De Vries et al., 1998a*; *Gao et al., 2000*; *Naccache et al., 2006*; *Reed et al., 2005*). Remarkably, GIPC1, like PLXND1, modulates vascular development and acts within endothelial cells (*Zygmunt et al., 2011*; *Zhang et al., 2009*; *Moraes et al., 2013*; *Moraes et al., 2014*).

GIPCs are dimeric proteins with a central PDZ (PSD-95/Dlg/ZO-1) domain that binds to the C-terminal PDZ-Binding Motif (PBM) of transmembrane proteins harboring the type I PDZ-binding consensus sequence (S/T)-X-**A**/V/L/I. GIPCs also contain GH (GIPC Homology) domains at their amino (GH1), and carboxyl (GH2) ends. The GH1 domain promotes self-dimerization, while the GH2 domain binds the globular domain of MYOSIN6 (MYO6), the retrograde actin-based motor driving

endocytic vesicle internalization (*Shang et al., 2017*; *Katoh, 2013*; *Naccache et al., 2006*; *Salikhova et al., 2008*; *Lou et al., 2001*; *Cai and Reed, 1999*; *Gao et al., 2000*; *Aschenbrenner et al., 2003*). The divergent C-terminal region of Plxns (T-segment; 40–60 aa long) likely mediates subfamily-specific interactions between Plxns and other proteins. Indeed, the PLXND1-specific carboxyl sequence SEA-COOH is a consensus GIPC1-binding motif that serves as the PBM in the GIPC1-binding protein NRP1, a VEGF co-receptor (*Gay et al., 2011*; *Lanahan et al., 2013*; *Lanahan et al., 2010*; *Wang et al., 2006*; *Horowitz and Seerapu, 2012*; *Lanahan et al., 2014*; *Chittenden et al., 2006*).

To understand how PLXND1 and GIPC1 interact physically we performed a crystallographic study with bacterially purified recombinant mouse proteins. This structural analysis revealed that the receptor's nonameric C-terminal sequence NIYECYSEA-COOH (residues 1917–1925) is the primary GIPC-Binding Motif (GBM) and makes N- and C-terminal contacts with GIPC1's GH1 and PDZ domains. A pair of receptor's helices (residues 1893–1908) and the GIPC1's PDZ domain form a secondary heteromeric interface. Pulldown experiments of GIPC1 by MYO6 in the presence of PLXND1 indicate that MYO6 associates with PLXND1-bound GIPC1 and that deletion of the receptor's canonical PBM (PLXND1ΔSEA) or I1918A-Y1919A GBM substitutions significantly reduces PLXND1-GIPC1 binding. Conversely, the GIPC1 G114Y mutation disrupting the interaction between PLXND1's I1918/Y1919 GBM residues and GIPC1's GH1 hydrophobic pocket also impairs PLXND1-GIPC1 binding (*Shang et al., 2017*).

Using exogenous expression of murine proteins and a eukaryotic cellular environment, we interrogated the requirement for PLXND1's canonical PBM and nonameric GBM and also tested the involvement of GIPC1's PDZ and GH1 domains. To this end, we performed CoIP experiments with COS7 cells, which lack endogenous expression of both PLXND1 (*Gu et al., 2005*; *Uesugi et al., 2009*; *Sakurai et al., 2010*; *Takahashi et al., 1999*) and NRP1 (*Tordjman et al., 2002*) (*Figure 1*; see also *Supplementary file 1* and *Supplementary file 2*).

To address the requirement for PLXND1's canonical PBM and nonameric GBM, we used V5-tagged PLXND1 cytosolic tails (*Figure 1A*), either with (V5-C-mPLXND1$^{WT}$) or without (V5-C-mPLXND1Δ$^{CYSEA}$) the canonical PBM. For GIPC1, we used a FLAG-tagged version of the full protein, FLAG-mGIPC1$^{WT}$ (*Figure 1B*). We found that deletion of PLXND1's canonical PBM (V5-C-mPLXND1Δ$^{CYSEA}$) significantly reduced binding to FLAG-mGIPC1$^{WT}$ (*Figure 1C*, lanes 2–3 and graph below), consistent with the existence of additional contacts, as suggested by our crystallographic findings (*Shang et al., 2017*; see also *Lee and Zheng, 2010*). Accordingly, the PLXND1 form lacking the GBM motif (V5-C-mPLXND1Δ$^{GBM}$) shows a further reduction in FLAG-mGIPC1$^{WT}$-binding capacity (*Figure 1C*, lane 4 and graph below). Importantly, the minimum GIPC1-binding capacity of the V5-C-mPLXND1Δ$^{GBM}$ form observed under these protein over-expression conditions suggests that the secondary GIPC-binding interface formed by the C-terminal pair of helices of PLXND1 (*Shang et al., 2017*) plays a minor role as a molecular determinant for PLXND1-GIPC1 complex formation.

To test the involvement of the two GIPC1 domains that interact with PLXND1 in the crystal structure (*Shang et al., 2017*), we used FLAG-tagged GIPC1 fragments harboring the GH1 and PDZ domains individually (FLAG-mGIPC1$^{GH1}$ and FLAG-mGIPC1$^{PDZ}$, respectively) and V5-C-mPLXND1$^{WT}$. Our CoIP data confirmed that each of these domains is sufficient for PLXND1 association (*Figure 1D*). The ability of GIPC1's GH1 domain to interact with PLXND1 is consistent with prior observations implicating GIPC1's N-terminal region in binding to other transmembrane proteins (*Naccache et al., 2006*; *Giese et al., 2012*).

Taken together, the data from our yeast two-hybrid (*Gay et al., 2011*) and crystallographic (*Shang et al., 2017*) studies demonstrate that PLXND1 and GIPC1 can interact directly and that NRP1 is dispensable for their interaction. These findings are consistent with our CoIP of PLXND1-GIPC1 complexes from COS7 cells, which lack endogenous NRP1 expression (see *Tordjman et al., 2002*). Together with our dissection of the molecular determinants of PLXND1-GIPC1 complex formation, the high sequence identity of GIPCs, and our observation that GIPC2 and GIPC3 also associate with PLXND1 (*Shang et al., 2017*), these results establish the critical importance of PLXND1's GBM and GIPC's GH1 and PDZ domains for PLXND1-GIPC complex formation.

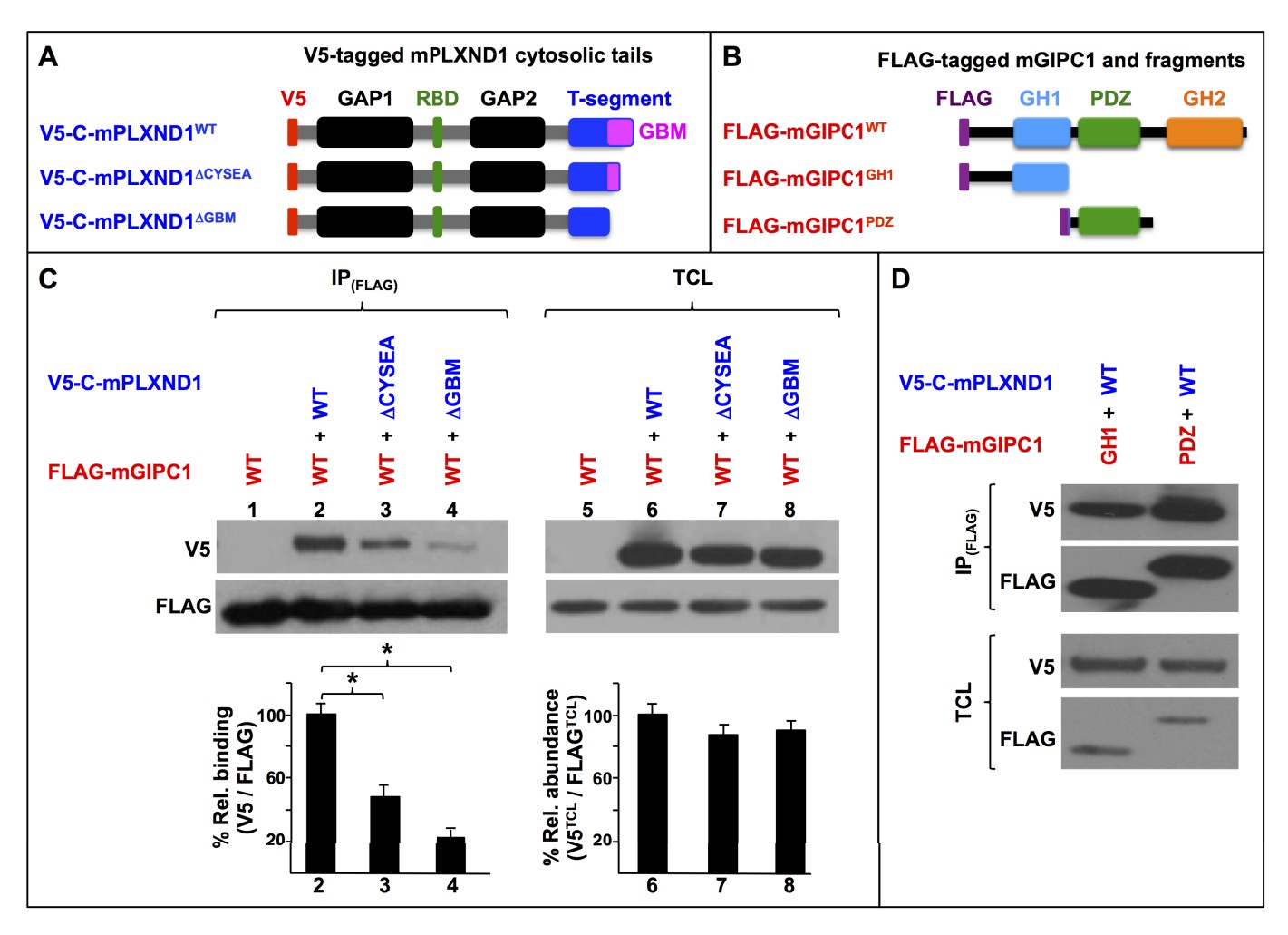

**Figure 1.** PLXND1's C-terminal GBM and GIPC's GH1 and PDZ domains are the key molecular determinants of the PLXND1-GIPC interaction. (A) Diagrams of the wild-type (WT) and truncated V5-tagged (red) forms of the cytosolic tail of murine PLXND1 (V5-C-mPLXND1) used for co-immunoprecipitation experiments. Color-coding is used to highlight the following domains and motifs. GAP1 and GAP2 (Guanosine triphosphatase-Activating Protein domains 1 and 2; black), RBD (Rho GTPase-Binding Domain; green), T-segment (C-terminal segment, includes the GBM; blue) and, GBM (GIPC-Binding Motif; magenta); see (*Gay et al., 2011*). (B) Diagrams of the wild-type (WT) and truncated FLAG-tagged (purple) forms of murine GIPC1 (FLAG-mGIPC) used for co-immunoprecipitation experiments. Domains indicated as follows: PDZ (PSD-95/Dlg/ZO-1; green) and GH (GIPC Homology domain) 1 (blue) and 2 (orange); see (*Katoh, 2013*). (C) Western blots (top) and their quantification (bottom, bar graphs). Numbers indicate lane positions. Left-side Western blot (IP$_{FLAG}$): FLAG immunoprecipitates and V5 co-immunoprecipitates showing interactions between the indicated V5-C-mPLXND1 and FLAG-mGIPC forms. Right-side Western blot (TCL), expression levels of these proteins in total cell lysates as detected with V5 and FLAG antibodies. Quantifications. *n* = 3 independent experiments for each protein pair. Left-side bar graph (C). Means of percentual V5/FLAG relative binding [(V5$^{CoIP}$/V5$^{TCL}$)/(FLAG$^{IP}$/FLAG$^{TCL}$)] between the indicated protein pairs from the IP$_{FLAG}$ Western blot (top left) and the relative abundance of the expression levels of these proteins from the TCL Western blot (top right). Error bars, ± SEM. V5/FLAG relative binding was significantly different (p<0.05) between V5-C-mPLXND1 forms and FLAG-mGIPC, $F_{(2, 6)}=22.376$, p=0.002, as determined by a one-way ANOVA test. A Tukey post hoc analysis was conducted to determine whether the percentual V5/FLAG relative binding between the three tested pairs of proteins was significantly different (p<0.05; asterisks). Right-side bar graph (C). Means of percentual V5$^{TCL}$/FLAG$^{TCL}$ relative abundance between the indicated protein pairs from the TCL Western blot (top right). Error bars, ± SEM. A Kruskal-Wallis H test was conducted to determine significant differences (p<0.05) in V5$^{TCL}$/FLAG$^{TCL}$ relative abundance between the indicated protein pairs. Distributions of V5$^{TCL}$/FLAG$^{TCL}$ relative abundance were not similar for all groups. The medians of V5$^{TCL}$/FLAG$^{TCL}$ relative abundances were 92.64 (for V5-C-mPLXND1$^{WT}$/FLAG-mGIPC1), 87.79 (for V5-C-mPLXND1Δ$^{CYSEA}$/FLAG-mGIPC1), and 96.22 (for V5-C-mPLXND1Δ$^{GBM}$/FLAG-mGIPC1), but were not statistically significantly different between them (*Ramasamy et al., 2015*), $\chi_{(2)}=0.8$, p=0.670. (D) Western blots: Top (IP$_{FLAG}$), FLAG immunoprecipitates and their V5 co-immunoprecipitates showing interactions between the indicated V5-C-mPLXND1 and FLAG-mGIPC forms; bottom, (TCL) detection of the expression levels of these proteins in total cell lysates using antibodies against V5 and FLAG. *n* = 3 independent experiments for each protein pair. For additional data and statistical comparisons related to this figure, see *Supplementary file 1*, *Supplementary file 2* and *Supplementary file 8*.

DOI: https://doi.org/10.7554/eLife.30454.002

# Zebrafish *plxnd1^skt6* homozygous mutants, which express a Plxnd1 receptor with a predicted impairment in GIPC binding, display angiogenesis deficits with low frequency

To determine the role that GIPC binding exerts on antiangiogenic PLXND1 signaling, we sought to specifically impair PLXND1's ability to associate with GIPC endocytic adaptors in an in vivo model of vascular development. To do this, we performed CRISPR/Cas9-based genome editing (*Auer and Del Bene, 2014*; *Auer et al., 2014*; *Chang et al., 2013*; *Cong et al., 2013*; *Cong and Zhang, 2015*; *Gagnon et al., 2014*; *Hill et al., 2014*; *Hruscha et al., 2013*; *Hwang et al., 2013*; *Irion et al., 2014*; *Kimura et al., 2014*; *Mali et al., 2013*; *Talbot and Amacher, 2014*) of the last coding exon of the zebrafish *plxnd1* locus to introduce disrupting mutations into the receptor's GBM (NIYECSSEA-COOH, canonical PBM underlined; *Figure 2A*). The resulting *plxnd1^skt6* allele encodes a Plxnd1 receptor missing the PBM because of replacement of the five C-terminal residues by a stretch of 31 amino acids (*Figure 2B*; see also *Supplementary file 1* and *Supplementary file 8*). Because adding just a single C-terminal residue to the PBM of proteins that interact with PDZ domain-containing partners is sufficient to block their cognate association (*Rickhag et al., 2013*; *Saras et al., 1997*; *Cao et al., 1999*; *Garbett and Bretscher, 2012*), and deletion of PLXND1's PBM reduces GIPC binding significantly (*Figure 1A–C*; see also *Shang et al., 2017*), the *plxnd1^skt6* mutant allele is expected to encode a Plxnd1 receptor with reduced or null GIPC-binding ability.

In the trunk of WT embryos, Se sprouts arise bilaterally from the Dorsal Aorta (DA) just anterior to each somite boundary (SB) at 21 h post-fertilization (hpf). Se sprouts grow dorsally with a chevron shape connecting with their ipsilateral neighbors above the spinal cord's roof to form the paired Dorsal Longitudinal Anastomotic Vessels (DLAVs) by 32 hpf (*Isogai et al., 2003*) (*Figure 2C*). In contrast, null *plxnd1* alleles, such as *plxnd1^fov01b*, show a recessive and hyperangiogenic mutant phenotype characterized by a dramatic mispatterning of the trunk's arterial tree anatomy. As such, the homozygous mutants form too many, and misplaced, Se sprouts that branch excessively and interconnect ectopically (*Figure 2D*) as a result of increased proangiogenic VEGF activity (*Torres-Vázquez et al., 2004*; *Zygmunt et al., 2011*).

GIPCs limit or promote the activity of their transmembrane consorts by regulating their surface expression, G-protein signaling, and vesicular trafficking (*Lanahan et al., 2010*; *De Vries et al., 1998a*; *Naccache et al., 2006*; *Reed et al., 2005*; *Giese et al., 2012*; *Lou et al., 2001*; *Bunn et al., 1999*; *Wieman et al., 2009*; *Blobe et al., 2001*; *Chak and Kolodkin, 2014*; *Hu et al., 2003*; *Booth et al., 2002*; *Jeanneteau et al., 2004a*; *Arango-Lievano et al., 2016*; *Varsano et al., 2012*). Whether GIPC binding limits, versus enables, PLXND1 signaling gives different predictions regarding the nature of the *plxnd1^skt6* allele, which can be assessed by its ability to complement the *plxnd1^fov01b* null mutation, the vascular phenotype of *plxnd1^skt6* homozygotes and their sensitivity to antiangiogenic compounds. If GIPCs limit PLXND1's antiangiogenic signaling, *plxnd1^skt6* should be a *plxnd1^fov01b*-complementing hypermorph, and *plxnd1^skt6* homozygotes have angiogenesis deficits and SU5416 hypersensitivity. If GIPCs directly enable PLXND1's antiangiogenic signaling (as in *Burk et al., 2017*), then *plxnd1^skt6* should be a null incapable of complementing *plxnd1^fov01b*, and *plxnd1^skt6* mutants show excessive and disorganized angiogenesis and SU5416 hyposensitivity. If PLXND1 indirectly exerts a GIPC-dependent antiangiogenic effect by sequestering GIPCs to antagonize GIPC-dependent proangiogenic VEGFR2/Nrp1-mediated VEGF signaling (see *Gay et al., 2011*; *Lanahan et al., 2013*; *Lanahan et al., 2010*; *Wang et al., 2006*; *Horowitz and Seerapu, 2012*; *Lanahan et al., 2014*; *Chittenden et al., 2006*), then *plxnd1^skt6* should be a hypomorph with partial or full *plxnd1^fov01b* complementing capacity, and *plxnd1^skt6* homozygotes should have hyperangiogenic or normal vascular phenotypes and reduced or normal SU5416 sensitivity. Finally, if GIPCs are irrelevant for PLXND1 signaling, then *plxnd1^skt6* should fully complement *plxnd1^fov01b*, and *plxnd1^skt6* homozygotes should show proper vascular patterning, normal angiogenic growth, and WT-like SU5416 sensitivity.

A comparative analysis of WT embryos, *plxnd1^fov01b* homozygous mutants, and *plxnd1^skt6*/*plxnd1^fov01b* transheterozygotes (*Figure 2C–E*) revealed that, like the WT, the transheterozygotes display a properly patterned arterial tree and normal Se and DLAV angiogenic development (*Figure 2C,E*; see also *Supplementary file 3* and Materials and methods). These observations argue against the model that GIPCs directly enable PLXND1's antiangiogenic signaling (see *Burk et al.,*

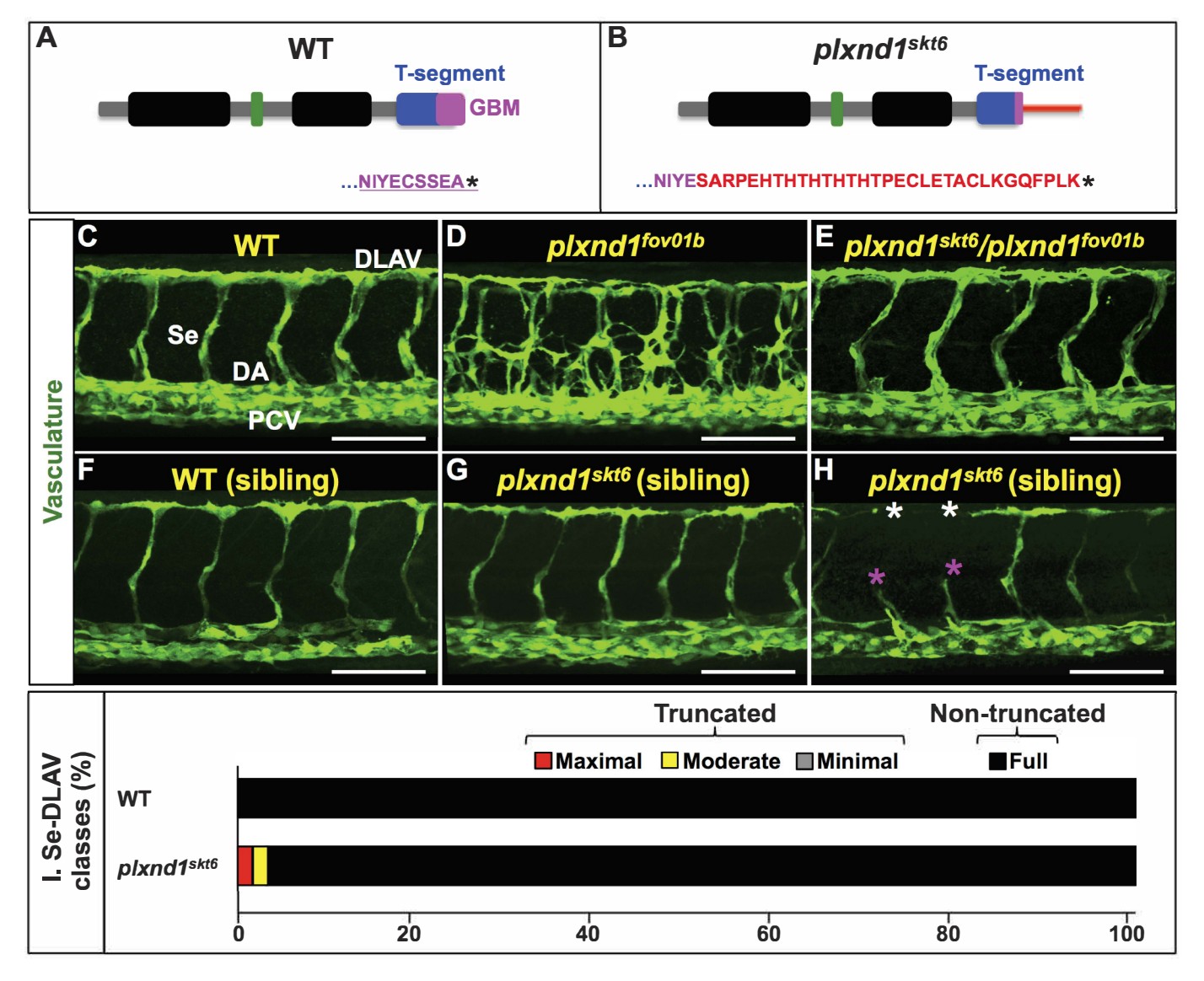

**Figure 2.** The *plxnd1*^*skt6* allele encodes a functional Plxnd1 receptor putatively impaired in GIPC binding, and its homozygosity induces angiogenesis deficits with low frequency. (A, B) Diagrams of the cytosolic tails of the zebrafish Plxnd1 proteins encoded by the WT (A) and *plxnd1*^*skt6* mutant (B) alleles including their C-terminal amino acid sequences. Color-coding is used to highlight the following domains and motifs. GAP1 and GAP2 (Guanosine triphosphatase-Activating Protein domains 1 (left) and 2 (right); black), RBD (Rho GTPase-Binding Domain; green), T-segment (C-terminal segment, includes the GBM; blue), and GBM (GIPC-Binding Motif; magenta). In the WT protein diagram (A), the canonical PBM (PDZ-Binding Motif) is underlined. In the mutant protein diagram (B), the thin horizontal red bar denotes the amino acid sequence replacing the PBM. (C–H) Confocal lateral images of the trunk vasculature (green) of 32 hpf embryos (region dorsal to the yolk extension). Anterior, left; dorsal, up. Scale bars (white horizontal lines), 100 µm. Genotypes indicated on top of each image in yellow font. Angiogenesis deficits are indicated with asterisks as follows: white (DLAV gaps), magenta (truncated Se). In the WT image (C), the vessels are designated with the white font as follows: DLAV (Dorsal Longitudinal Anastomotic Vessel), Se (Segmental Vessel), DA (Dorsal Aorta), and PCV (Posterior Cardinal Vein). The homozygous WT and homozygous *plxnd1*^*skt6* mutant embryos (F–H) are siblings derived from the incross of *plxnd1*^*skt6*/+heterozygotes. (I) Bar graph. Percentage of Se-DLAV in 32 hpf embryos of the indicated genotypes belonging to each of the following four phenotypic classes. Truncated: maximal (red; includes missing Se), moderate (yellow), and minimal (gray). Non-truncated: Full (black). There was no statistically significant difference in the distribution of the four phenotypic classes between WT and *plxnd1*^*skt6* mutants as assessed by a two-sided Fisher Exact test, p=0.05905. Quantifications. To determine whether *plxnd1*^*skt6* complements the *plxnd1*^*fov01b* null, we analyzed vascular patterning (C–E) and scored Se-DLAV angiogenesis (C, E) in embryos of the following three genotypes: WT (124 Se-DLAV, 11 embryos, an average of 11.27 Se-DLAV/embryo), *plxnd1*^*fov01b* homozygotes (12 embryos), and *plxnd1*^*fov01b*/*plxnd1*^*fov01b* transheterozygotes (162 Se-DLAV, 16 embryos, an average of 10.13 Se-DLAV/embryo). All the WT and transheterozygotes displayed proper vascular patterns indistinguishable from each other and lacked Se-DLAV truncations. All the *plxnd1*^*fov01b* mutants displayed hyperangiogenic vascular mispatterning. To determine how Plxnd1's inability to interact with GIPCs impacts angiogenic growth, we scored Se-DLAV angiogenesis (F–I) in sibling embryos of the

*Figure 2 continued on next page*

*Figure 2 continued*

following two genotypes: Homozygous WT (126 Se-DLAV, 12 embryos, an average of 10.5 Se-DLAV/embryo) and *plxnd1^skt6* homozygous mutants (124 Se-DLAV, 12 embryos, an average of 10.33 Se-DLAV/embryo). For additional data, graphs and statistical comparisons related to this figure, see *Figure 2—figure supplement 1* and *Supplementary file 3*.

DOI: https://doi.org/10.7554/eLife.30454.003

The following figure supplements are available for figure 2:

**Figure supplement 1.** Penetrance and expressivity of Se-DLAV truncations in *plxnd1^skt6* mutants at 32 hpf.

DOI: https://doi.org/10.7554/eLife.30454.004

**Figure supplement 2.** A Plxnd1 form deficient in GIPC binding because of deletion of the receptor's GBM (Plxnd1Δ^GBM) is active in vivo.

DOI: https://doi.org/10.7554/eLife.30454.005

*2017*). We conclude that the novel *plxnd1^skt6* allele complements the *plxnd1^fov01b* null and encodes a functional receptor capable of antiangiogenic signaling.

Further support for the notion that reducing Plxnd1's ability to associate with GIPCs does not inactivate the receptor comes from experiments driving mosaic transgenic endothelial expression of N-terminally tagged forms of zebrafish Plxnd1 in *plxnd1^fov01b* null mutants (*Figure 2—figure supplement 2*). Briefly, we found that the WT form (2xHA-Plxnd1^WT) and a receptor lacking the entire GBM because of removal of the nine C-terminal residues (2xHA-Plxnd1Δ^GBM) had the same capacity to cell-autonomously rescue the vascular defects of *plxnd1^fov01b* homozygotes (*Figure 2—figure supplement 2D–E,G–H,J*). In other words, endothelial cells within Se exogenously expressing either receptor form were absent from ectopic Se, had WT shapes concordant with their position within Se and DLAVs, and, when found within the base of sprouts, were positioned correctly just anterior to somite boundaries. These are the same features that WT endothelial cells display when transplanted into *plxnd1^fov01b* hosts; see (*Zygmunt et al., 2011*).

This experiment does not clarify whether impairing Plxnd1's GIPC-binding ability either increases or does not affect Plxnd1's signaling. To address this point, we compared homozygous WT and homozygous *plxnd1^skt6* mutant sibling embryos derived from incrosses of *plxnd1^skt6*/+heterozygotes. We found that *plxnd1^skt6* homozygous mutants showed an adequately organized vascular tree and a low frequency of Se and DLAV truncations (*Figure 2G–I*; see also the Materials and methods section 'Quantification of angiogenesis deficits in the trunk's arterial tree of WT and mutant zebrafish embryos'). However, the distributions of complete and truncated vessels and both the penetrance and expressivity of these defects were not statistically significantly different between WT and *plxnd1^skt6* mutants (*Figure 2I*, *Figure 2—figure supplement 1* and *Supplementary file 3*). The unique presence of angiogenic deficits in the mutants, such as maximal and moderate Se and DLAV truncations (*Figure 2H–I*), suggests that the *plxnd1^skt6* allele is hypermorphic.

## Homozygous *plxnd1^skt6* mutants are hypersensitive to the antiangiogenic drug SU5416

The results above show that the receptor encoded by the *plxnd1^skt6* allele is active, but do not fully clarify the nature of the *plxnd1^skt6* allele and the precise role of GIPCs in antiangiogenic PLXND1 signaling. Hence, we compared the sensitivity of homozygous WT and homozygous *plxnd1^skt6* mutants to the antiangiogenic compound SU5416, a tyrosine kinase inhibitor that preferentially targets VEGR2 (*Herbert et al., 2012*; *Covassin et al., 2006*; *Fong et al., 1999*).

If GIPCs limit PLXND1 antiangiogenic signaling, then *plxnd1^skt6* homozygotes should be hypersensitive to SU5416. In contrast, *plxnd1^skt6* mutants should be hyposensitive to SU5416 if GIPCs bind PLXND1 to directly enable the receptor's antiangiogenic signaling (as in *Burk et al., 2017*) or if PLXND1 indirectly exerts a GIPC-dependent antiangiogenic effect by sequestering rate-limiting GIPCs away from proangiogenic VEGFR2/Nrp1 signaling (see *Gay et al., 2011*; *Lanahan et al., 2013*; *Lanahan et al., 2010*; *Wang et al., 2006*; *Horowitz and Seerapu, 2012*; *Lanahan et al., 2014*; *Chittenden et al., 2006*). Finally, if GIPCs play no role in PLXND1 signaling, then WT and *plxnd1^skt6* mutants should have similar SU5416 sensitivities.

As a first step, we evaluated the impact of 18–32 hpf SU5416 treatments (0.1–1 μM range) on WT angiogenesis (not shown), which defined a suboptimal SU5416 dose (0.2 μM; in 0.1% DMSO vehicle) capable of inducing minimal antiangiogenic effects on Se and DLAV development by 32 hpf (as in *Covassin et al., 2006*; *Stahlhut et al., 2012*; see also *Torres-Vázquez et al., 2004*; *Zygmunt et al.,*

*2011*; *Childs et al., 2002*; *Isogai et al., 2003*; *Zygmunt et al., 2012*; *Yokota et al., 2015*). Accordingly, we treated homozygous WT and homozygous *plxnd1skt6* mutants with 0.2 µM SU5416 (experimental group) or 0.1% DMSO (control group); see *Figure 3*, *Figure 3—figure supplement 1* and *Supplementary file 4*.

DMSO and SU5416 had no apparent effects on embryonic shape, size, somite number, cardiac contractility, and circulation (not shown). DMSO slightly impaired Se and DLAV angiogenesis in both genotypes, consistent with its effects at higher doses (*Hallare et al., 2006*; *Chen et al., 2011*; *Maes et al., 2012*). The weak angiogenic impairment induced by DMSO was greater in *plxnd1skt6* mutants, albeit without a significant penetrance difference (*Figure 3A–B,E*, *Figure 3—figure supplement 1A* and *Supplementary file 4*).

As expected, SU5416 increased the frequency and severity of angiogenesis deficits in both genotypes (*Figure 3C–E*, *Figure 3—figure supplement 1* and *Supplementary file 4*). WT embryos and *plxnd1skt6* mutants treated with DMSO and SU5416 exhibited statistically significantly different distributions of Se and DLAV truncations (*Figure 3E* and *Supplementary file 4*). Notably, DMSO-treated WT embryos and *plxnd1skt6* mutants showed angiogenic deficits at statistically significantly different proportions, but only when comparing the 'minimal' category or the collective grouping of the three defective categories. In contrast, SU5416-treated WT embryos and *plxnd1skt6* mutants displayed angiogenic deficits at statistically significantly different proportions when comparing the three defective categories, either individually or as a group (*Figure 3E* and *Supplementary file 4*). For example, while SU5416 induced a non-significant penetrance difference in Se-DLAV truncations between WT embryos (89.7%) and *plxnd1skt6* mutants (100%) (*Figure 3—figure supplement 1A*), it significantly and disproportionally increased the frequency and severity of Se-DLAV angiogenesis deficits in *plxnd1skt6* mutants (*Figure 3E*, *Figure 3—figure supplement 1B* and *Supplementary file 4*). In particular, SU5416 disproportionally increased the frequency of the most severe class of angiogenesis deficit in *plxnd1skt6* mutants (maximal Se-DLAV truncation), a defect unique to SU5416-treated embryos of both genotypes (*Figure 3E*).

These effects indicate that *plxnd1skt6* mutants are hypersensitive to SU5416. This conclusion fits the prediction of a single hypothesis, namely that GIPC binding limits PLXND1 signaling, and implies that the *plxnd1skt6* allele hypermorphically increases antiangiogenic PLXND1 signaling.

## Zebrafish *gipc* mutants display angiogenesis deficits in the trunk's arterial tree

Mammalian studies implicate GIPC1 in promoting the NRP1-mediated proangiogenic activity of VEGF in the arterial tree (*Waters, 2009*; *Habeck et al., 2002*; *Wang et al., 2006*; *Ren et al., 2010*; *Covassin et al., 2009*; *Lanahan et al., 2013*; *Lawson et al., 2003*; *Hermans et al., 2010*; *Chittenden et al., 2006*; *Dedkov et al., 2007*; *Lanahan et al., 2010*). Vertebrates have three GIPCs, which are highly identical (*Katoh, 2013*), suggesting their potential functional redundancy. Indeed, in assays with recombinant mouse proteins purified from bacteria, all GIPCs can bind to PLXND1 (*Shang et al., 2017*). Impairing GIPC-PLXND1 complex formation at the receptor level, as in *plxnd1skt6* mutants, yields angiogenesis deficits (*Figure 2G–I*) and SU5416 hypersensitivity (*Figure 3*). These defects fit the hypothesis that GIPCs promote arterial angiogenic development, at least in part, by limiting antiangiogenic PlxnD1 signaling. Given these findings, we anticipated that genetic inactivation of *gipc*s would similarly impair angiogenic development in the zebrafish arterial tree.

To define the vascular roles of zebrafish GIPCs, we used CRISPR/Cas9-based genome editing (*Auer and Del Bene, 2014*; *Auer et al., 2014*; *Chang et al., 2013*; *Cong et al., 2013*; *Cong and Zhang, 2015*; *Gagnon et al., 2014*; *Hill et al., 2014*; *Hruscha et al., 2013*; *Hwang et al., 2013*; *Irion et al., 2014*; *Kimura et al., 2014*; *Mali et al., 2013*; *Talbot and Amacher, 2014*) to make the first *gipc1/synectin* (*gipc1skt1* and *gipc1skt2*), *gipc2* (*gipc2skt3* and *gipc2skt4*), and *gipc3* (*gipc3skt5*) zebrafish mutants. Our novel *gipc* alleles are putative nulls predicted to encode short, truncated proteins lacking all domains (*Figure 4—figure supplement 1*; see also *Supplementary file 8*), and are therefore unable to interact with themselves, each other, Plxnd1, and Myo6; see (*Shang et al., 2017*).

Given the reported coexpression of GIPC1 and GIPC2 in the endothelium (*Wang et al., 2006*; *Chittenden et al., 2006*; *Jiang et al., 2013*; *Butler et al., 2016*), we focused on the vascular phenotype of the corresponding single and double zebrafish mutants in both zygotic and maternal-zygotic (MZ) combinations at 32 hpf. We scored 112 *gipc* mutants (69 embryos without maternal and zygotic

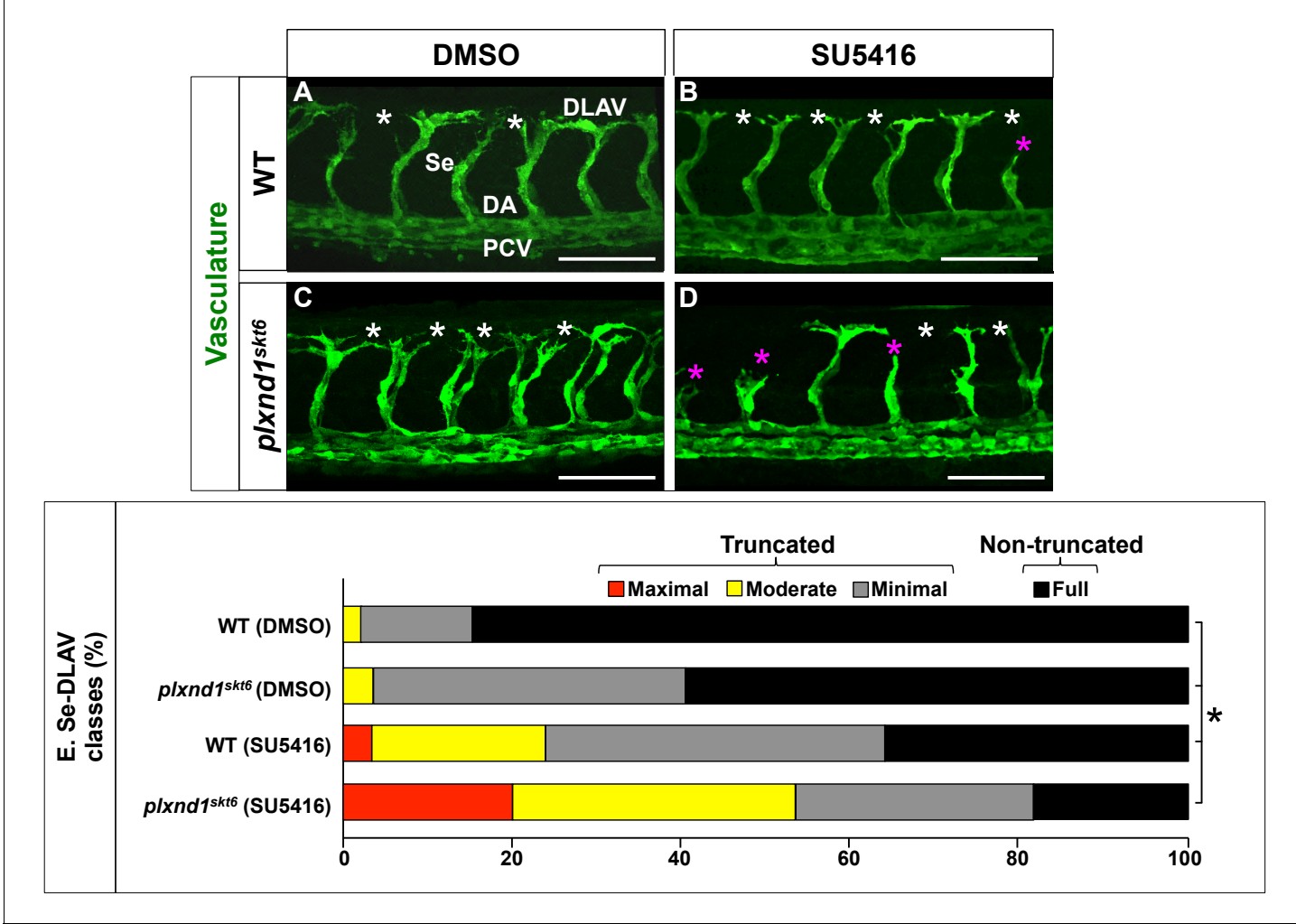

**Figure 3.** *plxnd1^{skt6}* mutants are hypersensitive to the antiangiogenic drug SU5416. (A–D) Confocal lateral images of the trunk vasculature (green) of 32 hpf embryos (region dorsal to the yolk extension). Anterior, left; dorsal, up. Scale bars (white horizontal lines), 100 μm. Treatments (DMSO or SU5416) indicated on top, genotypes (WT or *plxnd1^{skt6}*) indicated on the left. Angiogenesis deficits are indicated with asterisks as follows: white (DLAV gaps), magenta (truncated Se). In the image of the DMSO-treated WT (C), the vessels are designated with the white font as follows: DLAV (Dorsal Longitudinal Anastomotic Vessel), Se (Segmental Vessel), DA (Dorsal Aorta), and PCV (Posterior Cardinal Vein). (E) Bar graph. Percentage of Se-DLAV in 32 hpf embryos of the indicated genotype and treatment combinations belonging to each of the following four phenotypic classes. Truncated: maximal (red; includes missing Se), moderate (yellow), and minimal (gray). Non-truncated: Full (black). The distributions of these four phenotypic classes were statistically significantly different between all the possible pairwise comparisons of the four combinations of treatments and genotypes. Significance values were calculated using a two-sided Fisher Exact test and significant differences (p<0.0083) assigned using a Bonferroni-type adjustment for six pairwise genotype comparisons (0.05/6 = 0.0083). Quantifications. We scored Se-DLAV angiogenesis (A–E) in embryos of the following four combinations of treatments and genotypes: DMSO-treated WT (312 Se-DLAV, 28 embryos, an average of 11.14 Se-DLAV/embryo), DMSO-treated *plxnd1^{skt6}* (284 Se-DLAV, 26 embryos, an average of 10.92 Se-DLAV/embryo), SU5416-treated WT (322 Se-DLAV, 29 embryos, an average of 11.10 Se-DLAV/embryo), and SU5416-treated *plxnd1^{skt6}* (320 Se-DLAV, 28 embryos, an average of 11.43 Se-DLAV/embryo). For additional data, graphs and statistical comparisons related to this figure, see *Figure 3—figure supplement 1* and *Supplementary file 4*.

DOI: https://doi.org/10.7554/eLife.30454.006

The following figure supplement is available for figure 3:

**Figure supplement 1.** Penetrance and expressivity of Se-DLAV truncations in DMSO-treated and SU5416-treated WT and embryos and *plxnd1^{skt6}* mutants at 32 hpf.

DOI: https://doi.org/10.7554/eLife.30454.007

*gipc1* activity, including 19 animals devoid of zygotic *gipc2* activity; plus 24 embryos lacking zygotic *gipc1* activity, including 13 without *gipc2* activity). This analysis revealed that *gipc* mutants display recessive Se angiogenic deficits of partial penetrance (*Figure 4*, *Figure 4—figure supplement 2* and *Figure 4—figure supplement 3*; see also Figure 6C, *Supplementary file 5*, *Supplementary file 6* and Materials and methods). Specifically, the Se vessels were sometimes missing or truncated, leading to corresponding DLAV gaps. At times, they were abnormally thin and straight-shaped. Importantly, every *gipc* mutant lacked supernumerary, ectopic, or misguided Se vessels, the hallmark defects of zebrafish and mice *Plxnd1* nulls (*Torres-Vázquez et al., 2004*; *Zygmunt et al., 2011*; *Gitler et al., 2004*; *Zhang et al., 2009*; *Gu et al., 2005*; *Worzfeld et al., 2014*) and, reportedly, of murine *Gipc1* mutants (*Burk et al., 2017*). We also found that as maternal or zygotic *gipc* dosage decreases, the penetrance and expressivity of Se truncations tended to increase (*Figure 4—figure supplement 3*). Together, these findings indicate that *gipc1* and *gipc2* act redundantly and that, at the zygotic level, loss of *gipc1* causes a greater angiogenesis deficit than the removal of *gipc2* (*Figure 4*, *Figure 4—figure supplement 2* and *Figure 4—figure supplement 3*).

Overall, the angiogenic deficits of the single (*gipc1* and *gipc2*) and double (*gipc1; gipc2*) zebrafish mutants resemble those found in *gipc1* morphants (but without the delayed vasculogenesis, small aortic lumen, and abnormal body shape of the latter) (*Ren et al., 2010*; *Hermans et al., 2010*; *Chittenden et al., 2006*) and postnatal *Gipc1* mutant mice (impaired arterial branching and arteriogenesis) (*Ren et al., 2010*; *Chittenden et al., 2006*; *Dedkov et al., 2007*; *Lanahan et al., 2010*; *Moraes et al., 2013*; *Lanahan et al., 2014*; *Paye et al., 2009*). They are also similar to the vascular defects found in *kdrl* mutant fish (in which inactivation of a VEGFR2 ohnolog leads to reduced VEGF signaling) (*Habeck et al., 2002*; *Covassin et al., 2009*; *Bussmann et al., 2008*) and *plxnd1*^skt6^ homozygotes.

These observations are consistent with the involvement of GIPCs in VEGF-induced proangiogenic VEGFR2/Nrp1-mediated signaling (see *Gay et al., 2011*; *Lanahan et al., 2013*; *Lanahan et al., 2010*; *Wang et al., 2006*; *Horowitz and Seerapu, 2012*; *Lanahan et al., 2014*; *Chittenden et al., 2006*) and support the hypothesis that GIPCs limit PLXND1 antiangiogenic signaling. Finally, these observations directly argue against the possibility that GIPC-PLXND1 binding directly promotes PLXND1's antiangiogenic signaling (as in *Burk et al., 2017*) or that GIPCs are not involved in PLXND1 signaling.

## Reducing Plxnd1 signaling ameliorates the angiogenic deficits of maternal-zygotic (MZ) *gipc1* mutants

The hypothesis that the angiogenesis deficits of *gipc* mutants result, at least in part, from increased antiangiogenic PLXND1 signaling predicts that reducing the latter via heterozygosity for the null *plxnd1*^fov01b^ allele might have a restorative effect on the angiogenic growth of zebrafish *gipc* mutants.

To test this, we compared the Se and DLAV angiogenesis phenotypes of *gipc1*^skt1(MZ)^ and *gipc1*^skt1(MZ)^; *plxnd1*^fov01b^/+ mutant siblings (*Figure 5* and *Figure 5—figure supplement 1*; see also Materials and methods). Consistent with the proposed hypothesis, we found that *plxnd1*^fov01b^ heterozygosity led to a dramatic (*Figure 5A–B*) and statistically significant difference in distribution of the four phenotypic classes between both genotypes (*Figure 5C*), with a statistically significant reduction in both the penetrance and expressivity of Se and DLAV truncations in *gipc1*^skt1(MZ)^ mutants (*Figure 5—figure supplement 1*).

## Double maternal-zygotic (MZ) *gipc* mutants display excessive angiogenesis in the absence of Plxnd1 signaling

To further explore the relationship between GIPCs and PLXND1 signaling, we used a validated splice-blocking morpholino to determine the vascular consequences of removing *plxnd1* activity (*Torres-Vázquez et al., 2004*; *Morcos, 2007*; *Stainier et al., 2017*) from WT embryos and *gipc1*^skt1(MZ)^; *gipc2*^skt^(*Fischer et al., 2006*)^(MZ)^ double MZ mutants (*Figure 6*).

In contrast to the normal vasculature of WT embryos and the properly organized but hypoangiogenic vascular tree observed in *gipc1*^skt1(MZ)^; *gipc2*^skt(MZ)^ double MZ mutants (*Fischer et al., 2006*) (*Figure 6A,C*), we found that all the *plxnd1* morphants exhibited the hyperangiogenic phenotype of *plxnd1* nulls, namely excessive and disorganized Se angiogenesis (*Figure 6B,D*). The vascular

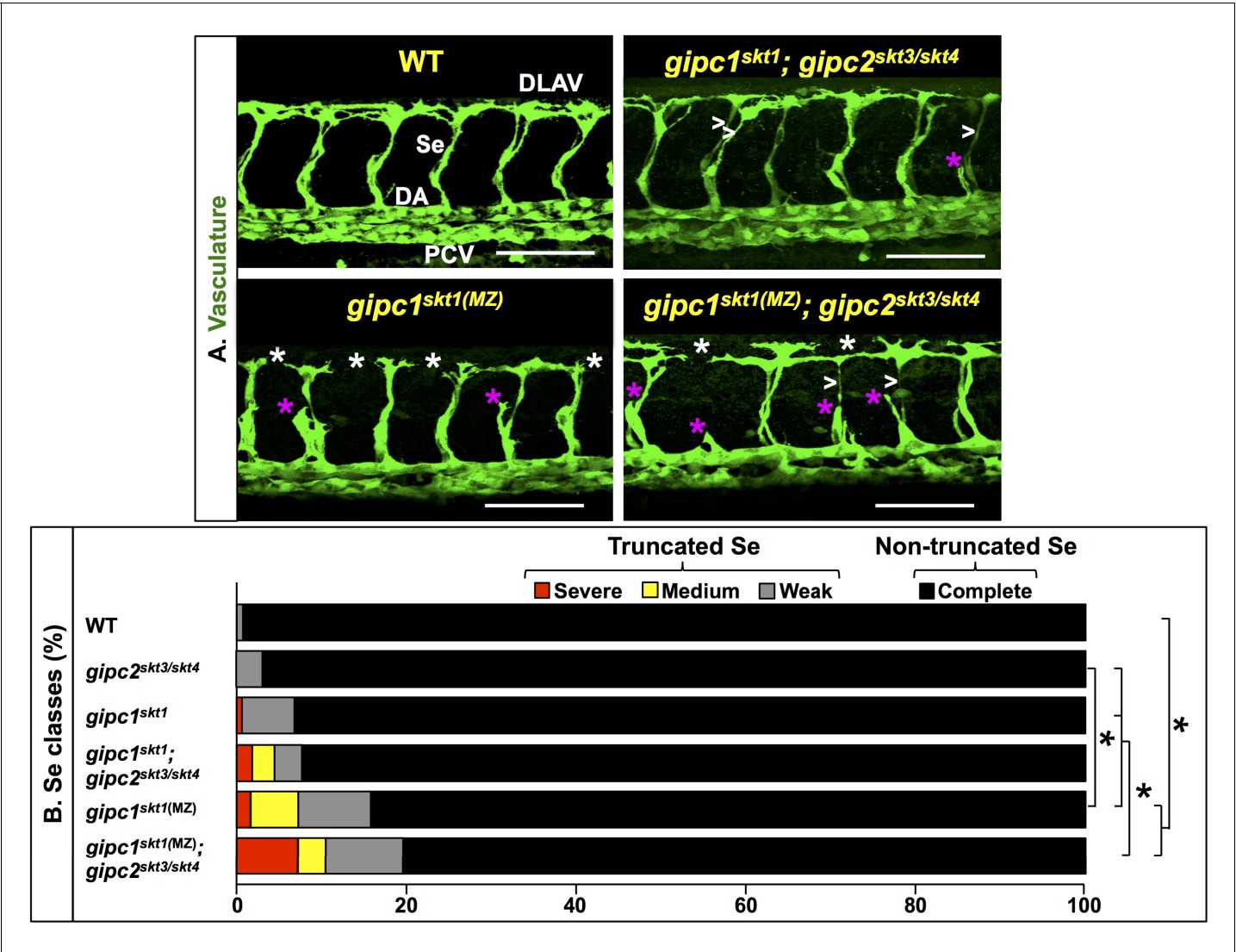

**Figure 4.** *gipc* mutants display angiogenesis deficits. (**A**) Confocal lateral images of the trunk vasculature (green) of 32 hpf embryos (region dorsal to the yolk extension). Anterior, left; dorsal, up. Scale bars (white horizontal lines), 100 µm. Genotypes indicated on top of each image in yellow font. Angiogenesis deficits are indicated as follows: white asterisks (DLAV gaps), magenta asterisks (truncated Se), white greater-than sign (thin Se). Maternal-zygotic (MZ) removal of *gipc* activity is denoted by the designation 'MZ' in superscript. In the WT image (top left), the vessels are designated with the white font as follows: DLAV (Dorsal Longitudinal Anastomotic Vessel), Se (Segmental Vessel), DA (Dorsal Aorta), and PCV (Posterior Cardinal Vein). (**B**) Bar graph. Percentage of Se in 32 hpf embryos of the indicated genotypes belonging to each of the following four phenotypic classes. Truncated: severe (includes missing Se), medium (yellow), and weak (gray). Non-truncated: complete (black). Significance values were calculated using a two-sided Fisher Exact test and significant differences ($p < 0.0033$) assigned using a Bonferroni-type adjustment for 15 pairwise genotype comparisons (0.05/15 = 0.0033). Brackets and asterisks indicate pairs of genotypes with significantly different distributions of these four phenotypic classes. Quantifications. We scored Se angiogenesis in embryos of the following six genotypes: WT (138 Se, 12 embryos; an average of 11.5 Se/embryo), *gipc1*$^{skt1}$ (130 Se, 11 embryos; an average of 11.8 Se/embryo), *gipc1*$^{skt1(MZ)}$ (380 Se, 33 embryos; an average of 11.5 Se/embryo), *gipc2*$^{skt3/skt4}$(130 Se, 11 embryos; an average of 11.8 Se/embryo), *gipc1*$^{skt1}$; *gipc2*$^{skt3/skt4}$ (152 Se, 13 embryos; an average of 11.6 Se/embryo), and *gipc1*$^{skt1(MZ)}$; *gipc2*$^{skt3/skt4}$ (220 Se, 19 embryos; an average of 11.5 Se/embryo). For additional data, graphs, and statistical comparisons related to this figure, see *Figure 4—figure supplement 2*, *Figure 4—figure supplement 3* and *Supplementary file 5*. Please note that given the use of different scales for scoring angiogenesis deficits, it is unfeasible to compare the quantifications in *Figure 4* and *Figure 5* directly.

DOI: https://doi.org/10.7554/eLife.30454.008

The following figure supplements are available for figure 4:

**Figure supplement 1.** GIPC proteins encoded by both the wild-type and mutant *gipc1*, *gipc2*, and *gipc3* alleles.

DOI: https://doi.org/10.7554/eLife.30454.009

**Figure supplement 2.** Angiogenesis deficits of *gipc1*$^{skt1}$ and *gipc2*$^{skt3/skt4}$ mutants at 32 hpf.

*Figure 4 continued on next page*

*Figure 4 continued*
DOI: https://doi.org/10.7554/eLife.30454.010
**Figure supplement 3.** Penetrance and expressivity of Se truncations in *gipc* mutants at 32 hpf.
DOI: https://doi.org/10.7554/eLife.30454.011

phenotypes of *plxnd1* morphants from these two backgrounds appeared qualitatively similar. This observation fits the notion that the angiogenesis deficits of *gipc* mutants are, at least partially, a result of increased PLXND1 signaling, consistent with the ameliorating effect of *plxnd1^fov01b^*

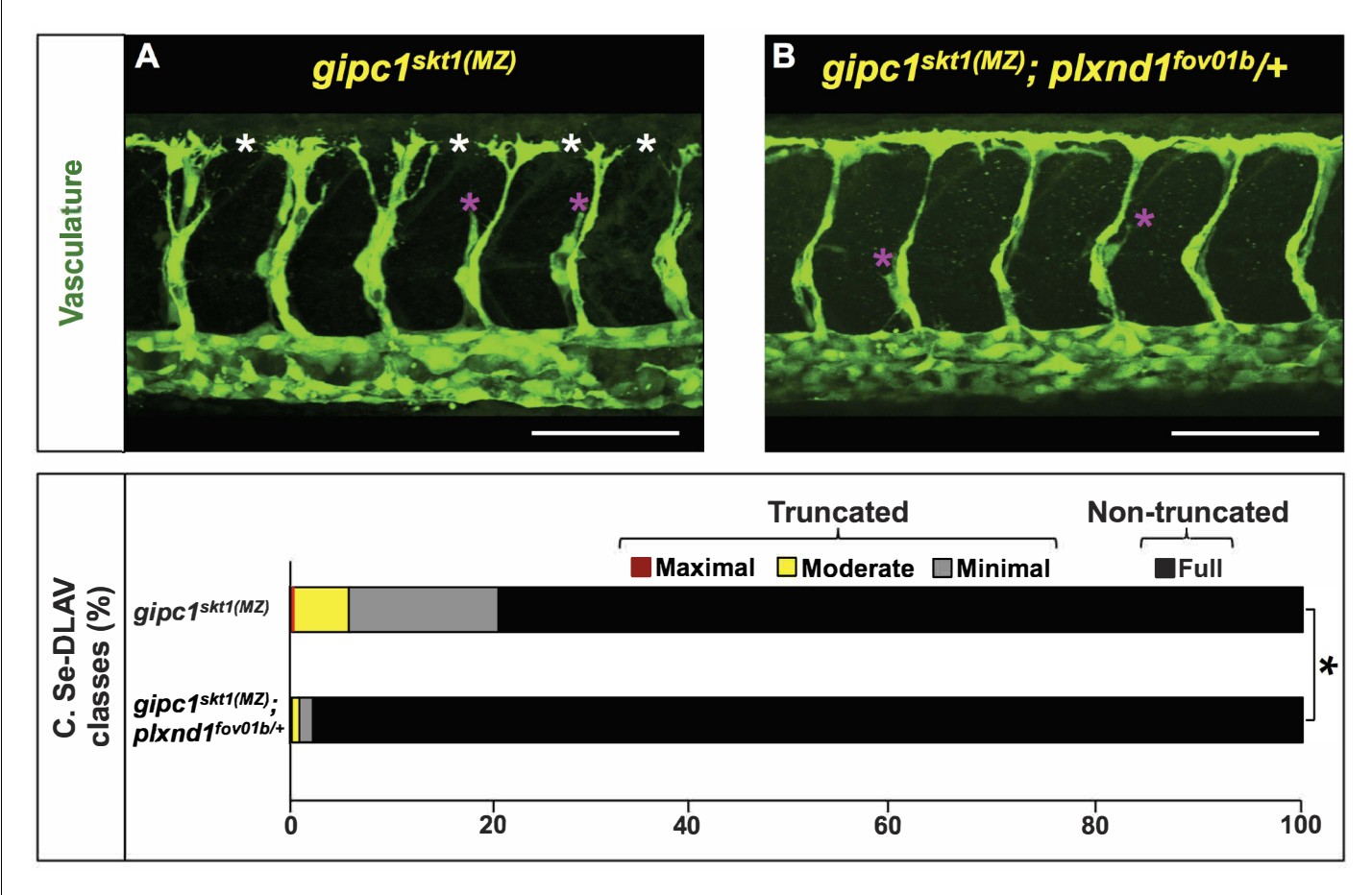

**Figure 5.** *plxnd1* heterozygosity suppresses the angiogenesis deficits of *gipc1^skt1(MZ)^* mutants. (A, B) Confocal lateral images of the trunk vasculature (green) of 32 hpf embryos (region dorsal to the yolk extension). Anterior, left; dorsal, up. Scale bars (white horizontal lines), 100 μm. Genotypes indicated on top of each image in yellow font. Angiogenesis deficits are indicated with asterisks as follows: white (DLAV gaps), magenta (truncated Se). (C) Bar graph. Percentage of Se-DLAV in 32 hpf embryos of the indicated genotypes belonging to each of the following four phenotypic classes. Truncated: maximal (red; includes missing Se), moderate (yellow), and minimal (gray). Non-truncated: full (black). There was a statistically significant difference (bracket with an asterisk) in distribution of the four phenotypic classes between *gipc1^skt1(MZ)^* and *gipc1^skt1(MZ)^; plxnd1^fov01b^/+* embryos, as assessed by a two-sided Fisher Exact test (p<0.05). Quantifications. We scored Se-DLAV angiogenesis in embryos of the following two genotypes: *gipc1^skt1(MZ)^* (390 Se-DLAV, 36 embryos, an average of 10.83 Se-DLAV/embryo) and *gipc1^skt1(MZ)^; plxnd1^fov01b^/+* (410 Se-DLAV, 38 embryos, an average of 10.79 Se-DLAV/embryo). For additional data, graphs, and statistical comparisons related to this figure, see *Figure 5—figure supplement 1* and *Supplementary file 6*. Please note that given the use of different scales for scoring angiogenesis deficits, it is unfeasible to compare the quantifications in *Figure 4* and *Figure 5* directly.
DOI: https://doi.org/10.7554/eLife.30454.012

The following figure supplement is available for figure 5:

**Figure supplement 1.** Penetrance and expressivity of Se-DLAV truncations in *gipc1^skt1(MZ)^* and *gipc1^skt1(MZ)^; plxnd1^fov01b^/+* embryos at 32 hpf.
DOI: https://doi.org/10.7554/eLife.30454.013

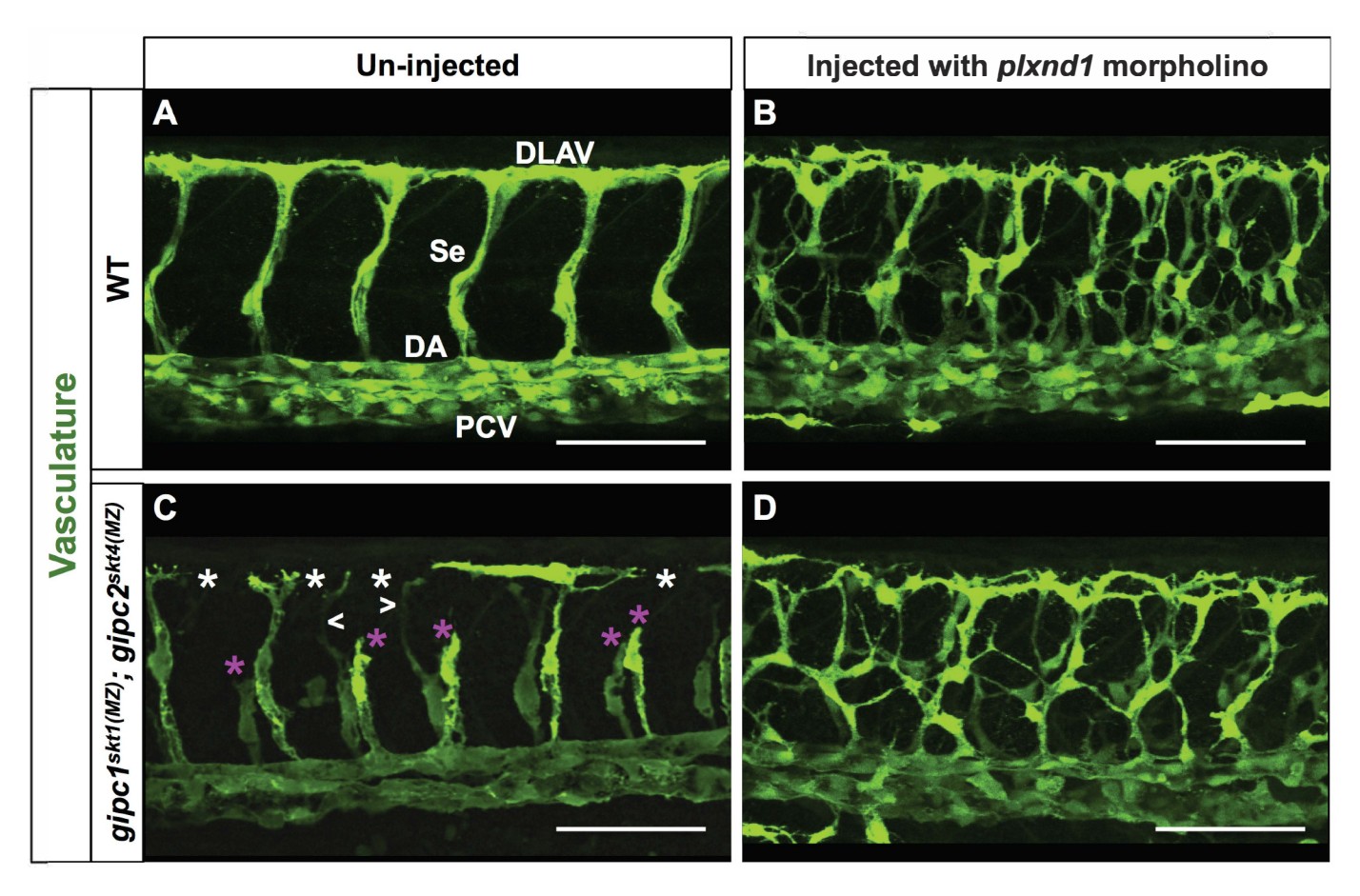

**Figure 6.** Removal of *plxnd1* activity from *gipc1*<sup>skt1(MZ)</sup>; *gipc2*<sup>skt4(MZ)</sup> maternal-zygotic (MZ) double mutants yields a phenotype similar to that of *plxnd1* nulls. (A–D) Confocal lateral images of the trunk vasculature (green) of 32 hpf embryos (region dorsal to the yolk extension). Anterior, left; dorsal, up. Scale bars (white horizontal lines), 100 μm. Morpholino injection (un-injected or injected with *plxnd1* morpholino) indicated on top, genotypes (WT or *gipc1*<sup>skt1(MZ)</sup>; *gipc2*<sup>skt4(MZ)</sup>) indicated on the left. The un-injected WT picture (A) shows the names of the major vessels in white font: DLAV (Dorsal Longitudinal Anastomotic Vessel), Se (Segmental Vessel), DA (Dorsal Aorta), and PCV (Posterior Cardinal Vein). Vascular defects highlighted as follows: truncated or missing Se (magenta asterisk), thin Se (white greater/less-than signs), DLAV gaps (white asterisk). Quantifications. The following number of embryos were analyzed: WT (four embryos), WT injected with *plxnd1* morpholino (four embryos; 4/4 showed a vascular phenotype similar to that of *plxnd1*<sup>fov01b</sup> nulls), *gipc1*<sup>skt1(MZ)</sup>; *gipc2*<sup>skt4(MZ)</sup> (12 embryos; 7/12 showed angiogenesis deficits), and *gipc1*<sup>skt1(MZ)</sup>; *gipc2*<sup>skt4(MZ)</sup> injected with *plxnd1* morpholino (11 embryos; 11/11 showed a vascular phenotype similar to that of *plxnd1*<sup>fov01b</sup> nulls).

DOI: https://doi.org/10.7554/eLife.30454.014

heterozygosity in the angiogenic deficits of MZ *gipc1* mutants (*Figure 5* and *Figure 5—figure supplement 1*).

### *GIPC* depletion potentiates SEMA3E-induced, PLXND1-dependent responses in HUVECs

To interrogate the role of GIPCs in endothelial PLXND1 signaling, we exploited the HUVEC model system in which PLXND1 signaling can be tuned by adjusting the exogenous supply of its canonical ligand, SEMA3E (*Sakurai et al., 2010*; *Tata et al., 2014*). This model offers well-defined molecular and cellular readouts of SEMA3E-induced PLXND1 signaling, such as a decreased level of phospho-active ERK1/2 (pERK1/2; Extracellular signal-related kinase 1 and 2) and cell collapse, caused by cytoskeletal disassembly and lowered adhesion to the substrate (*Moriya et al., 2010*; *Sakurai et al., 2010*; *Wang et al., 2012*; *Tata et al., 2014*; *Aghajanian et al., 2014*; *Moriya et al., 2010*; *Aghajanian et al., 2014*). Finally, these cells can be cultured without exogenous VEGF, which may have confounding effects (see *Lanahan et al., 2013*; *Lanahan et al., 2010*; *Chittenden et al., 2006*;

*Stahlhut et al., 2012*; *Lawson et al., 2002*; *Rossi et al., 2016*; *Jin et al., 2017*; *Lin et al., 2013*; *Carmeliet et al., 1996*; *Shalaby et al., 1997*; *Koch and Claesson-Welsh, 2012*; *Ruhrberg et al., 2002*; *Prahst et al., 2008*; *Soker et al., 1998*; *Yoshida et al., 2015*; *Lee et al., 2007*; *Domigan et al., 2015*).

If GIPC negatively modulates PLXND1 signaling, then *GIPC* depletion should potentiate SEMA3E-induced responses in a PLXND1-dependent manner. Alternatively, if GIPCs directly promote PLXND1 signaling, then *GIPC* depletion should decrease SEMA3E-induced responses in a PLXND1-dependent fashion. Finally, if GIPCs are indirect players in PLXND1 biology or GIPCs only have PLXND1-independent roles, then *GIPC* depletion should not affect SEMA3E-induced, PLXND1-dependent responses. To distinguish between these possibilities, we used an immortalized HUVEC line (HUVEC/TERT2), shRNA-mediated GIPC knockdowns, and genome editing to knockout (KO) *PLXND1* (see *Zygmunt et al., 2011*; *Wang et al., 2014*; *Shalem et al., 2014*; *Sanjana et al., 2014*; *Ulrich et al., 2016*); see *Figure 7*, *Figure 7—figure supplement 1* and *Figure 7—figure supplement 2*. This approach minimized noise from genetic variability, enabled long-term antibiotic selection for maximal *GIPC* knockdown, and enabled the isolation, sequencing, and protein-level validation of *PLXND1* KO clones (see *Supplementary file 1*, *Supplementary file 8* and *Figure 7—figure supplement 2*). We accounted for potential off-target effects from constitutive expression of gene-specific shRNAs and guide RNAs (gRNAs) using validated, non-targeting shRNAs (*Zygmunt et al., 2011*; *Wang et al., 2014*; *Ulrich et al., 2016*) as controls.

Using this setup, we measured the effects of SEMA3E treatment on relative pERK abundance over time in control cells and cells with *GIPC* loss, *PLXND1* loss, and *GIPC-PLXND1* double loss (*Figure 7*; see also *Figure 7—figure supplement 1* and *Figure 7—figure supplement 2*). As expected, SEMA3E stimulation reduced relative pERK abundance in control cells (*Figure 7A*, black bars in *Figure 7E*). Importantly, we found that this effect is PLXND1-dependent because *PLXND1* loss abrogated it (*Figure 7B*, red bars in *Figure 7E*). *GIPC* loss significantly potentiated the SEMA3E-induced decrease in relative pERK abundance at 45 min (*Figure 7C*, green bars in *Figure 7E*). Moreover, this potentiating effect is also PLXND1-dependent, because pERK levels failed to decrease under SEMA3E stimulation in *GIPC-PLXND1* double loss cells (*Figure 7D*, blue bars in *Figure 7E*). Importantly, cells with *GIPC-PLXND1* double loss did not show an intermediate relative level of pERK abundance between that of cells with *PLXND1* loss (*Figure 7B*, red bars in *Figure 7E*) and cells with *GIPC* loss (*Figure 7C*, green bars in *Figure 7E*). Instead, the relative pERK abundance of SEMA3E stimulated cells with *PLXND1* loss (*Figure 7B*, red bars in *Figure 7E*) and with *GIPC-PLXND1* double loss (*Figure 7D*, blue bars in *Figure 7E*) were not statistically significantly different. These quantitative findings align with the qualitatively similar vascular phenotypes of *plxnd1* morphants in the WT and *gipc1*<sup>skt1(MZ)</sup>; *gipc2*<sup>skt(MZ)</sup> backgrounds (*Fischer et al., 2006*) (*Figure 6B,D*).

The results of additional cell culture experiments with primary HUVEC agree with these findings (*Figure 7—figure supplement 3*; see also *Figure 7—figure supplement 4*). In these cells, addition of exogenous SEMA3E induces a PLXND1-dependent collapsing response (*Tata et al., 2014*). As expected, SEMA3E induced the collapse of HUVEC treated with the non-targeting, shRNA control (*Figure 7—figure supplement 3A,D*) and this morphological change did not occur in *PLXND1*-depleted cells (*Figure 7—figure supplement 3C,F*). *GIPC* depletion potentiated the SEMA3E-induced morphological response, resulting in cell hypercollapse (*Figure 7—figure supplement 3B, E*).

In summary, the results of HUVEC assays (*Figure 7* and *Figure 7—figure supplement 3*) support the hypothesis that *GIPC* depletion potentiates antiangiogenic PLXND1 signaling, consistent with the angiogenesis deficits and SU5416 hypersensitivity of *plxnd1*<sup>skt6</sup> homozygous mutants (*Figures 2–3*). Given the demonstrated capacity of GIPCs and PLXND1 to bind directly to each other independently of NRP1 (*Gay et al., 2011*; *Shang et al., 2017*) (*Figure 1*), the alignment of the *plxnd1*<sup>skt6</sup> and HUVEC data argues that GIPCs bind to PLXND1 to directly limit its antiangiogenic signaling. Furthermore, the role of GIPCs as negative regulators of antiangiogenic PLXND1 signaling is also consistent with the angiogenesis deficits of *gipc* mutants (*Figure 4*) and the ameliorating effect that the partial removal of Plxnd1 signaling exerts on the angiogenic deficits of MZ *gipc1* mutants (*Figure 5*).

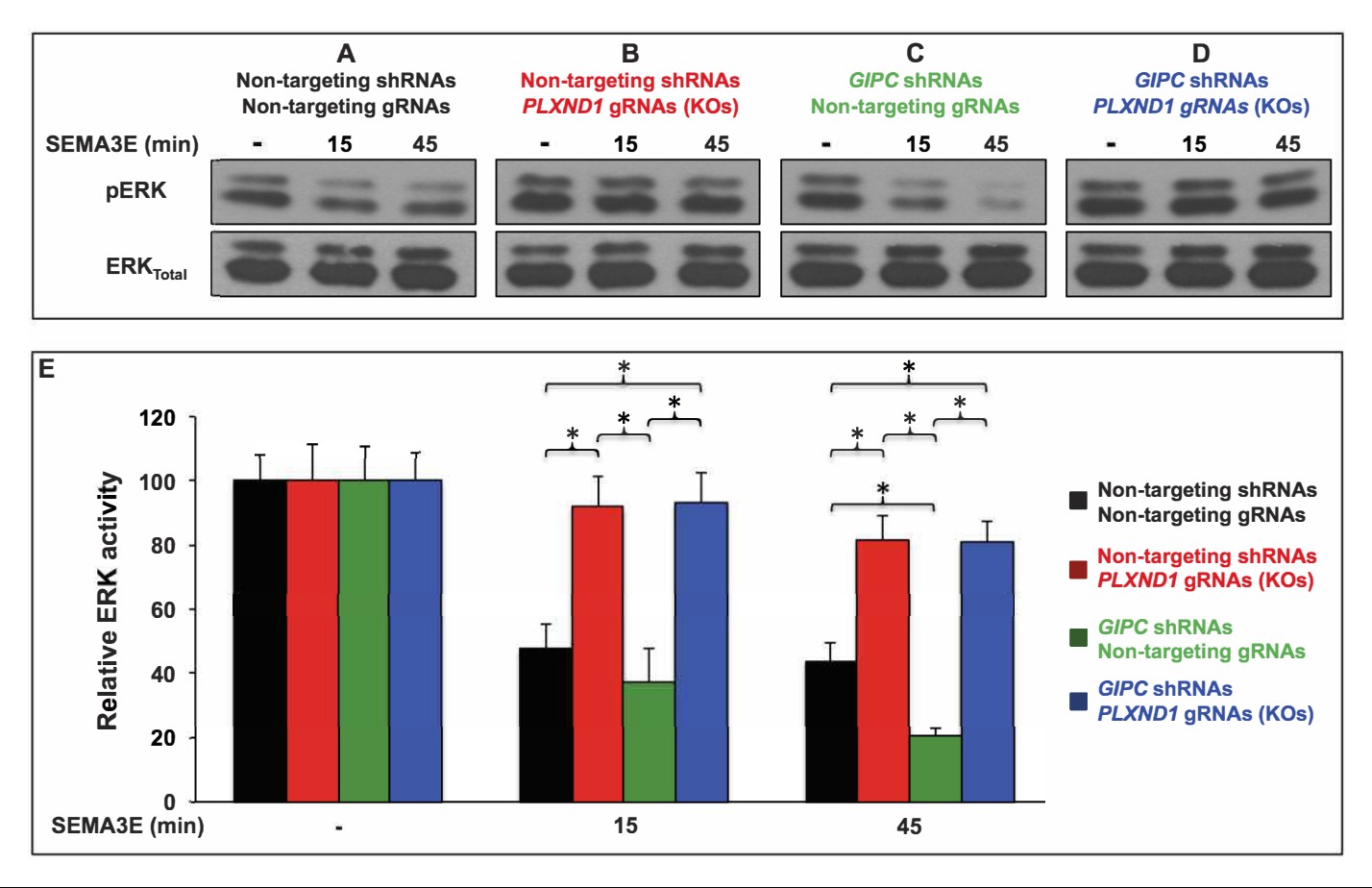

**Figure 7.** *GIPC* depletion potentiates SEMA3E-induced, PLXND1-dependent ERK inactivation in HUVEC/TERT2 cells. (A–D) Representative Western blot of active ERK1/2 (pERK) and total ERK1/2 (ERK$_{Total}$) from total cell lysates of HUVEC/TERT2 cells under the four conditions (shRNA and gRNA combinations) and the three ligand treatments indicated. Conditions. Control (A; bold black font), *PLXND1* loss (B; bold red font), *GIPC* loss (C; bold green font), and *GIPC-PLXND1* double loss (D; bold blue font). Treatments. Vehicle (-) and 2 nM SEMA3E for the indicated times. Cells stably carried a vector coexpressing the Cas9 nuclease and the indicated gRNAs. The alleles of the *PLXND1* gRNAs (KOs) are stable and defined (see **Supplementary file 1** and **Supplementary file 8**). (E) Bar graph. Means of percentual relative ERK activity (pERK/ERK$_{Total}$) under the described conditions (color coded as above) and treatments. Error bars, ± SEM. Relative ERK activity. Statistically significant differences between pairwise combinations of conditions and treatments are indicated (brackets and asterisks). Quantifications. $n$ = 4 independent experiments per *PLXND1* gRNA KO (for a pooled total of 8 experiments); $n$ = 4 independent experiments per non-targeting gRNA (for a pooled total of 8 experiments). One-way ANOVA tests were conducted to determine whether relative ERK activity was significantly different between cells in the control, *PLXND1* loss, *GIPC* loss, and *GIPC-PLXND1* double loss conditions across each treatment. There were no outliers in the data, as assessed by inspection of a boxplot. Relative ERK activity data were normally distributed, for each treatment, as determined by Shapiro-Wilk's test (p>0.05) except for the SEMA3E 15 min treatment; p=0.031. There was homogeneity of variances, as assessed by Levene's test (p>0.05) for equality of variances in all conditions. One-way ANOVA tests summary. Relative ERK activity was not statistically significantly different between conditions under vehicle treatment ($F$(3, 28)=0.004, p=1). Relative ERK activity was statistically significantly different between conditions under SEMA3E 15 min treatment ($F$(3, 28)=10.291, p<0.0005), effect size was $\omega^2$ = 0.46. Relative ERK activity was statistically significantly different between conditions under SEMA3E 45 min treatment ($F$(3, 28)=28.738, p<0.0005), effect size was $\omega^2$ = 0.72. Summary of the four statistically significant differences revealed by Tukey post hoc analysis (between conditions under SEMA3E 15 min treatment). Control versus *PLXND1* loss was statistically significantly different (p<0.05): (95% CI (17.2450 to 67.2680); p=0.001). Control versus *GIPC- PLXND1* double loss was statistically significantly different (p<0.05): (95% CI (−80.9622 to −12.0628); p=0.005). *GIPC* loss versus *PLXND1* loss was statistically significantly different (p<0.05): (95% CI (17.1753 to 86.0747); p=0.002). *GIPC* loss versus *GIPC-PLXND1* double loss was statistically significantly different (p<0.05): (95% CI (−90.0872 to −21.1878); p=0.001). Summary of the five statistically significant differences revealed by Tukey post hoc analysis (between conditions under SEMA3E 45 min treatment). Control versus *PLXND1* loss was statistically significantly different (p<0.05): (95% CI (−59.2432 to −16.5068); p<0.0005). Control versus *GIPC-PLXND1* double loss was statistically significantly different (p<0.05): (95% CI (−57.3307 to −14.5943); p<0.0005). *GIPC* loss versus *PLXND1* loss was statistically significantly different (p<0.05): (95% CI 39.9193 to 82.6557); p<0.0005). *GIPC* loss versus *GIPC-PLXND1* loss was statistically significantly different (p<0.05): (95% CI-80.7432 to −38.0068); p<0.0005). Control versus *GIPC* loss was statistically significantly different (p<0.05): (95% CI (2.0443 to 44.7807); p=0.028). For additional data, graphs, and statistical comparisons related to this figure, see **Supplementary file 7**.

*Figure 7 continued on next page*

*Figure 7 continued*

DOI: https://doi.org/10.7554/eLife.30454.015

The following figure supplements are available for figure 7:

**Figure supplement 1.** *GIPC* depletion potentiates SEMA3E-induced, PLXND1-dependent ERK inactivation in HUVEC/TERT2 cells.
DOI: https://doi.org/10.7554/eLife.30454.016

**Figure supplement 2.** Efficient shRNA-mediated knockdown of GIPCs and CRISPR/Cas9-mediated knockout of PLXND1 in HUVEC/TERT2 cells.
DOI: https://doi.org/10.7554/eLife.30454.017

**Figure supplement 3.** *GIPC* knockdowns potentiate the SEMA3E-induced/PLXND1-dependent cell collapse of primary HUVEC.
DOI: https://doi.org/10.7554/eLife.30454.018

**Figure supplement 4.** Efficient shRNA-mediated GIPC (GIPC1, GIPC2, and GIPC3) and PLXND1 knockdowns in primary HUVEC used for cell collapse experiments.
DOI: https://doi.org/10.7554/eLife.30454.019

## Discussion

Our past crystallographic observations (*Shang et al., 2017*), together with the results of our CoIP, zebrafish, and cell culture experiments, and the consensus regarding the vascular and molecular roles of GIPC1 and PLXND1 in zebrafish (*Torres-Vázquez et al., 2004*; *Zygmunt et al., 2011*; *Childs et al., 2002*; *Ren et al., 2010*; *Hermans et al., 2010*; *Chittenden et al., 2006*; *Yokota et al., 2015*; *Alvarez et al., 2007*; *Goi and Childs, 2016*; *Minchin et al., 2015*) and mammals (*Gitler et al., 2004*; *Zhang et al., 2009*; *Gu et al., 2005*; *Moriya et al., 2010*; *Fukushima et al., 2011*; *Kim et al., 2011*; *Worzfeld et al., 2014*; *Aghajanian et al., 2014*; *Ren et al., 2010*; *Chittenden et al., 2006*; *Lanahan et al., 2010*; *Gao et al., 2000*; *Moraes et al., 2013*; *Paye et al., 2009*; *Kanda et al., 2007*; *Degenhardt et al., 2010*; *Liu et al., 2016*; *Hegan et al., 2015*) lead us to conclude that GIPCs association with PLXND1 negatively modulates the receptor's antiangiogenic activity. This conclusion also fits with data from additional studies of the mammalian PLXND1 and PLXNB1 receptors. Briefly, BAC-based transgenic expression of full-length PLXND1 tagged at its C-terminus (a modification predicted to disable PBM-dependent binding events (*Rickhag et al., 2013*; *Saras et al., 1997*; *Cao et al., 1999*; *Garbett and Bretscher, 2012*) fully rescues the lethality, cardiac, vascular, and skeletal defects of murine *Plxnd1* knockouts. In contrast, mutations that eliminate PLXND1's GAP function inactivate the receptor (*Worzfeld et al., 2014*). Similarly, PLXNB1 harbors a C-terminal PBM with specificity for guanine nucleotide exchange factors, and removal of PLXNB1's PBM does not inactivate the receptor (*Oinuma et al., 2004*).

Beyond its involvement in developmental angiogenesis, the GIPC-based modulation of endothelial PLXND1 signaling might promote the stabilization, repair, homeostasis, and arteriogenic remodeling of vessels, particularly in contexts with minimal proangiogenic stimulation. These conditions are found in the quiescent vascular beds of adults and are a hallmark of several diseases (see *Carmeliet, 2003*; *Carmeliet, 2005*).

We highlight that our results and conclusion challenge the model, recently proposed by Burk et al., that GIPC1 enables PLXND1 activity (*Burk et al., 2017*). These authors primarily address how *Gipc1* and *Plxnd1* modulate axonal circuit development using mouse embryos and neuronal explants. They propose that GIPC1's PDZ domain and PLXND1's canonical PBM drive GIPC1-PLXND1 heteromerization and, based on ex vivo forced expression experiments, conclude that the PBM is essential for PLXND1 activity. Finally, using a *Plxnd1* null allele and floxed *Gipc1* allele that might yield a truncated protein retaining PLXND1 binding (see *Shang et al., 2017*; *Moraes et al., 2013*), they peripherally explore the vascular role of the GIPC1-PLXND1 interaction using a qualitative phenotypic analysis that omits *Plxnd1* mutant homozygotes.

Yet prior reports describe the viability, cardiovascular defects, and other abnormalities of *Plxnd1* knockouts (*Gitler et al., 2004*; *Zhang et al., 2009*; *Gu et al., 2005*; *Fukushima et al., 2011*; *Kim et al., 2011*; *Worzfeld et al., 2014*; *Aghajanian et al., 2014*; *Kanda et al., 2007*; *Degenhardt et al., 2010*; *Liu et al., 2016*) and *Gipc1* mutants (*Ren et al., 2010*; *Chittenden et al., 2006*; *Dedkov et al., 2007*; *Lanahan et al., 2010*; *Moraes et al., 2013*; *Hegan et al., 2015*) as contrastingly different, and the molecular roles for PLXND1 (*Moriya et al., 2010*; *Aghajanian et al., 2014*) and GIPC1 (*Ren et al., 2010*; *Paye et al., 2009*; *Chittenden et al., 2006*; *Herzog et al., 2011*; *Moraes et al., 2014*) as distinct. Given that the molecular regulation of vascular development

is highly conserved (*Hogan and Schulte-Merker, 2017*), we favor the view that GIPCs limit, in both zebrafish and mammals, the antiangiogenic signaling of the PLXND1 receptor.

In all likelihood, the proangiogenic function of GIPCs is multifaceted and involves additional mechanisms besides the negative modulation of PLXND1 signaling. First, GIPCs physically interact with many other PDZ-binding transmembrane proteins and with the retrograde motor MYO6. Some of these GIPC-binding proteins, for example, NRP1, are expressed in endothelial cells and have vascular functions (*Shang et al., 2017*; *Katoh, 2013*; *Gao et al., 2000*; *Naccache et al., 2006*; *Salikhova et al., 2008*; *Lou et al., 2001*; *Cai and Reed, 1999*). Second, the angiogenesis deficits of *gipc1; gipc2* double mutants are more penetrant and severe than those of *plxnd1*$^{skt6}$ homozygotes, in agreement with the notion that the vascular defects of *gipc* mutants are partially a result of enhanced Plxnd1 signaling.

We highlight that the identification of GIPCs as the first negative intracellular regulators of PLXND1 signaling expands the vascular roles of GIPCs beyond the prevailing notion that GIPC1 promotes arterial branching by facilitating, in an NRP1-dependent manner, proangiogenic VEGF signaling (*Chittenden et al., 2006*; *Lanahan et al., 2010*; *Fantin et al., 2011*; *Lampropoulou and Ruhrberg, 2014*; *Herzog et al., 2011*; *Plein et al., 2014*). Importantly, our findings do not argue against the possibility that GIPCs might also, either directly or indirectly, modulate PLXND1 signaling in connection with NRP1. For example, proangiogenic VEGF signaling via GIPC-NRP1-VEGFR2 complexes might trigger posttranslational modifications in GIPC and PLXND1 that counteract the antiangiogenic activity of the latter by promoting GIPC-PLXND1 interactions or the degradation of the PLXND1 receptor. Alternatively, GIPCs might mediate the formation of novel complexes containing receptors from both pathways (for instance, PLXND1-GIPC-NRP1 and PLXND1-GIPC-NRP1-VEGFR2) that perform still uncharacterized functions (see *Chauvet et al., 2007*; *Bellon et al., 2010*).

Our findings suggest the potential involvement of endocytosis and recycling, intracellular processes that regulate other pathways (*Lanahan et al., 2010*; *Naccache et al., 2006*; *Reed et al., 2005*; *Salikhova et al., 2008*; *Hu et al., 2003*; *Jeanneteau et al., 2004a*; *Barbieri et al., 2016*; *Villaseñor et al., 2016*; *Koch et al., 2014*), in modulation of PLXND1 signaling (see also *Burk et al., 2017*; *Salikhova et al., 2008*; *Steinberg et al., 2013*). We favor the model that GIPCs and MYO6 work together to facilitate PLXND1's endocytic trafficking based on the following observations. GIPCs are MYO6-dependent internalization adaptors (*Katoh, 2013*; *Wollscheid et al., 2016*). Crystallographic findings indicate that the binding of PLXND1 conformationally releases GIPC1's domain-swapped autoinhibited dimer, promoting its interaction with MYO6 (*Shang et al., 2017*). Also, PLXND1, GIPC1, and MYO6 colocalize within intracellular puncta in SEMA3E-stimulated cells (*Shang et al., 2017*; see also *Burk et al., 2017*). Finally, the arterial tree deficits of *Gipc1* and *Myo6* mutants are similar (see *Chittenden et al., 2006*; *Lanahan et al., 2010*).

GIPCs (and MYO6) might limit PLXND1 signaling by restricting the amount of time that the activated receptor spends at the cell surface. This mechanism could serve a dual role of regulating access of the receptor to its catalytic targets such as the Rap1 GTPase, found at the inner cell membrane (*Wang et al., 2012*; *Worzfeld et al., 2014*; *Wang et al., 2013*), as well as shaping the inactivation kinetics of ligand-bound PLXND1. We note that recent structural studies indicate that mammalian MYO6 also functions as a ubiquitin receptor, thereby suggesting a potential link between GIPCs and the proteasome-mediated degradation of their cargo (*He et al., 2016*). Alternatively, GIPCs might function in a non-endocytic manner to limit PLXND1 signaling independently of MYO6. For example, GIPCs might recruit a third protein, which in turn, directly antagonizes PLXND1 signaling (see *Lampropoulou and Ruhrberg, 2014*; *De Vries et al., 1998a*; *Lou et al., 2001*; *Wieman et al., 2009*; *Cai and Reed, 1999*; *Kofler and Simons, 2016*; *Guo and Vander Kooi, 2015*; *De Vries et al., 1998b*; *Fischer et al., 1999*; *Fischer et al., 2003*; *Jeanneteau et al., 2004b*; *Jean-Alphonse et al., 2014*; *Lin et al., 2006*). Finally, the interaction between PLXND1 and GIPCs might directly influence cell morphology via modulation of cytoskeletal dynamics. Support for this notion comes from genetic experiments in *Drosophila* and mammalian proteomic studies that implicate GIPCs and MYO6 in actin network stabilization (*Djiane and Mlodzik, 2010*; *O'Loughlin et al., 2018*; *Isaji et al., 2011*; *Noguchi et al., 2006*).

The mechanisms that regulate GIPC-PLXND1 interaction are unclear. SEMA3E stimulates colocalization of GIPC1 and PLXND1 (*Burk et al., 2017*; *Shang et al., 2017*), but the kinetics of their association are not yet defined. Another open question is whether non-canonical PLXND1 ligands (SEMA3A, SEMA3C, SEMA3D, SEMA3G, and SEMA4A) (*Epstein et al., 2015*; *Liu et al., 2016*;

*Hamm et al., 2016*) also promote GIPC-PLXND1 complex formation. How SEMA-induced changes in the conformation and oligomerization of PLXND1 (see *Pascoe et al., 2015*) impact the receptor's ability to interact with GIPCs, or the specificity of their interaction is also unexplored. Finally, reversible posttranslational modifications might modulate the GIPC-PLXND1 interaction. Because phosphorylation of PDZ domains modulates the recruitment of their partners (*Lee and Zheng, 2010*; *Liu et al., 2013*), this is a plausible mechanism for GIPC-based control of GIPC-PLXND1 interactions. At the PLXND1 level, the phosphorylation and S-palmitoylation of the receptor's cytosolic tail could also play a modulatory role. Plxns are phosphorylated (*Cagnoni and Tamagnone, 2014*; *Franco and Tamagnone, 2008*), and the GBM of PLXND1 harbors a conserved tyrosine located within a consensus Src family kinase phosphorylation site. This residue plugs into a hydrophobic GIPC pocket between the GH1 and PDZ domains (*Gay et al., 2011*; *Shang et al., 2017*). On the other hand, some type I transmembrane proteins, including Plxns, are S-palmitoylated (*Holland and Thomas, 2017*; *Blaskovic et al., 2013*), and S-palmitoylation of the carboxy tail of the GIPC1-binding dopamine Drd3 receptor buries the PBM within the cell membrane to prevent the Drd3-GIPC1 interaction (*Arango-Lievano et al., 2016*).

In conclusion, we have identified a novel role for the GIPCs as pioneer negative regulators of SEMA-PLXND1 signaling. Given the prominent role of this pathway in shaping organogenesis of cardiovascular, nervous, and other systems and its central importance in cancer biology (*Moriya et al., 2010*; *Gay et al., 2011*; *Valdembri et al., 2016*; *Oh and Gu, 2013*; *Gaur et al., 2009*; *Gu and Giraudo, 2013*; *Neufeld et al., 2016*; *Bielenberg and Klagsbrun, 2007*), our findings suggest a broad human health relevance for therapeutic targeting of the SEMA-PLXND1 pathway at the GIPC level.

# Materials and methods

**Key resources table**

| Reagent type (species) or resource | Designation | Source or reference | Identifiers | Additional information |
|---|---|---|---|---|
| Genetic reagent (*Danio rerio*) | *Tg(kdrl:HsHRAS-mCherry)*[s896] | DOI: 10.1101/gad.1629408 | ZFIN ID: ZDB-ALT-081212–4 | Transgenic insertion |
| Genetic reagent (*Danio rerio*) | *Tg(fli1a:EGFP)*[y1] | PMID:12167406 | ZFIN ID: ZDB-ALT-011017–8 | Transgenic insertion |
| Genetic reagent (*Danio rerio*) | *Tg(fli1a:GAL4FF)*[ubs4] | DOI: 10.1016/j.devcel.2011.06.033 | ZFIN ID: ZDB-ALT-110921–1 | Transgenic insertion |
| Genetic reagent (*Danio rerio*) | *Tg(flt1:nlsmCherry)*[skt7] | This paper | | Transgenic insertion. Made using Torres-Vázquez lab plasmid #1208 |
| Genetic reagent (*Danio rerio*) | *gipc1*[skt1] | This paper | | Putative null mutant allele |
| Genetic reagent (*Danio rerio*) | *gipc1*[skt2] | This paper | | Putative null mutant allele |
| Genetic reagent (*Danio rerio*) | *gipc2*[skt3] | This paper | | Putative null mutant allele |
| Genetic reagent (*Danio rerio*) | *gipc2*[skt4] | This paper | | Putative null mutant allele |
| Genetic reagent (*Danio rerio*) | *gipc3*[skt5] | This paper | | Putative null mutant allele |
| Genetic reagent (*Danio rerio*) | *plxnd1*[fov01b] | PMID: 11861480 DOI: 10.1016/j.devcel.2004.06.008 | ZFIN ID: ZDB-ALT-010621–6 | Null mutant allele (point mutation) |
| Genetic reagent (*Danio rerio*) | *plxnd1*[skt6] | This paper | | Hypermorphic mutant allele |
| Cell line (*Cercopithecus aethiops*) | COS-7 (Monkey Kidney Fibroblasts) | American Type Culture Collection | Cat. #CRL-1651. RRID:CVCL_0224 | https://www.atcc.org/products/All/CRL-1651.aspx |

*Continued on next page*

*Continued*

| Reagent type (species) or resource | Designation | Source or reference | Identifiers | Additional information |
|---|---|---|---|---|
| Cell line (*Homo sapiens*) | HUVEC/TERT2 (Immortalized Human Umbilical Vein Endothelial Cells) | American Type Culture Collection | Cat. #CRL-4053. RRID:CVCL_9Q53 | https://www.atcc.org/Products/All/CRL-4053.aspx |
| Cell line (*Homo sapiens*) | HUVEC (Normal Primary Human Umbilical Vein Endothelial Cells) | Lifeline Cell Technology | Cat. #FC-0003 | https://www.lifelinecelltech.com/shop/cells/human-endothelial-cells/umbilical-vein-endothelial-cells/huvec-fc-0003/ |
| Cell line (*Homo sapiens*) | Non-targeting gRNA1. Pool of HUVEC/TERT2 cells. | This paper | | Derived from HUVEC/TERT 2 cell line (ATCC CRL4053). Cells were grown under blasticidin (4 µg/ml) selection and used between 7th and 10th passages. Cells stably coexpress Cas9 nuclease and non-targeting gRNA1 (from Torres-Vázquez lab plasmid #1859) |
| Cell line (*Homo sapiens*) | Non-targeting gRNA2. Pool of HUVEC/TERT2 cells. | This paper | | Derived from HUVEC/TERT 2 cell line (ATCC CRL4053). Cells were grown under blasticidin (4 µg/ml) selection and used between 7th-10th passages. Cells are stably coexpressing Cas9 nuclease and non-targeting gRNA2 (from Torres-Vázquez lab plasmid #1860) |
| Cell line (*Homo sapiens*) | *PLXND1* gRNA KO1. Monoclonal *PLXND1* KO HUVEC/TERT2 cell line. | This paper | | Biallelic (transheterozygous) *PLXND1* knockout line. Derived from HUVEC/TERT 2 cell line (ATCC CRL4053). Cells were grown under blasticidin (4 µg/ml) selection and used between 7th-10th passages. Cells are stably coexpressing Cas9 nuclease and *PLXND1* gRNA KO1 (from Torres-Vázquez lab plasmid #1846) |
| Cell line (*Homo sapiens*) | *PLXND1* gRNA KO2. Monoclonal *PLXND1* KO HUVEC/TERT2 cell line. | This paper | | Biallelic (transheterozygous) *PLXND1* knockout line. Derived from HUVEC/TERT 2 cell line (ATCC CRL4053). Cells were grown under blasticidin (4 µg/ml) selection and used between 7th-10th passages. Cells are stably coexpressing Cas9 nuclease and *PLXND1* gRNA KO2 (from Torres-Vázquez lab plasmid # 1847) |
| Cell line (*Homo sapiens*) | HEK293T (embryonic kidney cells) | Matthias Stadtfeld lab, NYU | | |
| Recombinant DNA reagent | V5-C-mPLXND1^WT | This paper | | Torres-Vázquez lab plasmid #862. Vector backbone: *pcDNA3.1/nV5-DEST-V5* |
| Recombinant DNA reagent | V5-C-mPLXND1Δ^CYSEA | This paper | | Torres-Vázquez lab plasmid #863. Vector backbone: *pcDNA3.1/nV5-DEST-V5* |
| Recombinant DNA reagent | V5-C-mPLXND1Δ^GBM | This paper | | Torres-Vázquez lab plasmid #1774. Vector backbone: *pcDNA3.1/nV5-DEST-V5* |

*Continued on next page*

*Continued*

| Reagent type (species) or resource | Designation | Source or reference | Identifiers | Additional information |
|---|---|---|---|---|
| Recombinant DNA reagent | FLAG-mGIPC1$^{WT}$ | DOI: 10.1091/mbc.12.3.615 | | Torres-Vázquez lab plasmid #864. Vector backbone: *pFLAG-CMV1* |
| Recombinant DNA reagent | FLAG-mGIPC1$^{GH1}$ | DOI: 10.1091/mbc.12.3.615 | | Torres-Vázquez lab plasmid #868. Vector backbone: *pFLAG-CMV2* |
| Recombinant DNA reagent | FLAG-mGIPC1$^{PDZ}$ | DOI: 10.1091/mbc.12.3.615 | | Torres-Vázquez lab plasmid #866. Vector backbone: *pFLAG-CMV3* |
| Recombinant DNA reagent | 2xHA-Plxnd1$^{WT}$ | This paper | | GAL4-responsive, Gateway and *IRES*-based bicistronic vector for *Tol2*-mediated zebrafish transgenesis. Torres-Vázquez lab plasmid #1414 |
| Recombinant DNA reagent | 2xHA-Plxnd1Δ$^{GBM}$ | This paper | | GAL4-responsive, Gateway and *IRES*-based bicistronic vector for *Tol2*-mediated zebrafish transgenesis. Torres-Vázquez lab plasmid #1685 |
| Recombinant DNA reagent | lentiCRISPR v2-Blast | Addgene | Cat. #83480 | A gift from Mohan Babu. https://www.addgene.org/83480/ |
| Recombinant DNA reagent | Non-targeting gRNA1 | This paper | | Torres-Vázquez lab plasmid #1859. Vector backbone: lentiCRISPR v2-Blast |
| Recombinant DNA reagent | Non-targeting gRNA2 | This paper | | Torres-Vázquez lab plasmid #1860. Vector backbone: lentiCRISPR v2-Blast |
| Recombinant DNA reagent | PLXND1-KO1 | This paper | | Torres-Vázquez lab plasmid #1846. Vector backbone: lentiCRISPR v2-Blast |
| Recombinant DNA reagent | PLXND1-KO2 | This paper | | Torres-Vázquez lab plasmid #1847. Vector backbone: lentiCRISPR v2-Blast |
| Recombinant DNA reagent | Control shRNA Lentiviral Particles-A (Non-targeting control shRNA) | Santa Cruz Biotechnology | Cat. #sc-108080 | Encodes a non-targeting shRNA sequence, will not lead to the specific degradation of any known cellular mRNA |
| Recombinant DNA reagent | GIPC shRNA (h) Lentiviral Particles | Santa Cruz Biotechnology | Cat. #sc-35475-V | shRNA pool (three target-specific constructs against human *GIPC1* that encode 19–25 nt (plus hairpin) shRNAs). Target sequences (sense sequences 5' to 3'): (1) CUGACGAGUUCGUCUUUGA (2) CCACCACUUUCCACCAUCA (3) CUGAAUUUGCUGUCUUGAA |
| Recombinant DNA reagent | GIPC2 shRNA (h) Lentiviral Particles | Santa Cruz Biotechnology | Cat. # sc-75132-V | shRNA pool (three target-specific constructs against human *GIPC2* that encode 19–25 nt (plus hairpin) shRNAs). Target sequences (sense sequences 5' to 3'): (1) CAGACGAAUUUGUCUUUGA (2) GGACACCUUUACUAACUCU (3) CCAACUUUCUCUCUUUGUA |

*Continued on next page*

*Continued*

| Reagent type (species) or resource | Designation | Source or reference | Identifiers | Additional information |
|---|---|---|---|---|
| Recombinant DNA reagent | GIPC3 shRNA (h) Lentiviral Particles | Santa Cruz Biotechnology | Cat. #sc-62376-V | shRNA pool (three target-specific constructs against human *GIPC3* that encode 19–25 nt (plus hairpin) shRNAs). Target sequences (sense sequences 5' to 3'): (1) CCUUCAUCAAGAGAAUCAA (2) GGAGUUUGCACGCUGUUUA (3) GACAAGUUCCUCUCUAGAA |
| Recombinant DNA reagent | Plexin-D1 shRNA (h) Lentiviral Particles | Santa Cruz Biotechnology | Cat. #sc-45585-V | shRNA pool (three target-specific constructs against human *PLXND1* that encode 19–25 nt (plus hairpin) shRNAs). Target sequences (sense sequences 5' to 3'): (1) GUCAAGAUAGGCCAAGUAA (2) CCAUGAGUCUCAUAGACAA (3) CCACAGACAGUUUCAAGUA |
| Sequenced-based reagent | *plxnd1*$^{3207-3462}$ morpholino | DOI: 10.1016/j.devcel.2004.06.008 | | Validated splice-blocking morpholino against zebrafish *plxnd1*. Synthesized by GENE TOOLS, LLC). Sequence (5' to 3'): CACACACACTCACGTTGATGATGAG |
| Antibody | Chicken anti-GFP | Invitrogen | Cat. #A10262 | IF (1:1,000); zebrafish |
| Antibody | Sheep anti-mCherry | Holger Knaut lab, NYU | | IF (1:1,000); zebrafish. Custom made antibody |
| Antibody | Mouse anti-pFAK Tyr397 | Millipore | Cat. #05–1140 | IF (1:1,000); zebrafish |
| Antibody | Rabbit anti-GIPC1 | Proteintech Group | Cat. #14822–1-AP. RRID:AB_2263269 | WB (1:3,000). This antibody detects GIPC1, GIPC2, and GIPC3 (our data) |
| Antibody | Rabbit anti-GIPC2 | Abcam | Cat. #ab175272 | WB (1:5,000). This antibody detects GIPC1 and GIPC2 (our data) |
| Antibody | Rabbit anti-GIPC3 | Abcam | Cat. #ab186426 | WB (1:5000). This antibody is specific for GIPC3 (our data). Validated against HeLa TCL (positive control; a gift from Mamta Tahiliani's lab, NYU) |
| Antibody | Mouse anti-PLXND1 | R and D Systems | Cat. #MAB41601 Clone #752815 | WB (1:250). Lyophilized reagent reconstituted in 200 μl of sterile PBS (GIBCO, Cat. #10010–023) |
| Antibody | Rabbit anti-Phospho-p44/42 MAPK (Erk1/2) (Thr202/Tyr204) (D13.14.4E) XPTM | Cell Signaling Technology | Cat. #4370S. RRID:AB_2315112 | WB (1:20,000) |
| Antibody | Mouse anti-p44/42 MAPK (Erk1/2) (L34F12) | Cell Signaling Technology | Cat. #4696S | WB (1:10,000) |
| Antibody | Rabbit anti-GAPDH (D16H11) | Cell Signaling Technology | Cat. #5174P. RRID:AB_10622025 | WB (1:20,000) |
| Antibody | Mouse anti-FLAG M2 | SIGMA-ALDRICH | Cat. #F3165, clone M2. RRID:AB_259529 | WB (1:20,000) |
| Antibody | Rabbit anti-V5-Tag (D3H8Q) | Cell Signaling Technology | Cat. #13202S. RRID:AB_2687461 | WB (1:10,000) |

*Continued on next page*

*Continued*

| Reagent type (species) or resource | Designation | Source or reference | Identifiers | Additional information |
|---|---|---|---|---|
| Peptide, recombinant protein | Human Semaphorin 3E | R and D Systems | Cat. #3239-S3B | Working concentration of 2 nM (prepared in 1xPBS with 0.1%BSA (SIGMA_ALDRICH, Cat.A8022) |
| Chemical compound, drug | SU5416 | SIGMA-ALDRICH | Cat. #S8442 | Working concentration of 0.2 μM in fish water. From 10.5 mM stock solution in DMSO (SIGMA-ALDRICH, Cat. #D8418) |
| Chemical compound, drug | Gelatin, from porcine skin | SIGMA-ALDRICH | Cat. # G1890-100G | Working concentration of 0.1% (prepared in distilled water and then autoclaved) |
| Chemical compound, drug | Blasticidin S HCl, powder | ThermoFisher Scientific | Cat. #R21001 | From a stock solution of 10 mg/ml. Prepared in UltraPure Distilled water (Invitrogen Cat. # 10977–015) |
| Chemical compound, drug | Puromycin Dihydrochloride | ThermoFisher Scientific | Cat. #A1113803 | From a stock solution of 10 mg/ml |

## Biochemistry and cell culture experiments

### Yeast two-hybrid screen

The *pGBK-T7* and *pACT2* vectors (Clonetech Laboratories, Inc) were used, respectively, for the bait and the preys. The bait consisted of a 96 kDa fusion protein harboring N-terminally the Myc-tagged DNA-binding domain of GAL4 and the cytosolic tail of mPLXND1 (632 aa) at its carboxyl end. A pre-transformed cDNA library from E11 stage mouse embryos undergoing PLXND1-dependent vascular and neuronal development (*Gitler et al., 2004*; *van der Zwaag et al., 2002*) (Clonetech Laboratories, Inc) was used to make prey proteins fused to the C-terminus of an HA-tagged GAL4 activation domain. The screening ($2.8 \times 10^6$ preys) was performed by Dualsystems Biotech AG and yielded 15 bait-dependent clones.

## Co-immunoprecipitation of mGIPC-mPLXND1 complexes in COS7 cells

### Details for the cell line used

COS-7 cells (Monkey Kidney Fibroblasts) purchased from the American Type Culture Collection (ATCC); ATCC #CRL-1651 (unrecorded lot number, thus the Certificate of Analysis containing detailed authentication and mycoplasma contamination information is unretrievable from ATCC's website).

### Cell culture and transfection

COS-7 cells were cultured in 10 cm diameter culture dishes (Falcon #379096) with Dulbecco's modified Eagle's medium (DMEM; Corning cellgro #10–013-CV) and 10% fetal bovine serum (FBS; Gemini Bio-Products #100–106). Cells were grown using a 37°C humidified 5% $CO_2$ atmosphere. Transfections were performed at 70% confluency with Lipofectamine2000 (Thermofisher Scientific #11668027) according to the manufacturer's specifications using 2.5 ug of each plasmid. Four hours post-transfection the medium was replaced with fresh complete DMEM. Expression vectors and their V5-tagged mPLXND1 and FLAG-tagged mGIPC protein payloads are described in *Supplementary file 1* and *Supplementary file 8*.

### Immunoprecipitation and western blotting

48 post-transfection, whole cell lysates were prepared as follows. Dishes were placed on ice, rinsed once with PBS (phosphate-buffered saline) and lyzed in 1 ml of ice-cold lysis buffer (50 mM Tris pH 7.5, 5 mM EDTA, 150 mM NaCl, 1% Triton X-100) supplemented with the EDTA-free protease inhibitor cocktail c0mplete (Roche #12683400). FLAG-tagged GIPC proteins were immunoprecipitated overnight at 4°C with rocking using an anti-FLAG antibody (SIGMA #F1804) and 30 ul of Protein A/G PLUS-Agarose (Santa Cruz Biotechnology #sc-2003). Next day, the protein immunoprecipitates were

washed 3x with washing buffer (50 mM Tris-HCl pH 7.5 and 150 mM NaCl). Samples were loaded onto pre-casted NuPAGE 4–12% Bis-Tris protein gels (Invitrogen #NP0335BOX). The immunoprecipitated FLAG-GIPC1 forms and the interacting co-immunoprecipitated V5-tagged mPLXND1 proteins were detected by Western blotting using anti-FLAG (SIGMA #F1804) and anti-V5 antibodies (CST #13202S), respectively.

## Quantification of protein-protein interactions between V5-C-mPLXND1 forms and FLAG-mGIPC1$^{WT}$ from FLAG immunoprecipitates and V5 co-immunoprecipitates

Data were collected from three independent experiments for each protein interaction and analyzed using a one-way ANOVA test followed by a Tukey post hoc analysis. See *Figure 1C* (left-side bar graph and legend) and *Supplementary file 2*. Data analysis was carried out using the SPSS Statistics 23.0 software package and the Laerd Statistics tutorial (*Statistics, 2015a*).

## Quantification of the relative abundance of V5-C-mPLXND1 forms and FLAG-mGIPC1$^{WT}$ from total cell lysates (TCL)

Data were collected from three independent experiments for each protein interaction and analyzed using a Kruskal-Wallis H test. See *Figure 1C* (right-side bar graph and legend) and *Supplementary file 2*. Data analysis was carried out using the SPSS Statistics 23.0 software package and the Laerd Statistics tutorial (*Statistics, 2015b*).

## Experiments with HUVEC/TERT2 cells

### Details for the cell line used

Immortalized HUVEC cells (HUVEC/TERT2) purchased from the American Type Culture Collection (ATCC); ATCC #CRL-4053 (authenticated via STR profiling and mycoplasma negative).

### Construction of vectors for Cas9 and gRNA coexpression

We used restriction cloning (BsmBI) to clone into the lentiviral vector lentiCRISPR v2-Blast (a gift from Mohan Babu; Addgene plasmid # 83480) pairs of aligned oligos to make the following gRNAs: non-targeting gRNA1, non-targeting gRNA2, *PLXND1*-KO1, and *PLXND1*-KO2 (*Supplementary file 1*); as in (*Shalem et al., 2014*; *Sanjana et al., 2014*). The four resulting vectors were propagated in Stbl3 bacteria (ThermoFisher Scientific #C737303) at 32°C.

### Cell culture, lentivirus production, and infection with lentiCRISPR v2-Blast vectors for Cas9 and gRNA coexpression

Immortalized HUVEC cells were cultured in low serum medium optimized for human endothelial cells and without human growth factors (VascuLife EnGS Endothelial Medium Complete Kit from Lifeline Cell Technology #LL-0002). Cells were grown in 10 cm diameter culture plates (Falcon #379096) precovered with 0.1% gelatin (SIGMA #G1890-100G) and incubated in a humidified 5% $CO_2$ atmosphere at 37°C. Human embryonic kidney 293 T cells (HEK293T; gift of Matthias Stadtfeld; NYU) were cultured in high D-Glucose Dulbecco's modified Eagle's media (Gibco # 10313–021) containing 10% FBS and 2 nM L-glutamine (Gibco #25030–081). Plates (10 cm) of human embryonic kidney 293T (90% confluence) were individually transfected with one of the four lentiCRISPR v2-Blast constructs along with lentivirus packaging and envelope plasmids (gift of Matthias Stadtfeld; NYU) using TransIT-293 reagent (Mirus Bio #Mir2700). Each plate was transfected with 15 µg of the lentiCRISPR v2-Blast construct, 0.75 µg of each lentivirus packaging vector (tat, rev, and gag/pol) and 1.5 µg of pVSV-G. Viral supernatants were harvested and filtered through 0.45 µm at 48 and 72 h. HUVEC cells were infected at ~50% confluency using 100 µl of viral supernatants and 5 µg/ml polybrene (Millipore #TR-1003-G) for 2 consecutive days. Pools of stable HUVEC cells were selected for 5 days with 4 µg/ml blasticidin (ThermoFisher Scientific # R21001). Pilot studies confirmed that these selection conditions killed 100% of the uninfected cells.

## Isolation of monoclonal cell populations of *PLXND1* knockout cells infected with lentiCRISPR v2-Blast vectors for Cas9 and gRNA coexpression

Pools of stable HUVEC cells were harvested using trypsin (Gibco #25300–054) and grown at a very low density in 15 cm diameter culture plates (Corning #430599). Twelve individual cells from each of the *PLXND1*-KO1 and *PLXND1*-KO2 gRNAs expressing lentiCRISPR v2-Blast vectors were grown to 100% confluence in 96-well plates. These cells were then expanded in larger plates until confluent 10 cm plates were obtained. The resulting monoclonal cell populations were used for characterizing the DNA sequence (GENEWIZ performed sequencing) and protein level via Western blot to identify *PLXND1* knockout lines. Two of these (*PLXND1* gRNA KO1 and KO2) were used for the experiments presented here.

## shRNA infection

Blasticidin-resistant stable cells infected with lentiCRISPR v2-Blast vectors for Cas9 and gRNA coexpression were infected as follows. Briefly, pools of cells expressing a non-targeting vector (gRNA1 or gRNA2) and monoclonal *PLXND1* KO lines (*PLXND1* gRNA KO1 or KO2) were infected at ~ 50% confluency using 5 µg/ml polybrene (Millipore #TR-1003-G) for 2 consecutive days. Infections were performed with the puromycin-resistant shRNA lentiviral particles, with the non-targeting control, *GIPC1/2* pool (Santa Cruz Biotechnology #sc-108080, sc-35475-V, and sc-75132-V). The medium was changed 24 h after the second infection, and infected cells were selected with 1.5 µg/ml puromycin (ThermoFisher Scientific #A1113803) for 72 hr.

## Detection of phospho-active (pERK) and total ERK (ERK$_{Total}$) and quantification of relative ERK activity

Cells were stimulated with either vehicle or 2 nM recombinant human Semaphorin 3E (R and D Systems #3239-S3B) for 15 and 45 mins. pERK and ERK$_{Total}$ levels were measured using Western blots from total cell lysates (TCLs) prepared as follows. Cell plates were placed on an ice bed, rinsed with PBS and lyzed in 250 µl of lysis buffer (50 mM Tris pH 7.5, 0.5 mM EDTA, 150 mM NaCl, 1 mM EDTA, 1% SDS) supplemented with protease (cOmplete, Roche #12683400) and phosphatase (1 mM sodium fluoride, 10 mM β-glycerophosphate, and 1 mM sodium vanadate) inhibitors. Lysates were loaded onto NuPAGE 4–12% Bis-Tris protein gels (Invitrogen #NP0336BOX) and proteins transferred to Immobilon-P membranes (Millipore; #IPVH00010). pERK and ERK$_{Total}$ were immunodetected with rabbit and mouse antibodies against ERK1/2 (CST #4370S and #4696S, respectively) and revealed with Western Lightning PLUS-ECL (PerkinElmer #NEL103001EA). Relative ERK activity was calculated ratiometrically as pERK/ERK$_{Total}$.

## **Experiments with primary HUVEC cells**

### Details for the cell line used

Normal primary human umbilical vein endothelial cells (HUVEC) were purchased from Lifeline Cell Technology; #FC-0003 (STR profiling data unavailable, mycoplasma negative).

### Cell collapse assays

Cells between passages 2 and 6 were infected with shRNA lentiviral particles and puromycin-selected for 48 h as described above. Cells were divided into two pools, one for validating the knockdown and the other for performing the cellular morphology assay. Cells for the latter experiment were starved for 6 h and then treated with either vehicle or 10 nM recombinant human Semaphorin 3E (R and D Systems #3239-S3B) for 45 min. Cells were washed with ice-cold 1xPBS, fixed at room temperature (RT) for 15 min with 4% PFA in PBS. Cells were then washed 2x with PBS and permeabilized with 0.1% Triton-X100 in PBS for 5 min at RT. Next, cells were incubated for 20 min at RT in 20 µg/ml phalloidin-tetramethylrhodamine B isothiocyanate (SIGMA #P1951) to label the F-actin cytoskeleton. Cells were next washed 3x in PBS, incubated for 5 min at RT in 0.5 µg/ml DAPI (Molecular Probes #D1306) to visualize nuclei, and kept in PBS at 4°C until imaging. Fluorescent images were acquired with an Eclipse Ti-E inverted microscope (Nikon) using the 10X objective (NA 0.3).

## Cell collapse data collection and evaluation

Images were collected from 50 to 100 cells from each of three independent experiments per knockdown and treatment condition. Images were processed with FIJI (https://imagej.net/Fiji). As automated extraction of cellular contours was unfeasible, cells were qualitatively classified as uncollapsed, collapsed, or hyper-collapsed based on their size and morphology (*Figure 7—figure supplement 3*).

## Western blot validation of *GIPC* knockdowns and *PLXND1* knockout in primary and immortalized HUVEC

Western blots of TCLs were performed with rabbit antibodies for GIPC1 (Proteintech Group; 14822–1-AP) and GIPC2 (Abcam #ab175272). A mouse antibody was used for PLXND1 (R and D Systems #MAB41601). GAPDH (loading control) was detected with a rabbit antibody (CST; #5174P).

## Zebrafish experiments

### Genome editing for making *gipc* and *plxnd1* mutants

gRNA design and genomic target site selection were performed using the *CHOPCHOP* and *CRISP-Rscan* web tools (*Labun et al., 2016*; *Montague et al., 2014*; *Moreno-Mateos et al., 2015*). gRNAs were in vitro transcribed from PCR-assembled oligo templates (Integrated DNA Technologies) as in (*Gagnon et al., 2014*) (see *Supplementary file 1*). G0 fish were made by delivering 1 nl of a genome editing mix containing *S. pyogenes* Cas9 nuclease and one or more gene-specific gRNAs (125 pg/per gRNA) into the cytoplasm of one cell stage *Tg(fli1a:EGFP)^y1* embryos (see *Auer and Del Bene, 2014*; *Auer et al., 2014*; *Chang et al., 2013*; *Cong et al., 2013*; *Cong and Zhang, 2015*; *Gagnon et al., 2014*; *Hruscha et al., 2013*; *Hwang et al., 2013*; *Irion et al., 2014*; *Kimura et al., 2014*; *Mali et al., 2013*; *Talbot and Amacher, 2014*). Cas9 was provided as mRNA (300 pg; in vitro transcribed from the *pST1374-NLS-flag-linker-Cas9* vector (*Shen et al., 2013*), a kind gift from Xingxu Huang; Addgene plasmid #44758) or protein (500 pg; PNA Bio #CP01). Genome editing of G0 embryos was assessed by evaluating PCR products amplified with BioReady Taq DNA Polymerase (Bulldog Bio #BSAX050) from gDNA of 4–12 pooled individuals via agarose gel electrophoresis and single-colony Sanger sequencing. For the latter, PCR products were TOPO-TA cloned into the *pCRII-TOPO* or *pCR2.1-TOPO* vector (ThermoFisher Scientific; #450640 or #450641), and 8–20 individual colonies were sequenced. To identify F1 heterozygous mutant carriers, gDNA from individuals was PCR-amplified, and the biallelic sequence trace analyzed using the *Poly Peak Parser* and *TIDE* web tools (*Hill et al., 2014*; *Brinkman et al., 2014*). The molecular nature of the mutant alleles of F1 parents was confirmed using single-colony Sanger sequencing of PCR products amplified from fin clip-derived gDNA GENEWIZ performed sequencing.

### Forced endothelial expression of 2xHA-Plxnd1 forms in *plxnd1^fov01b* mutants

We used the GAL4/*UAS* system (*Scheer and Campos-Ortega, 1999*) and *Tol2*-based transient transgenesis (*Kikuta and Kawakami, 2009*) to drive forced mosaic endothelial-specific expression of 2xHA-tagged forms of zebrafish Plxnd1 (2xHA-Plxnd1^WT or 2xHA-Plxnd1Δ^GBM) and the green fluorescent marker EGFP. Briefly, 1 nl of a 100 pg of Tol2 mRNA and 20 pg of vector DNA solution was injected into the cytoplasm of one-cell *plxnd1^fov01b* mutants carrying both the endothelial *Tg(fli1a:GAL4FF)^ubs4* GAL4 driver (*Zygmunt et al., 2011*) and the red nuclear arterial *Tg(flt1:nls-mCherry)^skt7* reporter (this study). Faithful coexpression of 2xHA-Plxnd1 and EGFP was accomplished using an IRES (internal ribosomal entry site element)-based bicistronic *UAS* cassette (*Kwan et al., 2007*). Embryos were fixed at 32 hpf, immunostained and imaged as described below.

### Quantification of the vascular patterning activity of exogenous 2xHA-Plxnd1 forms in *plxnd1^fov01b* mutants

Arterial endothelial clones with exogenous expression of 2xHA-Plxnd1 forms (EGFP$^+$) that occupied the Se, DLAV, or both (Se-DLAV) positions were qualitatively scored as exhibiting a WT-like phenotype if they were properly shaped and positioned according to the following definitions. For Se sprouts: unbranched chevron morphology and, if connected to the DA, a base located just anterior to the somite boundary. For DLAVs: unbranched linear morphology parallel to the roof of the spinal cord (*Figure 2—figure supplement 2D–J* and *Supplementary file 3*).

## Embryo treatments with SU5416 and DMSO

SU5416 (SIGMA #S8442) was prepared and used as in *Covassin et al. (2006)* and *Stahlhut et al. (2012)*. Briefly, a 200 µM SU5416 stock solution in DMSO (SIGMA #D8418; vehicle) was dissolved in fish water to a final concentration of 0.2 µM SU5416 (a suboptimal dose) and 0.1% DMSO. Control, vehicle-only (0.1% DMSO) treatments were also performed. Homozygous WT and homozygous *plxnd1$^{skt6}$* mutant embryos were manually dechorionated before receiving the SU5416 and DMSO treatments using a common solution for both genotypes. Embryos were treated from 18 to 32 hpf to specifically target both Se and DLAV angiogenesis (see *Torres-Vázquez et al., 2004*; *Zygmunt et al., 2011*; *Childs et al., 2002*; *Isogai et al., 2003*; *Zygmunt et al., 2012*; *Yokota et al., 2015*), and then fixed for immunostaining.

## Quantification of angiogenesis deficits in the trunk's arterial tree of WT and mutant zebrafish embryos

Embryos carrying the *Tg(fli1a:EGFP)$^{y1}$* vascular reporter were fixed at 32 hpf, genotyped using a tail biopsy, immunostained to visualize both the vasculature and the somite boundaries and confocally imaged as described below. Scoring of angiogenesis deficits was performed bilaterally in the ~six somite-long region dorsal to the yolk extension using one of the two following scoring methods and with knowledge of the genotype. Scoring of 'Se and DLAV truncations': this scoring method emphasizes truncations found within the dorsal side of the trunk's arterial tree. This method was used for *Figure 2C,E,F–I* (see also *Figure 2—figure supplement 1* and *Supplementary file 3*), *Figure 3* (see also *Figure 3—figure supplement 1* and *Supplementary file 4*), and *Figure 5* (see also *Figure 5—figure supplement 1* and *Supplementary file 6*). The classification of Se and DLAV truncations is based on the relative span of Se and DLAVs along the dorsoventral and anteroposterior axes, respectively, using as reference the following landmarks. The horizontal myoseptum and the actual (or expected) level of the DLAV. Four phenotypic classes are used to define Se and DLAV spans. The three Se-DLAV truncation categories are as follows: maximal (includes both Se that are missing and those that fail to grow dorsally past the horizontal myoseptum); moderate (includes Se that grow dorsally past the horizontal myoseptum but not further than half the distance between the horizontal myoseptum and the level of the DLAV); and minimal (Se that grow dorsally past half the distance between the horizontal myoseptum and the level of the DLAV, but which form an incomplete DLAV). An incomplete DLAV is one that fails to span the distance between the anteriorly and posteriorly flanking ipsilateral somite boundaries. Non-truncated Se-DLAV: full (Se that grow dorsally to the level of the DLAV and that form a complete DLAV). Scoring of 'Se truncations': this scoring method emphasizes truncations found within the ventral side of the trunk's arterial tree. This method was used for *Figure 4* (see also *Figure 4—figure supplements 2–3* and *Supplementary file 5*). The classification of Se truncations is based on the relative span of Se vessels measured along the dorsoventral axis. Briefly, the length of a perpendicular line traced between the actual (or expected) Se sprouting site and the level of the DLAV was assigned a value of 100%. Four phenotypic classes are used to define Se span. Se truncation categories: severe (0–25%, includes missing Se), medium (26–50%), and weak (51–75%). Non-truncated: complete (76–100%). Note that both the Se-DLAV and Se scales used for quantifying angiogenesis deficits are based on the relative span of the vascular structures scored, which eliminates the noise that variations in size between embryos with similar body proportions would otherwise introduce.

## Comparisons of angiogenesis deficits between genotypes and treatments

Significance statistical values were calculated using a two-sided Fisher Exact test with the aid of the SISA web tool (http://www.quantitativeskills.com/sisa/statistics/fiveby2.htm). When performing more than one pairwise comparison (as in *Figures 3–4*), significant differences were assigned using a Bonferroni adjustment. This conservative adjustment reduces the likelihood of obtaining false-positive results (type I errors or the rejection of true null hypotheses) when simultaneously applying many statistical tests to a dataset (*Statistics, 2016*). The Bonferroni adjustment is commonly used to compare drug effects and in angiogenesis studies (see *Hamada, 2018*; *Weichand et al., 2017*; *Basagiannis et al., 2016*).

## Morpholino injection and analysis of *plxnd1* morphants

A previously validated splice-blocking *plxnd1* morpholino (Gene Tools, LLC) that faithfully phenocopies the vascular defects of the *plxnd1^fov01b* null allele (*plxnd1^3207-3462*; see **Torres-Vázquez et al., 2004**) was used. *plxnd1^3207-3462* morpholino (2.5 ng) was injected into each of the one-cell stage embryos used in the experiment. The morpholino was injected into both WT *gipc1^skt1(MZ)*; *gipc2^skt4(MZ)* double mutants carrying the *Tg(fli1a:EGFP)^y1* reporter. Both un-injected and embryos injected with the *plxnd1* morpholino were fixed at 32 hpf to evaluate the trunk vasculature.

## Fixing, immunostaining, and mounting of zebrafish embryos

Embryos were fixed overnight at 4°C in 4% paraformaldehyde (Santa Cruz #sc-281692) dissolved in PBS (1x phosphate buffered saline pH 7.5). Embryos were then washed 6x with PBST (PBS with 0.2% Tween; Sigma #P1379) for 10 min/each and placed in blocking solution (PBST with 1% bovine serum albumin; Sigma #A8022) for 1 h. Embryos were then incubated overnight at 4°C with primary antibodies diluted in blocking solution and then washed with PBST 6x for 10 min/each. Embryos were then incubated overnight at 4°C with fluorescent secondary antibodies diluted in blocking solution for 2 h at RT or overnight at 4°C. Embryos were then washed in PBST 4x for 15 min/each and mounted in an aqueous 0.5% agarose solution before imaging. Primary antibody dilutions used as follows. To label somite boundaries, mouse anti-pFAK Tyr397 (Millipore #05–1140 at 1:1000); to enhance the visualization of EGFP, chicken anti-GFP (Invitrogen #A10262 at 1:1000); to improve mCherry detection, sheep anti-mCherry (custom-made at 1:1000; a gift from Holger Knaut). The following secondary antibody dilutions were used: Alexa Fluor 488 goat anti-chicken (Invitrogen #A11039 at 1:1000), Cy3 donkey anti-sheep (Jackson ImmunoResearch #713-165-147 at 1:1000), and Alexa Fluor 647 donkey anti-mouse (Invitrogen #A31571 at 1:1000).

## Confocal imaging of zebrafish embryos

Confocal images of fixed immunostained embryos were taken with a Leica TCS SP5 microscope using a water dipping 40x lens (0.8 NA). The 488, 561, and 647 nm laser lines were used. Images were processed with FIJI (https://imagej.net/Fiji).

## Genotyping of *gipc* and *plxnd1* mutants and *Tg(fli1a:GAL4FF)^ubs4* transgenics

Genotyping was done using PCR products amplified from gDNA (see **Supplementary file 1** for primers), with *plxnd1^fov01b* genotyping as in **Zygmunt et al. (2011)**. All PCR products were treated with ExoSAP-IT (ThermoFisher Scientific; #78201.1 ML) before sequencing. PCR conditions included BioReady Taq DNA Polymerase (Bulldog Bio #BSAX050), used according to the manufacturer's recommendations. Abbreviations. Size: length of the PCR amplicon from the WT (wild-type) or M (mutant) allele. Notes: N-PCR (nested PCR). Genotyping methods: AGE (agarose gel electrophoresis), HMA (heteroduplex mobility assay (**Ota et al., 2013**), and Sanger sequencing (sequencing; performed by GENEWIZ). Detailed protocols are available upon request.

| Allele | PCR conditions and comments | | | | | | Genotping methods | | |
| --- | --- | --- | --- | --- | --- | --- | --- | --- | --- |
| | Primer pair (ID#s) | Align (T°C) | Extend (secs) | Cycles | Size in bp (WT/M) | Notes | AGE | HMA | Sequencing Primer ID# |
| *gipc1^skt1* | 12891–12889 | 62 | 25 | 34 | 372/368 | N-PCR (if fixed) | n/a | For hetz | 12891 |
| | 13655–13653 | 64 | 19 | 40 | 300/296 | | n/a | For hetz | 13653 |
| *gipc1^skt2* | 12891–12889 | 57 | 30 | 34 | 372/379 | n/a | n/a | For hetz | 12891 |
| *gipc2^skt3* | 13667–13422 | 60 | 14 | 35 | 218/216 | n/a | n/a | n/a | 13421 |
| *gipc2^skt4* | 13667–13422 | 60 | 14 | 35 | 218/172 | n/a | Yes | n/a | 13421 |
| *gipc3^skt5* | 13510–13512 | 67.7 | 25 | 38 | 377/337 | n/a | Yes | n/a | 13512 |

*Continued on next page*

*Continued*

| | PCR conditions and comments | | | | | | Genotping methods | | |
|---|---|---|---|---|---|---|---|---|---|
| Allele | Primer pair (ID#s) | Align (T°C) | Extend (secs) | Cycles | Size in bp (WT/M) | Notes | AGE | HMA | Sequencing Primer ID# |
| $plxnd1^{skt6}$ | 11844–11845 | 65.9 | 30 | 34 | 442/438 | N-PCR | n/a | n/a | n/a |
| | 12532–11845 | 57.7 | 30 | 34 | 262/258 | | n/a | n/a | 12532 |
| $plxnd1^{fov01b}$ | 13498–10167 | 63 | 16 | 38 | 519/519 | N-PCR | n/a | n/a | 13646 |
| | 13646–13647 | 57.5 | 11 | 38 | 395/395 | | n/a | n/a | |
| $Tg(fli1a: GAL4FF)^{ubs4}$ | 11730–11731 | 57.5 | 45 | 35 | 402 | Only Tg presence | Yes | n/a | n/a |

## Zebrafish lines and maintenance

### Zebrafish mutant alleles

The *plxnd1* (*plxnd1^{skt6}*), *gipc1* (*gipc1^{skt1}* and *gipc1^{skt2}*), *gipc2* (*gipc2^{skt3}* and *gipc2^{skt4}*), and *gipc3* (*gipc3^{skt5}*) alleles were made via genome editing for this study (above) and are described here at the genomic DNA (*Supplementary file 1*) and protein (*Figure 2B*, *Figure 4—figure supplement 1* and *Supplementary file 8*) levels. The null *plxnd1^{fov01b}* allele is described in *Torres-Vázquez et al. (2004)*. All *gipc* alleles are predicted nulls encoding small truncated proteins lacking all three domains based on the sequence of the targeted genomic regions (the corresponding cDNAs were not sequenced). The potential instability of mutant *gipc* transcripts and the occurrence of transcriptional adaptation in *gipc* mutants were left untested (see *El-Brolosy and Stainier, 2017*).

### Zebrafish transgenic lines

The red fluorescent reporters *Tg(kdrl:HsHRAS-mCherry)^{s896}* and *Tg(flt1:nls-mCherry)^{skt-1}* were used to visualize, respectively, endothelial cell membranes (*Chi et al., 2008*) and arterial endothelial cell nuclei (this study; see also *Bussmann et al., 2010*). The green fluorescent cytosolic reporter *Tg(fli1a: EGFP)^{y1}* was used to label the endothelium (*Lawson and Weinstein, 2002*). The *Tg(fli1a:GAL4FF)^{ubs4}* endothelial GAL4 line (*Zygmunt et al., 2011*) was used to drive expression of *UAS* constructs and was genotyped via PCR (see *Supplementary file 1* and the Materials and methods section 'Genotyping of *gipc* and *plxnd1* mutants and *Tg(fli1a:GAL4FF)^{ubs4}* transgenics').

### Ethics statement

Zebrafish embryos and adults were kept and handled using standard laboratory conditions at New York University and under IACUC-app roved animal protocols (#151202–01 and #170103–01).

## Acknowledgements

Funding sources: This work was supported by postdoctoral fellowships from Mexico's Consejo Nacional de Ciencia y Tecnología (CONACyT 187031 and 203862, to JC-O) and by the National Heart Lung and Blood Institute, NIH (1R01HL133687-01, to JT-V). For reagents and protocols: Moses Chao (FLAG-mGIPC1 construct; see *Lou et al., 2001*), Holger Knaut (mCherry antibody), Gregory Palardy and Ajay Chitnis (zebrafish genome editing protocol with Cas9 protein), and Xingxu Huang (*pST1374-NLS-flag-linker-Cas9* vector; Addgene plasmid #44758; see *Shen et al., 2013*). For useful discussions: Knaut lab members and other members of the Skirball Institute's Developmental Genetics Program. Microscopy: We thank Yang Deng, Michael Cammer and NYU School of Medicine's Microscopy Laboratory for data collected with the Leica SP5 confocal system (grant NCRR S10 RR024708). We thank Agnel Sfeir, Mamta V Tahiliani and Matthias Stadfeld for access to the Eclipse Ti-E inverted microscope (Nikon) used for collecting HUVEC collapse data and for reagents.

# Additional information

## Funding

| Funder | Grant reference number | Author |
| --- | --- | --- |
| National Institutes of Health | 1R01HL133687-01 | Jesús Torres-Vázquez |
| Consejo Nacional de Ciencia y Tecnología | 187031 | Jorge Carretero-Ortega |
| Consejo Nacional de Ciencia y Tecnología | 203862 | Jorge Carretero-Ortega |

The funders had no role in study design, data collection and interpretation, or the decision to submit the work for publication.

## Author contributions

Jorge Carretero-Ortega, Jesús Torres-Vázquez, Conceptualization, Formal analysis, Funding acquisition, Methodology, Project administration, Resources, Supervision, Visualization, Writing—original draft, Writing—review and editing, Investigation; Zinal Chhangawala, Conceptualization, Formal analysis, Investigation, Resources, Methodology, Writing—review and editing, Visualization; Shane Hunt, Carlos Narvaez, Javier Menéndez-González, Carl M Gay, Tomasz Zygmunt, Resources, Investigation, Methodology, Writing—review and editing; Xiaochun Li, Formal analysis, Resources, Writing—review and editing

## Author ORCIDs

Jesús Torres-Vázquez (iD) http://orcid.org/0000-0002-3808-3978

## Decision letter and Author response

Decision letter https://doi.org/10.7554/eLife.30454.030
Author response https://doi.org/10.7554/eLife.30454.031

# Additional files

## Supplementary files

• Supplementary file 1. Miscellaneous tables listing the following information. Vectors for expressing PLXND1 and GIPC proteins/fragments, primers for genotyping Tg(fli1a:GAL4FF)$^{ubs4}$ zebrafish, oligos for assembling DNA templates for in vitro transcription of gRNAs for zebrafish genome editing and for making lentiCRISPRv2-Blast vectors for Cas9 and gRNA coexpression for use in HUVEC, cognate sequences of WT alleles and mutant alleles generated in this study via genome editing, and primers for genotyping mutant alleles generated in this study via genome editing. Related to *Figures 1–7*, *Figure 2—figure supplement 1*, *Figure 2—figure supplement 2*, *Figure 4—figure supplement 1*, *Figure 4—figure supplement 2*, *Figure 5—figure supplement 1*, *Figure 7—figure supplement 1* and, *Figure 7—figure supplement 2*.
DOI: https://doi.org/10.7554/eLife.30454.021

• Supplementary file 2. Tables of the raw and average densitometry values of tagged proteins in Western blots of CoIP experiments and their statistical significances. Related to *Figure 1*.
DOI: https://doi.org/10.7554/eLife.30454.022

• Supplementary file 3. Tables of the *plxnd1*$^{skt6}$ complementation of *plxnd1*$^{fov01b}$ (related to *Figure 2C–E*), the comparison of the vascular phenotypes of homozygous WT and homozygous *plxnd1*$^{skt6}$ mutant siblings (related to *Figure 2F–I*, *Figure 2—figure supplement 1*), and the mosaic transgenic endothelial expression of tagged forms of zebrafish Plxnd1 in *plxnd1*$^{fov01b}$ null mutants (related to *Figure 2—figure supplement 2J*).
DOI: https://doi.org/10.7554/eLife.30454.023

• Supplementary file 4. Tables comparing the Se-DLAV truncations of wild-type embryos and *plxnd1*$^{skt6}$ mutants (at 32 hpf) in animals treated with DMSO and SU5416. Related to *Figure 3E* and *Figure 3—figure supplement 1*.

DOI: https://doi.org/10.7554/eLife.30454.024

• Supplementary file 5. Tables comparing the Se truncations of wild-type embryos and *gipc* mutants at 32 hpf. Related to *Figure 4B* and *Figure 4—figure supplement 3*.
DOI: https://doi.org/10.7554/eLife.30454.025

• Supplementary file 6. Tables comparing the Se-DLAV truncations of *gipc1*[skt1 (MZ)] and *gipc1*[skt1 (MZ)]; *plxnd1*[fov01]/+ mutants at 32 hpf. Related to *Figure 5C* and *Figure 5—figure supplement 1*.
DOI: https://doi.org/10.7554/eLife.30454.026

• Supplementary file 7. Tables of raw and average densitometry values for both pERK and $ERK_{Total}$, relative ERK activities and the statistical significances of the latter. Related to *Figure 7E* and *Figure 7—figure supplement 1*.
DOI: https://doi.org/10.7554/eLife.30454.027

• Supplementary file 8. Protein sequences. Related to *Figure 1*, *Figure 2A–B*, *Figure 4—figure supplement 1*, *Figure 7—figure supplement 2*, *Supplementary file 1* (see 'Vectors for expressing PLXND1 and GIPC proteins/fragments' and 'Cognate sequences of WT alleles and mutant alleles generated in this study via genome editing'), and *Supplementary file 2*.
DOI: https://doi.org/10.7554/eLife.30454.028

• Transparent reporting form
DOI: https://doi.org/10.7554/eLife.30454.029

### Data availability

All data generated or analysed during this study are included in the manuscript and supporting files.

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
