## [Decision Letter]

Thank you for sending your article entitled "GIPC proteins pattern the vasculature by limiting the antiangiogenic activity of the Semaphorin receptor PlexinD1" for peer review at *eLife*. Your article has been evaluated by three peer reviewers, and the evaluation is being overseen by a Reviewing Editor and Marianne Bronner as the Senior Editor.

The reviewers have discussed the reviews with one another, and their discussion has raised a number of issues (elaborated below) that would need to be addressed through revisions to your manuscript. Since some of these revisions would likely require you to perform new experiments, we ask that you respond to this letter within the next two to three weeks with a specific action plan and timetable for the completion of the additional work. Please bear in mind that the standard timeline for a revision for *eLife* is only 2-3 months. The Reviewing Editor and reviewers will then consider your plans, assess the feasibility of an effective response to their concerns, and offer a binding recommendation (such as an official invitation to revise your manuscript for *eLife*). Alternatively, if you would prefer to submit your manuscript elsewhere, rather than revising it for *eLife*, please let us know.

Summary:

In this interesting study, Carretero-Ortega and colleagues identify the GIPC endocytic adaptors as the first negative intracellular regulators of PlxnD1 signaling, expanding our knowledge of the network controlling early arterial angiogenesis. The authors first use biochemical assays with truncated forms of PlxnD1 to identify a GIPC-Binding Motif (GBM) in the cytosolic tail of the protein, which they find to be essential for GIPC binding. In addition, they confirm the involvement of the PDZ and GH1 domains of GIPCs in the formation of the PlxnD1-GIPC1 complex. In order to characterize the role of GIPCs on the vascular activity of PlxnD1, the authors compare the effects of focused, endothelium-specific overexpression of PlxnD1 and truncated forms of PlxnD1 that do not bind GIPCs. Finally, they use CRISPR/Cas9-mediated genome editing to generate the first gipc1, gipc2, and gipc3 zebrafish mutants, and they analyze their vascular phenotypes and make a case for their angiogenic defects being caused by increased PlxnD1 signaling. Finally, they show that GIPC depletion increases Sema3E-induced PlxnD1-dependent responses in HUVEC cells. Overall, the studies are straightforward, and the data have the potential to have an important impact on our understanding of angiogenesis signaling networks. However, there are several issues, described below, that need to be addressed more thoroughly in order to solidify support for the authors' conclusions.

Essential revisions:

1) This manuscript nicely uses an in vivo assay (Figure 2) to test whether GIPC-binding deficient mutants of PlxnD1 lose activity for vessel patterning. Surprisingly, both wild type PlxnD1 and PlxnD1ΔGBM are equally effective at restoring vessel patterning in PlxnD1 mutants. The authors interpret this to suggest that GIPCs are not required for PlxnD1 activity. The authors then suggest that either 'PlxnD1ΔGBM will be more potent than PlxnD1WT' or that 'both PlxnD1's activity and potency will be independent of GIPC-binding'. These two hypotheses are poorly worded as it is not clear what 'potency' and 'activity' refer to. By potency, do the authors mean that PlxnD1ΔGBM is insensitive to inhibition and therefore more active? What phenotype would a 'more active' PlxnD1 mutant have? For the second hypothesis, do the authors mean that while GIPC binds PlxnD1, it has no influence on activity? Also, and importantly, when comparing the influence of PlxnD1-WT and PlxnD1ΔGBM on "angiogenic potential", the authors speculate on the meaning of an observed trend that is not statistically significant. Could the authors increase the sample size for these observations? Would they then find evidence for a statistically significant difference? Finally, the conclusion that PlxnD1 signaling acts to limit endothelial proliferation should be confirmed through direct analysis of proliferation (e.g. by EdU labeling).

2) The intersegmental vessel truncation observed in zygotic gipc1 or gipc2 single mutants is not statistically significant. Therefore, rather than illustrating these so prominently in Figure 3, it might be more worthwhile to focus on the significant effects observed in double mutants and MZ mutants, which show a phenotype that has low penetrance. The compression of the graph's X-axis in Figure 3B to emphasize this phenotype is unusual and could be misleading. At least 80% of the embryo SeV are normal, and the way that the data are presented may give the impression that the phenotype is stronger than it actually is. It would also be helpful to provide statistical analysis in Figure 4—figure supplement 3 and to adjust the X-axis in Figure 4—figure supplement 3B. Since these results are key to the authors' conclusions, significance and robustness need to be clearly shown, particularly as the results in this manuscript do not agree with those in a published article on the same topic in mouse (Burk et al., 2017) It should be straightforward (in principle) for the authors to increase their n and procure a more robust result.

3) Even given its low penetrance, the gipc mutant phenotype is interesting as it is very different from the mouse GIPC knockout vascular phenotype that shows a PlxnD1-like phenotype in equivalent vessels (as opposed to fewer vessels in fish gipc mutants). Burk et al., 2017, claim (in their Figure 10D) that, in mouse embryos, loss of Gipc1 leads to ectopic ISV sprouting, the same phenotype observed in PlexinD1 mutants (not shown) and in Gipc1;PlexinD1 double heterozygote embryos. Although this is illustrated only briefly and not very convincingly by Burk et al., this published result seems to be in strong contrast to the gipc mutant phenotypes illustrated in Figure 3 of this manuscript. Why are fish and mouse null Gipc mutants different? An insightful discussion is needed, and the authors do not provide much material in this regard. It would also be helpful if the previous name of Gipc1 (Synectin) were to be mentioned in this manuscript.

4) The claim of Figure 4, that the angiogenesis defects of gipc mutants are partially due to increased PlxnD1 signaling, is based on the analysis of zygotic gipc1;plxnD1 double mutants. However, as gipc1 zygotic mutants had no statistically significant Se defect, this experiment might not be suitable for supporting the authors' claim. Notably, the double mutants are analyzed only qualitatively here, and no quantitation is provided. Importantly, if GIPC acts as a repressor of PlxnD1 activity, one would expect a rescue in the double mutant. Therefore, it is possible to argue that the authors have shown that, even though GIPC binds PlxnD1, its activity is not required to influence PlxnD1 signaling. However, the authors do not reach this conclusion. Perhaps PlxnD1 is epistatic to GIPC (i.e. acts downstream of GIPC). Even though GIPC binds PlxnD1, it could act upstream and/or in a different pathway to control angiogenesis of the SeV (such as in the neuropilin pathway or other pathways where GIPC is known to act). Finally, the fact that GIPC and PlxnD1 mouse double mutants (Burk et al.), have identical and not opposite phenotypes, unlike in the fish, is not discussed in any depth.

5) The data presented in Figure 6 clearly demonstrate that loss of GIPC potentiates loss of Sema3E-induced collapse. The authors conclude that this is PlxnD1-dependent because siRNA against PlxnD1 fails to induce the collapse. However, double knockdown of GIPC and PlxnD1, followed by Sema3E addition, is necessary to make this conclusion, and this experiment was not performed. Without double knockdown data, there is no evidence that PlxnD1, GIPC and Sema3E interact in collapse, as this not been tested in the same cells. The same argument can be made for the ERK data – again, PlxnD1-GIPC double knockdown, with and without Sema3E, in the same cells is a necessary experiment to support the conclusion. Alternatively, could the authors corroborate the ERK activation results (Figure 6) in zebrafish? Since all the zebrafish mutants are available, the use of HUVECs seems less relevant. ERK antibody staining has been extensively described in zebrafish, and it could be used to investigate ERK activation in the arteries of gipc mutants, as well as in gipc;plexind1 double mutants. This type of analysis has the potential to strengthen the authors' claims.

6) The authors seem to assume that GIPCs can mediate PlxnD1 activity only (or primarily) via direct binding through the identified domains. Although this was the way that the authors initially identified GIPCs (via 2-hybrid screening), can they truly rule out other possibilities? Additionally, the authors do not seem to consider other potential roles of GIPCs that are PlxnD1-independent. Although they state in their Results section that "the angiogenesis deficits of gipc mutants are, at least partially, due to increased PlxnD1 signaling", they do not assess or discuss additional possibilities. One relatively easy experiment that the authors could perform in order to address these questions would be to overexpress plexind1 (WT and truncated forms) in endothelial cells of gipc single and double mutants. Alternatively, the authors could consider the possible relevance of the known interactions of GIPC with Neuropilin. While Neuropilin can act as a Plexin coreceptor, it can also act in a Plexin-independent manner. This is of relevance especially in experiments analyzing growth cone collapses or other effects on morphology, where Neuropilin has been shown to act independently of Plexin. Can the authors distinguish the negative regulation of Plexin from a potential positive regulation of Neuropilin? This is of course hampered by the caveat that there are no reliable Neuropilin reagents in zebrafish available as morpholino and mutant data only partially match. Maybe different downstream effectors could be affected pharmacologically? At a minimum, it would be helpful for the authors to provide more discussion of alternate possibilities that could explain the observed ways in which GIPC function affects PlxnD1 activity.

[Editors' note: the authors’ plan for revisions was approved and the authors made a formal revised submission.]

Thank you for submitting your revised manuscript entitled "GIPC proteins pattern the vasculature by limiting the antiangiogenic signaling of the Semaphorin receptor Plexind1" for consideration by *eLife*. Your revised submission has been reviewed by two out of the three original peer reviewers, and the evaluation has been overseen by a Reviewing Editor and a Senior Editor. The reviewers have opted to remain anonymous.

Our decision has been reached after a process of initial independent review followed by extensive discussion between the reviewers and consultation with the editors. The initial independent reviews are reprinted at the bottom of this email, and the summary of the reviewers' discussion is provided in the next paragraph. Based on the overall consensus emerging from this discussion, we regret to inform you that your work will not be considered further for publication in *eLife*.

To summarize the reviewers' discussion: Both reviewers appreciate the improvements made during the revision of this manuscript, including the changes to the text and the addition of new data. However, both reviewers ultimately concluded that the revised material did not provide sufficient evidence in support of the primary conclusions of the manuscript. While much of the data in the paper is robust, the in vivo interaction of PlexinD1 and GIPC has not been robustly demonstrated via analysis of the PlexinD1 skt6 mutant. Specifically, reviewers were dissatisfied with the degree of statistical significance present in the data that was meant to demonstrate the effect of the skt6 mutation on angiogenesis, as well as in the data that was meant to demonstrate the rescue of the null allele by skt6. The lack of robust statistical significance here throws the conclusion that GIPCs are direct, negative modulators of PlexinD1 signaling into question. Among other concerns (see reviews below), reviewers were uncertain about the conclusions reached from the SU5416 treatments: they wondered whether the Bonferroni type adjustment is appropriate in this case and whether developmental delay could confound the interpretation of these data. Reviewers agreed that the strongest elements of the manuscript are the GIPC mutant data, both alone and when bred to PlexinD1 mutants. Also, the in vitro experiments looking at pERK after Sema3e treatment in GIPC/ PlexinD1 knockdown/knockouts are convincing. However, these experiments show association and not binding/mechanism. While there is clearly an association of PlexinD1 and GIPC and pERK, it remains unclear whether this interaction requires direct binding.

Reviewer #1:

The authors claim that the plxnd1 skt6 allele encodes a hypomorphic Plxnd1 receptor which induces angiogenesis deficits in homozygous mutant embryos.

However in the Figure 2 legend they clearly state that there is no statistical difference in phenotype occurrence between Wt and mutant embryos.

In the Results section, the authors claim that there is no statistical difference in Se phenotypes in Wt and plxnd1 skt6 siblings (without clarifying whether this denotes heterozygous or homozygous mutant embryos).

If the observed effect is not statistically significant, it cannot be used to claim a angiogenesis defect with low penetrance.

Additionally I disapprove of the quantification, as the authors should have analyzed always the exact same region and number of Se-DLAVs per embryo and not a varying average of 10.5 per embryo.

For me it is not clearly stated whether the less than 5% of affected Se-DLAVs are of all counted Se-DLAVs or are number of embryos with affected Se-DLAVs. I assume the first, as it would be difficult to get to 5% of 12 embryos. However it would be important to know how many of the embryos do show vessel truncations in one or two vessels.

The total number of only 12 embryos analyzed for each genotype is also really low.

The experiments addressing hypersensitivity to Su5416 have theoretically been done under the same conditions as when assessing the phenotype of homozygous *plxnd1^skt6^*mutants.

Why do the authors suddenly see dramatic difference in angiogenesis defects now in 40% of the vessels in 0.1% DMSO? Why does the DMSO also affect WT development? How are changes to more than 80% affected vessels only achieving one star significance? another explanation would be that the embryos have not really reached 32hpf but are a little developmentally delayed. The experimental procedure does not state whether and how the embryos were dechorionated for treatment from 18hpf on.

SU5416 has been previously used in zebrafish at concentrations of 1 µM and was shown to completely block Vegf mediated ISV sprouting. Here the authors use 0.2µM and show an effect on Wt ISV migration, which is slightly enhanced in the mutants, but there is no evidence that this does not just reflect a developmental delay caused by the chemical treatment. The authors do need to control for this alternative hypothesis. Also the embryo in Figure 3B seems to be "larger" or at higher magnification than the others, as the distance from the PCV to the top of the DLAV is larger than in the other three images. also in the supplement it is evident that already 90% of the Wt embryos are affected by the SU5416 treatment. The experimental conditions should have been chosen so that the Wt is not affected at all.

In sum for essential revision1B we requested to show a statistically significant effect on angiogenesis. Unfortunately, the data presented are still not convincing.

Likewise, why is there such a big variance in phenotype distribution when scoring gipc1 skt1(MZ) mutant embryos? Compare Figure 4B with Figure 5C.

It would also be helpful if the authors would indicate e.g. in Figure 2 or 3 and clearly define in the text what constitutes a maximal, a moderate or a minimal Se-truncation.

The authors claim that Gipc binding limits the antiangiogenic signaling of the PlxnD1 receptor. However, there are too many assumptions and too little data for such a statement.

For one: what is the antiangiogenic Plxnd1 signaling: where is that shown just based on a poorly controlled hypersensitivity to VegfR2 receptor inhibition, which was done for a very long time period? or based on the proposed hypomorphic identity of the plxnd1 skt6 allele, which as mentioned above is not clearly a significant phenotype.

The analysis of the gipc mutants is worthwhile, however there is too little evidence supporting the Gipc/Plexin interaction on a phenotypic level, with regards to their collaborating role in angiogenesis.

An alternative explanation for the partial reduction in phenotype would be that while in gipc1 skt1 Mz embryos gipc2 and gipc3 can still bind to Wt Plexin and therefore are less available for interaction with another receptor (like Nrp1) in the plxnd1 fov01b heterozygous less plexin molecules are available to limit the gipc signaling pool. This would also explain the reduction in phenotype, and be based on a Plexin/gipc interaction, but say nothing about the actual signaling cascades required for generating an angiogenesis phenotype (see our comment regarding essential revision 4B).

As the interaction studies between Gipc1/2 and plxnd1 are only analyzed qualitatively, meaning that you find massive ectopic sprouting, how were you expecting to see a double phenotype in case of no interaction? the gipc1skt1(Mz); gipc2 skt3/skt4 double phenotype did affect less than 20% of the embryos, only about 10% were affected medium to severely. that would mean that in case of NO interaction you would have found 10% of the Se's being truncated, but still having massive ectopic filopodia and sprouts. Given that there would be additional ectopic sprouts coming from the neighboring Se's, I doubt that this would have been detectable. I therefore believe that the experiment is not suitable to disprove alternative hypothesis…

In sum there is not enough evidence supporting a model whereby Gipc limits Plexind1 antiangiogenic signaling, as Gipc could still act on angiogenesis in a Plexin independent manner, eg via neuropilin (see essential revisions 6!). I don't think that the authors have sufficiently experimentally addressed this question. However, this affects the core of the authors claims by directly affecting the functionality of the Plexin/Gipc interactions in angiogenesis.

Reviewer #2:

The authors have cleared up misconceptions created by language and elegantly used hypotheses to set up outcomes of their experiments and what this means. This is a large improvement over the previous version of the manuscript.

The new skt6 allele of plexinD1 was used in very clear experiments to show it is functional and hypermorphic. There is also a plexinD1 null HUVEC line generated as well for the cell culture experiments.

Graphs have been improved. Numbers increased and emphasized.

Improved discussion of Burk results in context of the results of this manuscript.

Overall, there is a huge improvement to the manuscript. Many figures are edited or completely new and there is new data to answer my major points.

[Editors’ note: what now follows is the decision letter after the authors submitted for further consideration.]

Thank you for submitting your rebuttal letter regarding the manuscript entitled "GIPC proteins pattern the vasculature by limiting the antiangiogenic signaling of the Semaphorin receptor Plexind1" for consideration by *eLife*. I have read your rebuttal letter and conferred with the Reviewing Editor. We both agree that it would be appropriate for you to submit a revised version of the manuscript that addresses the comments of the reviewers.

We would like to clarify that both reviewer 1 and 2 were in good agreement during the second round of review. Therefore, we urge you to address their comments to the best of your ability. However, given your concerns we will not return the revised manuscripts to either reviewer, but will instead ask previous reviewer 3 (who did not respond to our previous request to re-review), plus a new reviewer. Of course we cannot guarantee the outcome but will do all that we can to be fair.

[Editors' note: further revisions were requested prior to acceptance, as described below.]

Thank you for resubmitting your work entitled "GIPC proteins pattern the vasculature by limiting the antiangiogenic signaling of the Semaphorin receptor Plexind1" for further consideration at *eLife*. Your revised article has been favorably evaluated by Marianne Bronner (Senior Editor), a Reviewing Editor, and two reviewers. One of these reviewers (Reviewer #3) had reviewed the original version of this manuscript, and the other (Reviewer #4) was a newly recruited reviewer.

As you can see from the reviewers' comments (copied below), the reviewers feel that the manuscript has been improved, but both agree that there are some remaining issues that need to be addressed before acceptance. The reviewers have discussed these issues with each other after submitting their reviews, and they agree that it is particularly important for the manuscript to be thoroughly edited, with the goal of creating a more succinct and streamlined experience for the reader. The reviewers also agree that you should make a series of specific adjustments to the text and figures in an effort to clarify or modify the message, in accordance with each of the individual points that have been raised in the reviews below. Finally, regarding reviewer #4's suggestion that you perform an additional co-IP experiment (see comment #1 from reviewer #4 below), the reviewers agree that this is not essential. If you have the data at hand, please do include it, but it will also be acceptable to simply make changes in the text describing the biochemical effects of this mutation. In that case, the idea that this mutant version of the zebrafish protein loses its ability to interact with Gipc proteins should be presented as speculation, not as fact.

Reviewer #3:

After carefully reading the full history of this manuscript, I tend to agree with the previous assessment of reviewer #2.

I believe the revised version provided by the authors successfully addressed the essential concerns raised during the first round of review. Moreover, I understand that the data added in the revised version was approved in advance in the context of the revision plan.

While I agree with reviewer #1, that some issues of statistical significance were not fully solved, the authors did a good job not only in providing extensive explanations for the observed results, but also by listing alternative hypotheses. In my view, the faithful presentation of the results, and the efforts made to explain them, are valuable, despite of the phenotypes themselves being not hugely penetrant.

As agreed by all 3 reviewers, the findings of this manuscript are important and add to our understanding of the mechanisms underlying Plexind1 signalling. Since most of the initial concerns were addressed properly, and the findings are carefully interpreted, I tend to support publication.

I do think however that the title/impact statement/conclusions could be toned down a bit to reflect more accurately the main findings.

Reviewer #4:

Previous work by Dr. Torres-Vazquez demonstrated a requirement for Semaphorin3E/PlexinD1 signaling in constraining segmental vessel sprouting and migration in zebrafish embryos, and two papers (Burk et al., 2017; Shang et al., 2017) more recently demonstrated an interaction between the C-terminus of PlexinD1 and the GIPC family of endocytic adaptor proteins. The goal here is to determine how GIPC proteins modulate PlexinD1 function in endothelial cells (ECs). Strengths of this work include the use of both cell culture and zebrafish models; the generation of multiple new zebrafish mutants; genetic interaction studies in both models; and clear data presentation. The authors conclude that GIPC functions as a negative regulator of PlexinD1, serving to dampen its antiangiogenic activity. In aggregate, the presented data strongly support this conclusion, which contrasts with findings in Burk et al. However, a major problem is that an assumption is made regarding the biochemical consequences of a key plxnd1 mutant without direct presentation of experimental data to support this assumption.

1) The CRISPR/Cas9-generated *plxnd1^skt6^* mutant harbors a deletion that removes the final 5 C-terminal amino acids and extends the protein with 31 amino acids. The authors assume that this mutant will abrogate interaction with Gipc proteins because 1) deletion of these 5 amino acids in human PLXND1 decreases interaction with human GIPC1 (as shown in Figure 1); and 2) addition of a single amino acid to the C-terminus of other PDZ-binding proteins abrogates their interaction with PDZ proteins (cite published work). However, deletion of these 5 amino acids in the human protein decreases but does not abolish PLXND1/GIPC1 interaction (Figure 1), and the authors do not test whether the extended Plxnd1 protein generated in these zebrafish mutants does indeed fail to bind Gipc proteins. As the conclusions are based entirely on this important assumption and this assumption is repeatedly stated as fact (for example, subsection “Homozygous *plxnd1^skt6^* mutants are hypersensitive to the antiangiogenic drug SU5416”; Figure 2 title), this should be tested with at least one recombinant Gipc protein as in Figure 1.

2) In Figure 2—figure supplement 2, the authors show that mosaic EC-specific expression of wild type or C-terminally truncated PlexinD1 (missing the 9 C-terminal amino acids that comprise the entire GIPC binding motif) can similarly rescue segmental vessel development in *plxnd1^fov01b^* null mutants. They interpret this to mean that Gipc1 binding is not required for antiangiogenic activity of Plxnd1 (truncated version is fully functional) and use these data to support their claim that the *plxnd1^skt6^* allele is hypermorphic due to inhibited interaction with Gipc1. However, the 9 amino acid deletion is not truly comparable to the 5 amino acid deletion/31 amino acid insertion in the mutant. If the authors can show that the skt6 protein is deficient in Gipc1 binding (Comment #1), this would minimize this concern.

---

## [Author Response]

Essential revisions:1A) This manuscript nicely uses an in vivo assay (Figure 2) to test whether GIPC-binding deficient mutants of PlxnD1 lose activity for vessel patterning. Surprisingly, both wild type PlxnD1 and PlxnD1ΔGBM are equally effective at restoring vessel patterning in PlxnD1 mutants. The authors interpret this to suggest that GIPCs are not required for PlxnD1 activity. The authors then suggest that either 'PlxnD1ΔGBM will be more potent than PlxnD1WT' or that 'both PlxnD1's activity and potency will be independent of GIPC-binding'. These two hypotheses are poorly worded as it is not clear what 'potency' and 'activity' refer to. By potency, do the authors mean that PlxnD1ΔGBM is insensitive to inhibition and therefore more active? What phenotype would a 'more active' PlxnD1 mutant have? For the second hypothesis, do the authors mean that while GIPC binds PlxnD1, it has no influence on activity?

We revised the text to highlight that PlxnD1 is antiangiogenic, avoiding the use of the terms “activity” and “potency.” As agreed, we thus instead speak of “Plxnd1 signaling”. Given the proposed plan for addressing essential revision 1B (below), we stated how each of the three competing hypotheses makes different forecasts regarding the nature of the *plxnd1^skt6^*allele (predicted to be deficient in GIPC-binding and made via genome editing). These hypotheses are assessed by the ability of the *plxnd1^skt6^* allele to complement the null *plxnd1^fov01b^* mutation and by the vascular phenotype and sensitivity to the antiangiogenic compound SU5416 of *plxnd1^skt6^* homozygotes.

In the revised manuscript, this information is found in the Results’ section entitled “Zebrafish mutants in which plxnd1 function is provided solely by a receptor deficient in GIPC binding display angiogenesis deficits with low penetrance and are hypersensitive to the antiangiogenic drug SU5416,” and in Figures 2-3, Figure 2—figure supplement 1 and Figure 3—figure supplement 1. Finally, the *plxnd1^skt6^* allele designation follows ZFIN-approved nomenclature.

1B) Also, and importantly, when comparing the influence of PlxnD1-WT and PlxnD1ΔGBM on "angiogenic potential", the authors speculate on the meaning of an observed trend that is not statistically significant. Could the authors increase the sample size for these observations? Would they then find evidence for a statistically significant difference?

Given that increasing the sample size for the transgenic rescue experiment was unlikely to resolve the issue, we did not pursue this direction. Briefly, few cells express the exogenous receptor. Hence, the assay is too noisy which makes the evaluation of differences in angiogenic potential an impractical proposition.

Instead, as proposed before, we characterized the novel zebrafish *plxnd1^skt6^*mutant allele that we made via genome editing (formerly referred as plxnd1ΔGBM allele in our prior reply). This allele is predicted to yield a Plxnd1 protein in which a scrambled amino-acid sequence replaces the PBM. Using this allele, we performed the following three experiments.

1) Performed complementation assays between our novel *plxnd1^skt6^* allele and the null mutant *plxnd1^fov01b^* to define the vascular phenotypes of the trans-heterozygotes (*plxnd1^skt6^* complements *plxnd1^fov01b^*).

2) Determined the phenotype of *plxnd1^skt6^* homozygotes (they exhibit angiogenesis deficits at low frequency).

3) To further characterize both the functionality and nature of the *plxnd1^skt6^* allele we decided, for simplicity, to compare the sensitivity of WT embryos and *plxnd1^skt6^* mutants to the antiangiogenic compound SU5416, rather than performing the optional cell transplantation experiment outlined previously (*plxnd1^skt6^* mutants are hypersensitive to SU5416).

These observations led us to conclude that the receptor deficient in GIPC binding encoded by the *plxnd1^skt6^* allele is both functional and hypermorphic. In the revised manuscript, these findings and conclusions are described in the Results’ section entitled “Zebrafish mutants in which plxnd1 function is provided solely by a receptor deficient in GIPC binding display angiogenesis deficits with low penetrance and are hypersensitive to the antiangiogenic drug SU5416” and in Figures 2-3, Figure 2—figure supplement 1 and Figure 3—figure supplement 1.

1C) Finally, the conclusion that PlxnD1 signaling acts to limit endothelial proliferation should be confirmed through direct analysis of proliferation (e.g. by EdU labeling).

This comment relates to the transgenic rescue experiment now found in Figure 2—figure supplement 2, and directly related to the graph in panel K (now inexistent). We believe that the reviewer’s comment arose from the potential ambiguity of the original text, which could be read as overstating one of our conclusions. Since prior studies already established a connection between Plxnd1 signaling and proliferation, we indicated in our prior reply that the EdU labeling experiment was unnecessary. More importantly, the relevant panel K is no longer part of the revised Figure 2—figure supplement 2, since the *plxnd1^skt6^* data supports better the notion that Plxnd1 proteins deficient in GIPC binding are hypermorphic. Accordingly, we revised the Results’ section entitled “Zebrafish mutants in which plxnd1 function is provided solely by a receptor deficient in GIPC binding display angiogenesis deficits with low penetrance and are hypersensitive to the antiangiogenic drug SU5416” and the legend of Figure 2—figure supplement 2.

2A) The intersegmental vessel truncation observed in zygotic gipc1 or gipc2 single mutants is not statistically significant. Therefore, rather than illustrating these so prominently in Figure 3, it might be more worthwhile to focus on the significant effects observed in double mutants and MZ mutants, which show a phenotype that has low penetrance. The compression of the graph's X-axis in Figure 3B to emphasize this phenotype is unusual and could be misleading. At least 80% of the embryo SeV are normal, and the way that the data are presented may give the impression that the phenotype is stronger than it actually is.

We modified Figure 3A (now Figure 4A) to show only the images of the vascular phenotypes of the four recommended genotypes. Accordingly, we presented the phenotypes of *gipc1* and *gipc2* single zygotic mutants as Supplementary Data (Figure 4—figure supplement 2).

As recommended for Figure 3B (now Figure 4), we adjusted the X-axis to 100% and showed the “No truncation” phenotypic category as a black bar. We made similar modifications to other graphs related to vascular phenotypes (see Figures 2I, 3E, 4B, 5, Figure 2—figure supplement 1, Figure 3—figure supplement 1, Figure 4—figure supplement 3 and Figure 5—figure supplement 1). We further improved all phenotypic graphs by displaying significant differences using the standard brackets and asterisks (rather than with text; see Figures 3E, 4B, 5C, Figure 3—figure supplement 1, Figure 4—figure supplement 3, and Figure 5—figure supplement 1B).

We note that, for simplicity and readability, the graphs of vascular phenotypes show one kind of significant difference. Supplementary Files 3-6 show the remaining significant differences.

2B) It would also be helpful to provide statistical analysis in Figure S2 and to adjust the X-axis in Figure S2B.

We calculated significant differences and adjusted the X-axis for both panels in now Figure 4—figure supplement 3. We applied similar changes to Figures 3E, 4B, 5C, Figure 3—figure supplement 1, Figure 4—figure supplement 3 and Figure 5—figure supplement 1B. We also now show the “No truncation” phenotypic category as a black bar in Figure 4—figure supplement 3B. We applied similar changes to Figures 2I, 3E, 4B, 5, Figure 2—figure supplement 1, Figure 3—figure supplement 1, Figure 4—figure supplement 3 and Figure 5—figure supplement 1.

2C) Since these results are key to the authors' conclusions, significance and robustness need to be clearly shown, particularly as the results in this manuscript do not agree with those in a published article on the same topic in mouse (Burk et al., 2017) It should be straightforward (in principle) for the authors to increase their n and procure a more robust result.

We agree with the reviewers that it is important to highlight the solid significance and robustness of our results, which are based on the analysis of one hundred twelve embryos (sixty-nine embryos without maternal and zygotic *gipc1* activity, including nineteen animals devoid of zygotic *gipc2* activity; plus twenty-four embryos lacking zygotic *gipc1* activity, including thirteen without *gipc2* activity). Hence and as stated before, the analysis of more embryos is unwarranted. We highlight that we found that *gipc* mutants display recessive arterial angiogenic deficits of partial penetrance. Moreover, none of the *gipc* mutants analyzed phenocopied, even partially, the hyperangiogenic and mispatterned vasculature of *plxnd1^fov01b^* nulls.

Accordingly, this information is mentioned in detail in the revised manuscript as follows. First, in the Results’ sections entitled “Zebrafish *gipc* mutants display angiogenesis deficits in the trunk’s arterial tree,” “Reducing Plxnd1 signaling ameliorates the angiogenic deficits of maternal-zygotic gipc1 mutants”, and “Double maternal-zygotic (MZ) *gipc* mutants display excessive angiogenesis in the absence of Plxnd1 signaling.” Second, in the corresponding figures: Figures 4-6, Figure 4—figure supplement 2, Figure 4—figure supplement 3 and Figure 5—figure supplement 1). Finally, in Supplementary Files 5-6. We note that the incorporation of essential revisions 2A-2B (now Figure 4—figure supplement 3; above) further highlights the significant differences in penetrance and expressivity between gipc mutants.

3A) Even given its low penetrance, the gipc mutant phenotype is interesting as it is very different from the mouse GIPC knockout vascular phenotype that shows a PlxnD1-like phenotype in equivalent vessels (as opposed to fewer vessels in fish gipc mutants). Burk et al., 2017, claim (in their Figure 10D) that, in mouse embryos, loss of Gipc1 leads to ectopic ISV sprouting, the same phenotype observed in PlexinD1 mutants (not shown) and in Gipc1;PlexinD1 double heterozygote embryos. Although this is illustrated only briefly and not very convincingly by Burk et al., this published result seems to be in strong contrast to the gipc mutant phenotypes illustrated in Figure 3 of this manuscript. Why are fish and mouse null Gipc mutants different? An insightful discussion is needed, and the authors do not provide much material in this regard.

We agree with the reviewers. Accordingly, we provide further details in two sections of the revision. First, we do this briefly in the last paragraph of the Results’ section entitled “Zebrafish *gipc* mutants display angiogenesis deficits in the trunk’s arterial tree.” Second, we address in great depth the potential reasons behind the discrepancy in the seventh, eighth and ninth paragraphs of the Discussion section. Briefly, we do not favor the possibility that *gipc* genes play distinct roles in the zebrafish and mice. Instead, we believe that their functions are conserved and list seven points that invite a reinterpretation of the findings by Burk et al. (see also our answer to essential revision 4C).

3B) It would also be helpful if the previous name of Gipc1 (Synectin) were to be mentioned in this manuscript.

The original text highlighted that GIPC1 and Synectin are synonymous. Now for emphasis, we repeatedly mention this equivalency in the revision. Specifically, Synectin is now mentioned six times throughout the main body of the manuscript.

4A) The claim of Figure 4, that the angiogenesis defects of gipc mutants are partially due to increased PlxnD1 signaling, is based on the analysis of zygotic gipc1;plxnD1 double mutants. However, as gipc1 zygotic mutants had no statistically significant Se defect, this experiment might not be suitable for supporting the authors' claim. Notably, the double mutants are analyzed only qualitatively here, and no quantitation is provided.

In the revised manuscript we addressed this point via the following two experiments. First, and using a quantitative phenotypic analysis, we determined that *plxnd1* heterozygosity ameliorates the angiogenesis deficits of *gipc1^skt1(MZ)^*maternal-zygotic mutants. We describe these findings in the Result’s section entitled “Reducing Plxnd1 signaling ameliorates the angiogenic deficits of maternal-zygotic gipc1 mutants”, Figure 5 and Figure 5—figure supplement 1 (see also Supplementary File 6).

Second, we compared the vascular phenotype of *gipc1^skt1(MZ^); gipc2^skt4(MZ)^*maternal-zygotic double mutants with and without morpholino-induced *plxnd1* knockdown at the 32 hpf stage. Importantly, this morpholino faithfully phenocopies the vascular phenotype of *plxnd1^fov01b^* null mutants, namely Se vessel over-sprouting and disorganization (see reference #5 in the revision). Since it is impossible to score the abundance and length of Se vessels at 32 hpf, we evaluated the vascular abnormalities qualitatively. We found that *gipc1^skt1(MZ^); gipc2^skt4(MZ)^*maternal-zygotic double mutants injected with the *plxnd1* morpholino phenocopy the excessive and disorganized angiogenesis of *plxnd1* nulls. We describe these findings in the Results’ section entitled “Double maternal-zygotic (MZ) *gipc* mutants display excessive angiogenesis in the absence of *Plxnd1* signaling” and in Figure 6.

4B) Importantly, if GIPC acts as a repressor of PlxnD1 activity, one would expect a rescue in the double mutant. Therefore, it is possible to argue that the authors have shown that, even though GIPC binds PlxnD1, its activity is not required to influence PlxnD1 signaling. However, the authors do not reach this conclusion. Perhaps PlxnD1 is epistatic to GIPC (i.e. acts downstream of GIPC). Even though GIPC binds PlxnD1, it could act upstream and/or in a different pathway to control angiogenesis of the SeV (such as in the neuropilin pathway or other pathways where GIPC is known to act).

To clarify, the model that GIPCs simultaneously promotes pro-angiogenic VEGF signaling and limits anti-angiogenic PLXND1 signaling predicts that the double *gipc* mutant/*plxnd1* morphant would have a *plxnd*1 null phenotype. The results of the revised experiment above agree with this prediction, supporting this model. However, if for example, the double *gipc* mutant/*plxnd1* morphant had a phenotype like that of *gipc* mutants, the model would be invalid. None of these potential results argues against the possibility that GIPC also acts in a PLXND1-independent manner. Please see also our reply to essential revision 4A.

4C) Finally, the fact that GIPC and PlxnD1 mouse double mutants (Burk et al.), have identical and not opposite phenotypes, unlike in the fish, is not discussed in any depth.

We rectified this omission in the revised manuscript. Please see our answer to essential revision 3A for further details.

5A) The data presented in Figure 6 clearly demonstrate that loss of GIPC potentiates loss of Sema3E-induced collapse. The authors conclude that this is PlxnD1-dependent because siRNA against PlxnD1 fails to induce the collapse. However, double knockdown of GIPC and PlxnD1, followed by Sema3E addition, is necessary to make this conclusion, and this experiment was not performed. Without double knockdown data, there is no evidence that PlxnD1, GIPC and Sema3E interact in collapse, as this not been tested in the same cells. The same argument can be made for the ERK data – again, PlxnD1-GIPC double knockdown, with and without Sema3E, in the same cells is a necessary experiment to support the conclusion.

As suggested and previously agreed to, we performed one combined *GIPC/PLXND1* loss of function experiment (using pERK abundance as a readout of PLXND1 activity; see Figure 7 and Figure 7—figure supplements 1-2 in the revision). Importantly, the results of this experiment indicate that loss of GIPC potentiates the SEMA3E-induced/PLXND1-dependent decrease in pERK abundance. The cell collapse data presented in the original Figure 6 is now found in Figure 7—figure supplement 3 (see also Figure 7—figure supplement 4). We describe the pERK and cell collapse experiments in the revised Results’ section entitled “GIPC depletion potentiates SEMA3E-induced, PLXND1-dependent responses in HUVECs”.

5B) Alternatively, could the authors corroborate the ERK activation results (Figure 6) in zebrafish? Since all the zebrafish mutants are available, the use of HUVECs seems less relevant. ERK antibody staining has been extensively described in zebrafish, and it could be used to investigate ERK activation in the arteries of gipc mutants, as well as in gipc;plexind1 double mutants. This type of analysis has the potential to strengthen the authors' claims.

We did not perform the suggested experiment with fish since we believe that the cell culture model is the best option to answer this question. Briefly, prior studies indicate that VEGF signaling via VEGFR2 and Nrp1 up-regulates pERK levels and that, in this context, GIPC1 is required for maximal VEGF-dependent pERK up-regulation. Also, PLXND1 signaling antagonizes VEGF signaling in part by reducing pERK levels. Importantly, in cell culture, we can activate PLXND1 signaling in the absence of exogenous VEGF stimulation, allowing us to use the relative abundance of pERK as a quantifiable PLXND1-specific signaling output. In contrast, in the embryo, VEGF signaling must be active to enable both endothelial ERK phosphorylation and Se angiogenesis. Moreover, endothelial pERK immunofluorescence is variable within and across WT embryos in our hands. Accordingly, we experimented with using HUVECs. We describe the pERK experiment in the revised Results’ section entitled “GIPC depletion potentiates SEMA3E-induced, PLXND1-dependent responses in HUVECs” and in Figure 7 and Figure 7—figure supplements 1-2. Importantly, the results corroborate the proposed model. Please see our answer to essential revision 5A.

6A) The authors seem to assume that GIPCs can mediate PlxnD1 activity only (or primarily) via direct binding through the identified domains. Although this was the way that the authors initially identified GIPCs (via 2-hybrid screening), can they truly rule out other possibilities?

Of course not. Our results do not rule out the possibility that GIPCs might also influence PLXND1 signaling independently of the formation of GIPC-PLXND1 complexes and we now highlight this scenario in the eleventh paragraph of the revised Discussion section.

See also the fourth paragraph in the Results’ section entitled “Isolation of GIPC1/Synectin as a PLXND1-binding protein and dissection of the molecular determinants required for PLXND1-GIPC complex formation” and remaining of the revised Discussion section.

6B) Additionally, the authors do not seem to consider other potential roles of GIPCs that are PlxnD1-independent. Although they state in their Results section that "the angiogenesis deficits of gipc mutants are, at least partially, due to increased PlxnD1 signaling", they do not assess or discuss additional possibilities.

We revised both the Introduction (penultimate paragraph) and the Discussion (tenth and eleventh paragraphs) sections to provide further details about the relationship between GIPC1 and NRP1-mediated VEGF signaling and to highlight that GIPC proteins have multiple partners, including additional molecules involved in the modulation of vascular development. See also our reply to essential revision 6A.

6C) One relatively easy experiment that the authors could perform in order to address these questions would be to overexpress plexind1 (WT and truncated forms) in endothelial cells of gipc single and double mutants. Alternatively, the authors could consider the possible relevance of the known interactions of GIPC with Neuropilin. While Neuropilin can act as a Plexin coreceptor, it can also act in a Plexin-independent manner. This is of relevance especially in experiments analyzing growth cone collapses or other effects on morphology, where Neuropilin has been shown to act independently of Plexin. Can the authors distinguish the negative regulation of Plexin from a potential positive regulation of Neuropilin? This is of course hampered by the caveat that there are no reliable Neuropilin reagents in zebrafish available as morpholino and mutant data only partially match. Maybe different downstream effectors could be affected pharmacologically? At a minimum, it would be helpful for the authors to provide more discussion of alternate possibilities that could explain the observed ways in which GIPC function affects PlxnD1 activity.

It was unfeasible for us to do the experiments mentioned above (we lacked the required stocks/reagents and building them would take a few zebrafish generations). Hence, we instead strengthened the Discussion (see tenth and eleventh paragraphs) to specify that in all likelihood, the proangiogenic function of GIPCs is multifaceted and involves additional mechanisms besides the negative modulation of PlxnD1 signaling. See also our reply to essential revisions 6A and 6B.

However, we highlight that the angiogenesis deficits and SU5416 hypersensitivity of *plxnd1^skt6^* mutants support the model that GIPCs directly influence PLXND1 signaling. This new data is found in the Results’ section of the revision entitled “Zebrafish mutants in which *plxnd1* function is provided solely by a receptor deficient in GIPC binding display angiogenesis deficits with low penetrance and are hypersensitive to the antiangiogenic drug SU5416” (see Figures 2-3). Further support for this model comes from the experiments described in the Results’ section entitled “GIPC depletion potentiates SEMA3E-induced, PLXND1-dependent responses in HUVECs” (see Figure 7).

[Editors' note: the authors’ plan for revisions was approved and the authors made a formal revised submission.]

Thank you for submitting your revised manuscript entitled "GIPC proteins pattern the vasculature by limiting the antiangiogenic signaling of the Semaphorin receptor Plexind1" for consideration by eLife. Your revised submission has been reviewed by two out of the three original peer reviewers, and the evaluation has been overseen by a Reviewing Editor and a Senior Editor. The reviewers have opted to remain anonymous.Our decision has been reached after a process of initial independent review followed by extensive discussion between the reviewers and consultation with the editors. The initial independent reviews are reprinted at the bottom of this email, and the summary of the reviewers' discussion is provided in the next paragraph. Based on the overall consensus emerging from this discussion, we regret to inform you that your work will not be considered further for publication in eLife.To summarize the reviewers' discussion: Both reviewers appreciate the improvements made during the revision of this manuscript, including the changes to the text and the addition of new data. However, both reviewers ultimately concluded that the revised material did not provide sufficient evidence in support of the primary conclusions of the manuscript. While much of the data in the paper is robust, the in vivo interaction of PlexinD1 and GIPC has not been robustly demonstrated via analysis of the PlexinD1 skt6 mutant. Specifically, reviewers were dissatisfied with the degree of statistical significance present in the data that was meant to demonstrate the effect of the skt6 mutation on angiogenesis, as well as in the data that was meant to demonstrate the rescue of the null allele by skt6. The lack of robust statistical significance here throws the conclusion that GIPCs are direct, negative modulators of PlexinD1 signaling into question. Among other concerns (see reviews below), reviewers were uncertain about the conclusions reached from the SU5416 treatments: they wondered whether the Bonferroni type adjustment is appropriate in this case and whether developmental delay could confound the interpretation of these data. Reviewers agreed that the strongest elements of the manuscript are the GIPC mutant data, both alone and when bred to PlexinD1 mutants. Also, the in vitro experiments looking at pERK after Sema3e treatment in GIPC/ PlexinD1 knockdown/knockouts are convincing. However, these experiments show association and not binding/mechanism. While there is clearly an association of PlexinD1 and GIPC and pERK, it remains unclear whether this interaction requires direct binding.Reviewer #1:The authors claim that the plxnd1 skt6 allele encodes a hypomorphic Plxnd1 receptor which induces angiogenesis deficits in homozygous mutant embryos.However in the Figure 2 legend they clearly state that there is no statistical difference in phenotype occurrence between Wt and mutant embryos.In the Results section, the authors claim that there is no statistical difference in Se phenotypes in Wt and plxnd1 skt6 siblings (without clarifying whether this denotes heterozygous or homozygous mutant embryos).If the observed effect is not statistically significant, it cannot be used to claim a angiogenesis defect with low penetrance.

Thank you for the time and effort that you have dedicated to the review of our manuscript.

1.1) As stated in the text, the frequency, penetrance and expressivity (Figure 2I, Figure 2—figure supplement 1) of the defects of *plxnd1^skt6^* mutants are low and not statistically significantly different from WT. We can edit the text to clarify further how these observations fit with the proposed nature of the allele. By scoring the progeny derived from an incross of *plxnd1^skt6/+^*heterozygotes, we found that only the *plxnd1^skt6^*homozygous mutants, but not their homozygous WT siblings, had “angiogenesis deficits with low penetrance”. This statement is neither self-contradictory nor misleading since the definition of penetrance (percentage of individuals with a given genotype displaying a phenotype associated with that genotype) is independent of the concept of statistical significance. Finally, we state: “…the unique presence of angiogenic deficits in the mutants…is consistent with the notion that the *plxnd1^skt6^*allele is hypermorphic”. However, to add more clarity and nuance, we propose to revise this text to read instead: “…the unique presence of angiogenic deficits in the mutants, such as maximal and moderate Se and DLAV truncations, suggests that the *plxnd1^skt6^*allele could be hypermorphic”.

1.2) To clarify, we claim that the *plxnd1^skt6^*behaves like a hypermorph (gain of function; but see typo-1). The bases for this assessment is the data on Figures 2-3. For example: “…the hypothesis that GIPCs limit PLXND1’s antiangiogenic signaling foretells that *plxnd1^skt6^*should be a *plxnd1^fov01b^* complementing hypermorph and that its homozygous mutants will display angiogenesis deficits and hypersensitivity to antiangiogenic compounds”. Hence, the definitive characterization of the *plxnd1^skt6^*allele as hypermorphic is based on the hypersensitivity of the homozygous mutants to SU5416. Thus, we state: “…the angiogenesis deficits of *plxnd1^skt6^*homozygotes and their hypersensitivity to the antiangiogenic drug SU5416 support the model that GIPC binding limits the antiangiogenic signaling of the PLXND1 receptor. In our interpretation, it is the loss of this inhibiting effect what confers to the *plxnd1^skt6^* allele its demonstrable hypermorphic character.” See also authors’ reply 6.4. For additional comments regarding the SU5416 hypersensitivity of *plxnd1^skt6^* mutants, see authors’ replies 3.1-3.5, 4.1-4.4, 6.2-6.4, 7.2 and 9.2.

Additionally I disapprove of the quantification, as the authors should have analyzed always the exact same region and number of Se-DLAVs per embryo and not a varying average of 10.5 per embryo.For me it is not clearly stated whether the less than 5% of affected Se-DLAVs are of all counted Se-DLAVs or are number of embryos with affected Se-DLAVs. I assume the first, as it would be difficult to get to 5% of 12 embryos. However it would be important to know how many of the embryos do show vessel truncations in one or two vessels.The total number of only 12 embryos analyzed for each genotype is also really low.

2.1) To clarify, we always analyzed the same anatomical region. In the Materials and methods section entitled “Quantification of angiogenesis deficits in the trunk’s arterial tree of WT and mutant zebrafish embryos” we say: “…Scoring of angiogenesis deficits was performed bilaterally in the ~six somite-long region dorsal to the yolk extension.” This method is used widely in the field. The small differences in the average number of Se-DLAVs analyzed (see Supplementary File 3, first table) are due to biological variability within and across genotypes. Briefly, embryos of the same stage are of similar, but not identical, size. The yolk extension’s length can vary slightly. Finally, the left and right sides of the embryo are in approximate register. Thus, our images capture 5 or 6 Se vessels per side (see the first table in Supplementary File 3).

2.2) To clarify, the quantification in Figure 2I reflects the percentage of affected Se-DLAVs of all counted Se-DLAVs. This information is provided in the legend of Figure 2.

2.3) To clarify, 3 out of 12 embryos show vessel truncations. This information is in the second table of Supplementary File 3 (as indicated in the legend of Figure 2).

2.4) The reviewer says that the number of embryos analyzed in Figure 2I is low. We analyzed 12 WT homozygotes and 12 *plxnd1^skt6^*homozygous mutants, or 124-126 Se-DLAVs/genotype (Supplementary File 3). It is unclear why the reviewer considers these numbers insufficient because the reviewer offers no justification for this comment. This comment is inconsistent with the reviewer’s praise (reviewer’s comment #7) of Figure 4B (based on the analysis of 11-33 embryos/genotype). The reviewer’s comment is also inconsistent with the editorially approved plan of our revision. Briefly, in essential revision 4A (related to a different experiment), we agreed to compare two sets of 10 embryos without objection. Finally, we are aware that increasing sample size would allow us to detect a smaller effect size. However, we thought it was more relevant to test our hypothesis further using the SU5416 assay because lowering VEGF signaling would be very illuminating, as outlined in the authors’ reply 7.2 (please see also authors’ replies 1.2, 4.2-4.4 and 6.4).

The experiments addressing hypersensitivity to Su5416 have theoretically been done under the same conditions as when assessing the phenotype of homozygous plxnd1skt6 mutants.Why do the authors suddenly see dramatic difference in angiogenesis defects now in 40% of the vessels in 0.1% DMSO? Why does the DMSO also affect WT development? How are changes to more than 80% affected vessels only achieving one star significance? another explanation would be that the embryos have not really reached 32hpf but are a little developmentally delayed. The experimental procedure does not state whether and how the embryos were dechorionated for treatment from 18hpf on.

3.1) To clarify, the data in Figures 2-3 was scored and analyzed identically (see legends and Materials and methods). We can add to the latter section the dechorionation procedure. In the legends of Figures 2-3 we indicate that the quantifications are for Se-DLAV angiogenesis. We can edit the Material and Methods to add that embryos were manually dechorionated before the treatments.

3.2) The reviewer incorrectly states that 40% of the vessels in DMSO-treated WT embryos show angiogenesis defects. Instead, the graph (Figure 3E; see also the first table of Supplementary File 4) shows that 15.3% of Se-DLAVs have an angiogenesis deficit. Most defects are minimal (13.1%), the rest are moderate (2.2%). The graph’s legend reads: “Percentage of Se-DLAV in 32 hpf embryos of the indicated genotype and treatment combinations belonging to each of the following four phenotypic classes”. The ~40% (35.7%) value that the reviewer cites is for the penetrance (percentage of embryos with Se-DLAV truncations).

3.3) The effects of DMSO are minimal (see above) and controlled for in our experiments. DMSO impacts embryonic development at high concentrations, as shown by zebrafish toxicology studies (see PMIDs 16125774, 21356178 and 23082109). Although zebrafish angiogenesis studies typically use only 0.1% DMSO as a drug vehicle (references #94 and #161), this dose can weakly impact WT vascular development, as our data shows. Thus, comparing the effects of DMSO and SU5416 (dissolved in DMSO) is an excellent experimental design feature of our study (and one that is absent from most other studies).

3.4) There is no evidence that DMSO or SU5416 induce developmental delay at the doses used. We can edit the text to highlight this information. Specifically, these treatments did not induce apparent effects on embryonic morphology and size, somite number, cardiac contractility and circulation (defects induced by high DMSO doses reported in toxicological studies). We also note that we used a suboptimal SU5416 dose. Together, these observations strongly suggest that SU5416 treatment induces specific effects, namely an increase in the frequency and severity of vascular defects (over those induced by DMSO only treatment). Moreover, the experiments with the two genotypes were run side by side using common dilutions. We can edit the Results to highlight these observations.

3.5) In Figure 3E the depiction of significant differences with one star is correct. As the legend states, the graph shows that “the distributions of the four phenotypic classes were statistically significantly different between all the possible pairwise comparisons of the four combinations of treatments and genotypes.” We also state that, the “…Significance values were calculated using two-sided Fisher Exact tests …”. In this statistical test, the level of significance (p-value) does not indicate the strength of the difference between groups. Thus, significant differences must be labeled with one star, regardless of the actual magnitude of the p-value (Laerd Statistics (2016). Fisher's exact test using SPSS Statistics. Statistical tutorials and software guides. Retrieved from https://statistics.laerd.com/). See Supplementary File 4 for the significance values.

SU5416 has been previously used in zebrafish at concentrations of 1 µM and was shown to completely block Vegf mediated ISV sprouting. Here the authors use 0.2µM and show an effect on Wt ISV migration, which is slightly enhanced in the mutants, but there is no evidence that this does not just reflect a developmental delay caused by the chemical treatment. The authors do need to control for this alternative hypothesis. Also the embryo in Figure 3B seems to be "larger" or at higher magnification than the others, as the distance from the PCV to the top of the DLAV is larger than in the other three images. also in the supplement it is evident that already 90% of the Wt embryos are affected by the SU5416 treatment. The experimental conditions should have been chosen so that the Wt is not affected at all.In sum for essential revision1B we requested to show a statistically significant effect on angiogenesis. Unfortunately, the data presented are still not convincing.

4.1) There is no evidence that DMSO or SU5416 induce developmental delay. Moreover, we controlled for the effects of DMSO and used a suboptimal SU5416 dose. See the authors’ replies 3.3-3.4, 6.3 and 9.2.

4.2) Using an SU5416 treatment that affects the WT is practical and valid. First, 0.1% DMSO is the recommended dose for dissolving SU5416 (which is a water-insoluble compound). Second, and as our data shows, 0.1% DMSO alone has a weak effect on angiogenesis in the WT. Hence, even an ineffectively low dose of SU5416 dose (dissolved in DMSO by necessity) would still affect the WT due to DMSO’s “background” effects. Third, we controlled for the effects of DMSO (see authors’ replies 3.3-3.4). Finally, it is valid to start with a condition affecting the WT. Briefly, our experimental design (Figure 3) follows the logic of enhancer and suppressor screens, where one tests the sensitivity to an insult (a drug or a mutation). One can start with genotypes that display phenotypes that are not significantly different (like WT and *plxnd1^skt6^*; see Figure 2I) and challenge with a second insult (like SU5416; see Figure 3). Alternatively, one can start with an abnormal phenotype (like that of SU5416-treated WTs) and ask if a second insult (*plxnd1^skt6^*homozygosity) modifies the phenotype (Figure 3). Either way, the conclusion is the same: SU5416-treated *plxnd1^skt6^*mutants have more severe angiogenesis deficits than SU5416-treated WTs.

4.3) Our data show a statistically significant effect on angiogenesis. SU5416-induced vascular defects are more frequent and severe in *plxnd1^skt6^*mutants than in WT embryos. SU5416 treatment induces defects in 89.7% of WTs and 100% of *plxnd1^skt6^*mutants (a non-significant difference in penetrance; see Figure 3—figure supplement 1A and Supplementary File 4). However, the critical point is that this 10.3% penetrance difference cannot account for the significant increase in the frequency and severity of angiogenesis deficits observed in SU5416-treated *plxnd1^skt6^*mutants (Figure 3E, Figure 3-supplement 1B and Supplementary File 4). To elaborate, the graph in Figure 3E (26-29 embryos analyzed) shows the “Percentage of Se-DLAV in 32 hpf embryos of the indicated genotype and treatment combinations belonging to each of the following four phenotypic classes…The distributions of these four phenotypic classes were statistically significantly different between all the possible pairwise comparisons of the four combinations of treatments and genotypes”. Figure 3—figure supplement 1 shows the penetrance (A) and expressivity (B) of Se-DLAV truncations for treated embryos. Note, for example, that for SU5416 treatment the penetrance of defects is not significantly different between WT and mutants, yet their expressivity is. Supplementary File 4 (second to third pages) shows eight tables of significance values (p) related to various comparisons of distributions, penetrance, and expressivity. Please note that in seven out of eight comparisons of significance values, the SU5416 treatment induces a significant difference between WT and *plxnd1^skt6^*mutants (with mutants being more affected). Again, the exception is for the penetrance of Se-DLAV truncations (first table on the third page) and, meaningfully, the expressivities are significantly different (second and third tables on the third page). Finally, we highlight the comment by reviewer #2 stating “the new skt6 allele of plexinD1 was used in very clear experiments to show it is functional and hypermorphic”.

4.4) The hypersensitivity of *plxnd1^skt6^*mutants to SU5416 supports the model that GIPCs limit Plxnd1 antiangiogenic signaling. As explained above, our data and statistical analysis show that DMSO and SU5416 treatments impair angiogenesis, with SU5416 inducing defects with both higher frequency and severity. Both DMSO and SU5416 have stronger effects on *plxnd1^skt6^*homozygotes. These mutants express only a Plxnd1 receptor that is deficient in GIPC binding. The hypothesis that GIPCs limit antiangiogenic signaling predicts that *plxnd1^skt6^*mutants should have potentiated antiangiogenic Plxnd1 signaling and thus be hypersensitive to antiangiogenic insults, precisely what our data shows. Accordingly, we highlight that the GIPC-based modulation of endothelial PLXND1 signaling is likely to be also of particular relevance in contexts with minimal proangiogenic stimulation.

4.5) We agree that the embryo in Figure 3B does appear larger. We could show the image of a different embryo instead. However, we note that all the images were taken using the same magnification. Importantly, our angiogenesis quantification scales (see authors’ reply 5.2) are based on the relative span of the vascular structure(s) scored, precisely to eliminate the noise that variations in embryo size would otherwise introduce.

Likewise, why is there such a big variance in phenotype distribution when scoring gipc1 skt1(MZ) mutant embryos? Compare Figure 4B with Figure 5C.It would also be helpful if the authors would indicate e.g. in Figure 2 or 3 and clearly define in the text what constitutes a maximal, a moderate or a minimal Se-truncation.

5.1) We clarify that it is invalid to compare the phenotype distributions of the *gipc1^skt1(MZ)^*mutants shown in Figures 4B and 5C because the angiogenesis deficits in these figures were scored using different scales, as indicated in their respective figure legends. Specifically, Figure 4B and Supplementary File 5 show classes of Se truncations (for example, in *gipc1^skt1(MZ^)* mutants 15.53% of the Se vessels are truncated). In contrast, Figure 5C and Supplementary File 6 show classes of Se-DLAV truncations (for example, in *gipc1^skt1(MZ)^*mutants 18.7% of Se-DLAV are truncated).

5.2) To clarify, the definitions for the truncation classes for both scoring scales are in the manuscript. In the Materials and methods, the section entitled “Quantification of angiogenesis deficits in the trunk’s arterial tree of WT and mutant zebrafish embryos” describes the definitions used for the scoring of “Se truncations” and “Se and DLAV truncations”. The Se truncation scale emphasizes ventral defects (Se truncation scale), while the Se-DLAV truncation scale emphasizes dorsal defects. Importantly, we do not compare the data in Figure 4 with that of Figure 5, as this would be inappropriate.

The authors claim that Gipc binding limits the antiangiogenic signaling of the PlxnD1 receptor. However, there are too many assumptions and too little data for such a statement.For one: what is the antiangiogenic Plxnd1 signaling: where is that shown just based on a poorly controlled hypersensitivity to VegfR2 receptor inhibition, which was done for a very long time period? or based on the proposed hypomorphic identity of the plxnd1 skt6 allele, which as mentioned above is not clearly a significant phenotype.

6.1) We speak of “antiangiogenic Plxnd1 signaling” because multiple published studies in both zebrafish and mice justify the use of this term. Moreover, we agreed to use this term in the editorially approved plan for our revision. First, inactivation of Plxnd1 increases angiogenesis (leads to the formation of more, ectopic, excessively branched and misguided Se sprouts). See the Introduction and Results and references cited therein. Second, ligand stimulation of PlxnD1 exerts a molecular effect opposite to that induced by proangiogenic VEGF signaling (reduces ERK activity, rather than increasing it). See Results and references cited therein. Finally, we agreed to use the term “Plxnd1 signaling”. In the editorially approved plan for our revision, part of our answer to essential revision 1A reads: “We will revise the text to highlight that PlxnD1 is anti-angiogenic, avoiding the use of the terms “activity” and “potency”. We will instead speak of “PlxnD1 signaling”.

6.2) To clarify, our use of the term “antiangiogenic Plxnd1 signaling” is unrelated to the proposed nature of the *plxnd1^skt6^*allele and the hypersensitivity of *plxnd1^skt6^*homozygotes to SU5416 (VegfR2 inhibitor).

6.3) The 14-hr span of our low dose SU5416 treatment (from 18 to 32 hpf) is within the range of similar studies and is designed to target both Se and DLAV angiogenesis specifically. It is unclear why the reviewer considers the treatment overtly long. In reference #94 (Figure 1 and Materials and methods) the authors treat for 12 hours longer (24-hours; from shield stage (6 hpf) to 30 hpf). In reference #161 (Materials and methods) the authors treat for 2 hours less (12-hr; from 18 to 30 hpf). Furthermore, we used a lower SU5416 dose than the studies cited above, which diminishes the likelihood of non-specific effects. For references about the timing of Se and DLAV angiogenesis, please see references #5, 6, 10, 82 and PMIDs 22899709 and 26588168.

6.4) To clarify, we do not propose that the *plxnd1^skt6^*allele is hypomorphic. On the contrary, our data support the notion that this mutant allele is hypermorphic. Please see the authors’ replies 1.1-1.2). Please see also the comment by reviewer #2 saying that: “the new skt6 allele of plexinD1 was used in very clear experiments to show it is functional and hypermorphic”.

The analysis of the gipc mutants is worthwile, however there is too little evidence supporting the Gipc/Plexin interaction on a phenotypic level, with regards to their collaborating role in angiogenesis.An alternative explanation for the partial reduction in phenotype would be that while in gipc1 skt1 Mz embryos gipc2 and gipc3 can still bind to Wt Plexin and therefore are less available for interaction with another recptor (like Nrp1) in the plxnd1 fov01b heterozygous less plexin molecules are available to limit the gipc signaling pool. This would also explain the reduction in phenotype, and be based on a Plexin/gipc interaction,but say nothing about the actual signaling cascades required for generating an angiogenesis phenotype (see our comment regarding essential revision 4B).

7.1) To clarify, we do not say that the role of GIPCs and Plxnd1 is collaborative. We state that GIPCs limit Plxnd1 signaling; see manuscript’s title, manuscript’s impact statement and at various points throughout the text.

7.2) The reviewer’s alternative model about why *plxnd1* heterozygosity suppresses the angiogenesis deficits of *gipc1^skt1(MZ)^*mutants (Figure 5) can explain this genetic interaction, but not the data in Figure 3 (the SU5416 hypersensitivity of *plxnd1^skt6^*mutants). In the reviewer’s model, GIPCs are rate limiting and *PlxnD1* acts like a sink for GIPCs, thereby reducing the amount of GIPC available for VEGFR2/Nrp1 signaling. Given that the receptor encoded by *plxnd1^skt6^*is deficient in GIPC binding, it should not act as a sink for GIPCs (or do so minimally). Hence, the reviewer’s model predicts that in *plxnd1^skt6^*mutants the pool of GIPC available for VEGFR2/Nrp1 signaling should be greater and thus the mutants should be insensitive or less sensitive to the antiangiogenic VEGFR2 inhibitor. However, Figure 3, Figure 3—figure supplement 1 and Supplementary File 4 show that *plxnd1^skt6^*mutants are instead hypersensitive to SU5416 (the opposite result predicted by the reviewer’s hypothesis). In contrast, the model that GIPC binding to Plxnd1 (see Figure 1) limits the antiangiogenic activity of the latter is consistent with all the zebrafish (Figures 2-6 and corresponding figure supplements and supplementary files) and cell culture data (Figure 7 and corresponding figure supplements and supplementary files) in our manuscript. See also authors’ reply to the summary of the reviewers’ discussion (E), authors’ replies 1.2, 3.5, 4.3-4.4, 7.3, 8.1-8.2, 9.1-9.2 and the comment by reviewer #2 (comment #12).

7.3) We highlight that our results and interpretations do not argue against the notion that GIPCs also have vascular effects that are PlxnD1-independent or Nrp1-dependent, as repeatedly stated in the manuscript.

As the interaction studies between Gipc1/2 and plxnd1 are only analyzed qualitatively, meaning that you find massive ectopic sprouting, how were you expecting to see a double phenotype in case of no interaction? the gipc1skt1(Mz); gipc2 skt3/skt4 double phenotype did affect less than 20% of the embryos, only about 10% were affected medium to severely. that would mean that in case of NO interaction you would have found 10% of the Se's being truncated, but still having massive ectopic filopodia and sprouts. Given that there would be additional ectopic sprouts coming from the neighboring Se's, I doubt that this would have been detectable. I therefore believe that the experiment is not suitable to disprove alternative hypothesis…

8.1) The HUVEC data in Figure 7E addresses the effect of simultaneous inactivation of *PLXND1* and *GIPC* quantitatively. Importantly, PLXND1 inactivation and the double loss of *PLXND1* and *GIPC* induce molecular phenotypes that are not statistically significantly different. At 15 and 45 min of SEMA3E stimulation, the relative ERK activity of GIPC knockdown cells (green bars) is significantly different from that of *PLXND1* KO (knockout) cells treated with either non-targeting shRNAs (red bars) or GIPC shRNAs (blue bars). However, the relative ERK activity of *PLXND1* KO (knockout) cells treated with non-targeting shRNAs (red bars) or treated with *GIPC* shRNAs (blue bars) is not significantly different. In other words, *PLXND1* KO cells with *GIPC* knockdown (blue bars) do not show an “intermediate” pERK level between that of PLXND1 KO cells (red bars) and *GIPC* knockdown cells (green bars). Instead, they show a level of relative ERK activity like that of *PLXND1* KO cells (red bars).

8.2) The results of the qualitative phenotypic analysis of MZ *gipc1; gipc2* double mutants with and without morpholino induced *plxnD1* knockdown are consistent with the model (rather than prove it). Our manuscript says that the fact that double maternal-zygotic (MZ) gipc mutants show excessive angiogenesis in the absence of Plxnd1 signaling is “consistent with the model that the angiogenesis deficits of gipc mutants are, at least partially, due to increased PLXND1 signaling”. Please see also the authors’ reply 7.2. Moreover, the experiment followed the editorially approved plan of our revision. Specifically, in our reply to essential revision 4A, we said that “…it is impossible to score the abundance and length of Se vessels at 32 hpf, we must evaluate the vascular abnormalities qualitatively”.

In sum there is not enough evidence supporting a model whereby Gipc limits Plexind1 antiangiogenic signaling, as Gipc could still act on angiogenesis in a Plexin independent manner, eg via neuropilin (see essential revisions 6!). I don't think that the authors have sufficiently experimentally addressed this question. However, this affects the core of the authors claims by directly affecting the functionality of the Plexin/Gipc interactions in angiogenesis.

9.1) We provide strong experimental support for the model that GIPCs directly limit antiangiogenic Plxnd1 signaling. Our model does not imply that GIPCs lack Neuropilin-dependent roles in angiogenesis and experiments directly addressing the involvement of GIPCs in Neuropilin-mediated VEGFR2 signaling are not part of the editorially approved plan of our revision. As summarized in the authors’ reply 7.2, the results of our fish and cell culture experiments support the notion that GIPCs directly limit antiangiogenic Plxnd1 signaling. Our model and results are compatible with the notion that GIPCs also play Plxnd1-independent vascular roles, including Neuropilin-dependent functions (see authors’ reply 7.3).

Moreover, in essential revision 6C of the editorially-approved plan for our revision, a reviewer wonders whether we could distinguish the negative regulation of Plexin from a potential positive regulation of Neuropilin. This reviewer acknowledges the caveat that “there are no reliable Neuropilin reagents in zebrafish available,” concluding that we should “provide more discussion of alternate possibilities that could explain the observed ways in which GIPC function affects PlxnD1 activity”. Accordingly, the Introduction and Discussion detail the relationship between GIPC1 and NRP1-mediated VEGF signaling and highlight that GIPCs have many partners, including other modulators of vascular development. See also our reply to essential revision 6A in the editorially approved plan for our revision and authors’ reply to the summary of the reviewer’s discussion (E).

9.2) The SU5416 hypersensitivity of *plxnd1^skt6^*mutants and the results of the HUVEC experiments directly argue that GIPCs act in a PLXND1-dependent fashion. Note that the HUVEC experiments were done without exogenous VEGF stimulation to avoid simultaneous activation of Neuropilin1/GIPC1-mediated VEGFR2 signaling. To review the arguments related to the SU5416 hypersensitivity, please see the authors’ reply 7.2 (and also the authors’ replies 1.2, 4.3-4.4). Importantly, the cell culture experiments with HUVECs provide a unique opportunity to interpret PLXND1 signaling without exogenous VEGF. Given that GIPC1 association with Neuropilin1 is thought to facilitate VEGFR2 signaling, the reviewer’s hypothesis predicts that GIPC depletion should not affect SEMA3E-induced, PLXND1-dependent responses in HUVECs. However (and in support of our model), GIPC depletion potentiates SEMA3E-induced, PLXND1-dependent responses, leading to a further decrease in relative pERK abundance and cell hypercollapse (see Figure 7 and Figure 7—figure supplements 1-4).

Reviewer #2:The authors have cleared up misconceptions created by language and elegantly used hypotheses to set up outcomes of their experiments and what this means. This is a large improvement over the previous version of the manuscript.The new skt6 allele of plexinD1 was used in very clear experiments to show it is functional and hypermorphic. There is also a plexinD1 null HUVEC line generated as well for the cell culture experiments.Graphs have been improved. Numbers increased and emphasized.Improved discussion of Burk results in context of the results of this manuscript.Overall, there is a huge improvement to the manuscript. Many figures are edited or completely new and there is new data to answer my major points.

We thank you for your help and are delighted to know that you appreciate how all the new reagents, experiments, and data that we collected and how these satisfactorily test the proposed model.

[Editors’ note: the author responses to the re-review follow.]

I have read your rebuttal letter and conferred with the Reviewing Editor. We both agree that it would be appropriate for you to submit a revised version of the manuscript that addresses the comments of the reviewers.We would like to clarify that both reviewer 1 and 2 were in good agreement during the second round of review. Therefore, we urge you to address their comments to the best of your ability. However, given your concerns we will not return the revised manuscripts to either reviewer, but will instead ask previous reviewer 3 (who did not respond to our previous request to re-review), plus a new reviewer. Of course we cannot guarantee the outcome but will do all that we can to be fair.

The re-revised manuscript addresses the comments that reviewer 1 and reviewer 2 provided after our initial revision. Hence, the present manuscript includes several changes. Please note that these modifications include an increased number of references and changes to their numbering.

We also modified two figures based on the feedback of these two reviewers. Briefly, the image in Figure 3B was changed. Also, the penetrance graph in Figure 5—figure supplement 1A was updated by including an asterisk to indicate that the penetrance of Se-DLAV truncations is significantly different between the two genotypes under.

Moreover, we also updated Supplementary Files 3-6. We inserted comments into each of these re-revised Supplementary Files explaining the rationale behind the changes that we performed.

[Editors' note: further revisions were requested prior to acceptance, as described below.]

As you can see from the reviewers' comments (copied below), the reviewers feel that the manuscript has been improved, but both agree that there are some remaining issues that need to be addressed before acceptance. The reviewers have discussed these issues with each other after submitting their reviews, and they agree that it is particularly important for the manuscript to be thoroughly edited, with the goal of creating a more succinct and streamlined experience for the reader.

Done. We edited the text in several places as follows.

Results’ second section (entitled “Zebrafish *plxnd1^skt6^* homozygous mutants, which express a…”): We revised the last line of the third paragraph and revised the sixth (last) paragraph.

Results’ third section (entitled “Homozygous *plxnd1^skt6^* mutants are hypersensitive…”): We revised the first line of the third paragraph and the last three lines (fourth and fifth paragraphs).

Results’ seventh (last) section (entitled “GIPC depletion potentiates …”): We performed several minor edits to the third, fourth and fifth paragraphs.

Discussion section: We edited the first line of the first paragraph and the last line of the third paragraph.

The reviewers also agree that you should make a series of specific adjustments to the text and figures in an effort to clarify or modify the message, in accordance with each of the individual points that have been raised in the reviews below.

Done. See our replies regarding the comments of reviewer 3 and reviewer 4.

Finally, regarding reviewer 4's suggestion that you perform an additional co-IP experiment (see comment #1 from reviewer 4 below), the reviewers agree that this is not essential. If you have the data at hand, please do include it, but it will also be acceptable to simply make changes in the text describing the biochemical effects of this mutation. In that case, the idea that this mutant version of the zebrafish protein loses its ability to interact with Gipc proteins should be presented as speculation, not as fact.

We do not have such CoIP data. Thus, we changed the Abstract by editing the line that said “Zebrafish with a truncated Plxnd1 receptor that is deficient in GIPC binding…” to “We found that zebrafish that endogenously express a Plxnd1 receptor putatively impaired in GIPC binding…” (fifth line). We also made other minor revisions to the Abstract to improve its readability.

Second, we also changed the title of the Results second section to “Zebrafish *plxnd1^skt6^* homozygous mutants, which express a Plxnd1 receptor with a predicted impairment in GIPC binding, display angiogenesis deficits with low frequency”. We also changed the end of the first paragraph of this section to: “…the *plxnd1^skt6^* mutant allele is expected to encode a Plxnd1 receptor with reduced or null GIPC binding ability.”

Third, we edited the title of Figure 2, which now reads: “The *plxnd1^skt6^* allele encodes a functional Plxnd1 receptor putatively impaired in GIPC binding, and its homozygosity induces angiogenesis deficits with low frequency.”

Reviewer #3:After carefully reading the full history of this manuscript, I tend to agree with the previous assessment of reviewer #2.I believe the revised version provided by the authors successfully addressed the essential concerns raised during the first round of review. Moreover, I understand that the data added in the revised version was approved in advance in the context of the revision plan.While I agree with reviewer #1, that some issues of statistical significance were not fully solved, the authors did a good job not only in providing extensive explanations for the observed results, but also by listing alternative hypotheses. In my view, the faithful presentation of the results, and the efforts made to explain them, are valuable, despite of the phenotypes themselves being not hugely penetrant.As agreed by all 3 reviewers, the findings of this manuscript are important and add to our understanding of the mechanisms underlying Plexind1 signalling. Since most of the initial concerns were addressed properly, and the findings are carefully interpreted, I tend to support publication.I do think however that the title/impact statement/conclusions could be toned down a bit to reflect more accurately the main findings.

Done. The impact statement now reads: “Zebrafish mutants and human endothelial cell experiments reveal that GIPC family endocytic adaptors bind to the Semaphorin receptor PLEXIND1, a critical regulator of vascular development, to negatively modulate its signaling.”